# Near-Optimal Offline Reinforcement Learning via Double Variance Reduction

**Ming Yin** [1,3], **Yu Bai**[2], and **Yu-Xiang Wang**[1]

[1]Department of Computer Science, UC Santa Barbara
[2]Salesforce Research
[3]Department of Statistics and Applied Probability, UC Santa Barbara
ming_yin@ucsb.edu   yu.bai@salesforce.com   yuxiangw@cs.ucsb.edu

## Abstract

We consider the problem of offline reinforcement learning (RL) — a well-motivated setting of RL that aims at policy optimization using only historical data. Despite its wide applicability, theoretical understandings of offline RL, such as its optimal sample complexity, remain largely open even in basic settings such as *tabular* Markov Decision Processes (MDPs). In this paper, we propose *Off-Policy Double Variance Reduction* (OPDVR), a new variance reduction based algorithm for offline RL. Our main result shows that OPDVR provably identifies an $\epsilon$-optimal policy with $\widetilde{O}(H^2/d_m\epsilon^2)$ episodes of offline data in the finite-horizon *stationary transition* setting, where $H$ is the horizon length and $d_m$ is the minimal marginal state-action distribution induced by the behavior policy. This improves over the best known upper bound by a factor of $H$. Moreover, we establish an information-theoretic lower bound of $\Omega(H^2/d_m\epsilon^2)$ which certifies that OPDVR is optimal up to logarithmic factors. Lastly, we show that OPDVR also achieves rate-optimal sample complexity under alternative settings such as the finite-horizon MDPs with non-stationary transitions and the infinite horizon MDPs with discounted rewards.

## 1 Introduction

Offline reinforcement learning (offline RL) aims at learning the near-optimal policy by using a static offline dataset that is collected by a certain behavior policy $\mu$ [Lange et al., 2012]. As offline RL agent has no access to interact with the environment, it is more widely applicable to problems where online interaction is infeasible, *e.g.* when trials-and-errors are expensive (robotics, education), risky (autonomous driving) or even unethical (healthcare) [see,e.g., a recent survey  Levine et al., 2020].

Despite its practical significance, a precise theoretical understanding of offline RL has been lacking. Previous sample complexity bounds for RL has primarily focused on the online setting [Azar et al., 2017, Jin et al., 2018, Zanette and Brunskill, 2019, Simchowitz and Jamieson, 2019, Efroni et al., 2019, Cai et al., 2019] or the generative model (simulator) setting [Azar et al., 2013, Sidford et al., 2018a,b, Yang and Wang, 2019, Agarwal et al., 2020, Wainwright, 2019, Lattimore and Szepesvari, 2019], both of which assuming interactive access to the environment and not applicable to offline RL. On the other hand, the sample complexity of offline RL remains unsettled even for environments with finitely many state and actions, a.k.a, the tabular MDPs (Markov Decision Processes). One major line of work is concerned with the off-policy evaluation (OPE) problem [Li et al., 2015, Jiang and Li, 2016, Liu et al., 2018, Kallus and Uehara, 2019a,b, Uehara and Jiang, 2019, Xie et al., 2019, Yin and Wang, 2020, Duan and Wang, 2020]. These works provide sample complexity bounds for evaluating the performance of a fixed policy, and do not imply guarantees for policy optimization. Another line of work studies the sample complexity of offline policy optimization in conjunction with function

35th Conference on Neural Information Processing Systems (NeurIPS 2021).

Table 1: Comparison of sample complexities for tabular offline RL interpretation.

| Method/Analysis | Setting | Assumptions | Sample complexity[a] |
|---|---|---|---|
| BFVT [Xie and Jiang, 2020a] | $\infty$-horizon | only realizability + MDP concentrability[b] | $\tilde{O}((1-\gamma)^{-8}C^2/\epsilon^4)$ |
| MBS-PI/QI [Liu et al., 2020b] | $\infty$-horizon | completeness+bounded density estimation error | $\tilde{O}((1-\gamma)^{-8}C^2/\epsilon^2)$ |
| Le et al. [2019] | $\infty$-horizon | Full Concentrability | $\tilde{O}((1-\gamma)^{-6}\beta_\mu/\epsilon^2)$ |
| FQI [Chen and Jiang, 2019] | $\infty$-horizon | Full Concentrability | $\tilde{O}((1-\gamma)^{-6}C/\epsilon^2)$ |
| MSBO/MABO [Xie and Jiang, 2020b] | $\infty$-horizon | Full Concentrability | $\widetilde{O}((1-\gamma)^{-4}C_\mu/\epsilon^2)$ |
| OPEMA [Yin et al., 2021] | $H$-horizon non-stationary | Full Concentrability | $\widetilde{O}(H^3/d_m\epsilon^2)$ |
| OPDVR (Section 3) | $H$-horizon non-stationary | Weak Coverage | $\widetilde{O}(H^3/d_m\epsilon^2)$ |
| OPDVR (Section 4) | $H$-horizon stationary | Weak Coverage | $\widetilde{O}(H^2/d_m\epsilon^2)$ |
| OPDVR (Section 4.2) | $\infty$-horizon | Weak Coverage | $\widetilde{O}((1-\gamma)^{-3}/d_m\epsilon^2)$ |

[a] Number of episodes in the finite horizon setting and number of steps in the infinite horizon.
[b] $\beta_\mu, C, C_\mu, 1/d_m$ are the concentrability-type coefficients that measure the state-action coverage. See Assumption 2.1 and also Section F.2 for discussions.

approximation [Chen and Jiang, 2019, Xie and Jiang, 2020b,a, Jin et al., 2020]. These results apply to offline RL with general function classes, but when specialized to the tabular setting, they give rather loose sample complexity bounds with suboptimal dependencies on various parameters [1].

The recent work of Yin et al. [2021] showed that the optimal sample complexity for finding an $\epsilon$-optimal policy in offline RL is $\widetilde{O}(H^3/d_m\epsilon^2)$ in the finite-horizon *non-stationary* setting (with matching upper and lower bounds), where $H$ is the horizon length and $d_m$ is a constant related to the data coverage of the behavior policy in the given MDP. However, the optimal sample complexity in alternative settings such as stationary transition or infinite-horizon settings remains unknown. Further, the $\widetilde{O}(H^3/d_m\epsilon^2)$ sample complexity is achieved by an off-policy evaluation + uniform convergence type algorithm; other more practical algorithms including (stochastic) optimal planning algorithms such as Q-Learning are not well understood in offline RL.

**Our Contributions** In this paper, we propose an algorithm ***OPDVR*** (Off-Policy Doubled Variance Reduction) for offline reinforcement learning based on an extension of the variance reduction technique initiated in [Sidford et al., 2018a, Yang and Wang, 2019]. ***OPDVR*** performs stochastic (minibatch style) value iterations using the available offline data, and can be seen as a version of stochastic optimal planning that interpolates value iteration and Q-learning. Our main contributions are summarized as follows.

- We show that ***OPDVR*** finds an $\epsilon$-optimal policy with high probability using $\widetilde{O}(H^2/d_m\epsilon^2)$ episodes of offline data (Section 4.1). This improves upon the best known sample complexity by an $H$ factor and to the best of our knowledge is the first that achieves an $O(H^2)$ horizon dependence, thus separating the stationary case with the non-stationary case for offline RL.
- We establish a sample (episode) complexity lower bound $\Omega(H^2/d_m\epsilon^2)$ for offline RL in the finite-horizon stationary setting (Theorem 4.2), showing that the sample complexity of ***OPDVR*** is optimal up to logarithmic factors.
- In the finite-horizon non-stationary setting, and infinite horizon $\gamma$-discounted setting, we show that ***OPDVR*** achieves $\widetilde{O}(H^3/d_m\epsilon^2)$ sample (episode) complexity (Section 3) and $\widetilde{O}((1-\gamma)^{-3}/d_m\epsilon^2)$ sample complexity (Section 4.2) respectively. They are both optimal up to logarithmic factors and our infinite-horizon result improves over the best known results, e.g., those derived for the fitted Q-iteration style algorithms [Xie and Jiang, 2020b].
- On the technical end, our algorithm presents a sharp analysis of offline RL with stationary transitions, and, *importantly*, the use of the doubling technique to resolve the initialization dependence defect which fails to make the original variance reduction algorithm of [Sidford et al., 2018a] to be optimal, see Appendix F.4. Running [Sidford et al., 2018a] may not yield the desired accuracy as they stated and our result is robust in preserving the optimality.

**Related work.** There is a large and growing body of work on the theory of offline RL and RL in general. We could not hope to provide a comprehensive survey, thus will instead highlight the few

---

[1] See Table 1 for a clear comparison.

prior work that we depend upon on the technical level. The variance reduction techniques that we use in this paper builds upon the work of [Sidford et al., 2018a] in the generative model setting, though it is nontrivial in adapting their techniques to the offline setting; and our two-stage variance reduction appears essential for obtaining optimal rate for $\epsilon > 1$ (see Section 5 and Appendix F.4 for more detailed discussions). We also used a fictitious estimator technique that originates from the OPE literature[Xie et al., 2019, Yin and Wang, 2020], but extended it to the stationary-transition case, and to the policy optimization problem. As we mentioned earlier, the optimal sample complexity in offline RL in the tabular MDPs with stationary transitions was not settled. The result of [Yin et al., 2021] is optimal in the non-stationary case, but is suboptimal by a factor of $H$ in the stationary case. Our lower bound is a variant of the construction of [Yin et al., 2021] that applies to the stationary case. Other existing work on offline RL has even weaker parameters (sometimes due to their setting being more general, see details in Table 1). We defer more detailed discussion related to the OPE literature and online RL / generative model literature to Appendix A due to space constraint.

## 2 Preliminaries

We consider reinforcement learning problems modeled by finite Markov Decision Processes (MDPs) (we focus on the finite-horizon episodic setting, and defer the infinite-horizon discounted setting to Section 4.2.) An MDP is denoted by a tuple $M = (\mathcal{S}, \mathcal{A}, r, T, d_1, H)$, where $\mathcal{S}$ and $\mathcal{A}$ are the state and action spaces with finite cardinality $|\mathcal{S}| = S$ and $|\mathcal{A}| = A$. $P_t : \mathcal{S} \times \mathcal{A} \times \mathcal{S} \to [0,1]$ is the transition kernel with $P_t(s'|s,a)$ be the probability of entering state $s'$ after taking action $a$ at state $s$. We consider both the stationary and non-stationary transition setting: The stationary transition setting assumes $P_t \equiv P$ is identical for all $t \in [H]$, and the non-stationary transition setting allows $P_t$ to be different for different $t$. $r_t : \mathcal{S} \times \mathcal{A} \to [0,1]$ is the reward function which we assume to be deterministic[2]. $d_1$ is the initial state distribution, and $H$ is the time horizon. A (non-stationary) policy $\pi : \mathcal{S} \to \mathbb{P}_{\mathcal{A}}^H$ assigns to each state $s_t \in \mathcal{S}$ a distribution over actions at each time $t$. We use $d_t^\pi(s,a)$ or $d_t^\pi(s)$ to denote the marginal state-action/state distribution induced by policy $\pi$ at time $t$, i.e. $d_t^\pi(s) := \mathbb{P}^\pi(s_t = s)$ and $d_t^\pi(s,a) := \mathbb{P}^\pi(s_t = s, a_t = a)$.

**$Q$-value and Bellman operator.** For any policy $\pi$ and any fixed time $t$, the value function $V_t^\pi(\cdot) \in \mathbb{R}^S$ and $Q$-value function $Q_t^\pi(\cdot, \cdot) \in \mathbb{R}^{S \times A}$, $\forall s, a$ is defined as: $V_t^\pi(s) = \mathbb{E}\left[\sum_{i=t}^H r_i \Big| s_t = s\right], Q_t^\pi(s,a) = \mathbb{E}\left[\sum_{i=t}^H r_i \Big| s_t = s, a_t = a\right]$. For the ease of exposition, we always enumerate $Q^\pi$ as a column vector and similarly for $P_t(\cdot|s,a)$. Moreover, for any vector $Q \in \mathbb{R}^{S \times A}$, the induced value vector and policy is defined in the greedy way: $\forall s_t \in \mathcal{S}, V_Q(s_t) = \max_{a_t \in \mathcal{A}} Q(s_t, a_t)$, $\pi_Q(s_t) = \mathrm{argmax}_{a_t \in \mathcal{A}} Q(s_t, a_t)$. Given an MDP, for any vector $V \in \mathbb{R}^{\mathcal{S}}$ and any deterministic policy $\pi$, $\forall t \in [H]$ the Bellman operator $\mathcal{T}_t^\pi : \mathbb{R}^{\mathcal{S}} \to \mathbb{R}^{\mathcal{S}}$ is defined as: $[\mathcal{T}_t^\pi(V)](s) := r(s, \pi_t(s)) + P_t^\top(\cdot|s, \pi_t(s))V$, and the corresponding Bellman optimality operator $\mathcal{T}_t : \mathbb{R}^{\mathcal{S}} \to \mathbb{R}^{\mathcal{S}}$, $[\mathcal{T}_t(V)](s) := \max_{a \in \mathcal{A}}[r(s,a) + P_t^\top(\cdot|s,a)V]$. Lastly, for a given value function $V_t$, we define backup function $z_t(s_t, a_t) := P_t^\top(\cdot|s_t, a_t)V_{t+1}$ and the one-step variance as $\sigma_{V_{t+1}}(s_t, a_t) := \mathrm{Var}_{s_{t+1}}[V_{t+1}(s_{t+1})|s_t, a_t]$.

**Offline learning problem.** In this paper we investigate the offline learning problem, where we do not have interactive access to the MDP, and can only observe a static dataset $\mathcal{D} = \left\{(s_t^{(i)}, a_t^{(i)}, r_t^{(i)}, s_{t+1}^{(i)})\right\}_{i \in [n]}^{t \in [H]}$. We assume that $\mathcal{D}$ is obtained by executing a pre-specified *behavior policy* $\mu$ (also known as the *logging policy*) for $n$ episodes and collecting the trajectories $\tau^{(i)} = (s_1^{(i)}, a_1^{(i)}, r_1^{(i)}, \ldots, s_H^{(i)}, a_H^{(i)}, r_H^{(i)}, s_{H+1}^{(i)})$, where each episode is rendered in the form: $s_1^{(i)} \sim d_1$, $a_t^{(i)} \sim \mu_t(\cdot|s_t^{(i)})$, $r_t^{(i)} = r(s_t^{(i)}, a_t^{(i)})$, and $s_{t+1}^{(i)} \sim P_t(\cdot|s_t^{(i)}, a_t^{(i)})$. Given the dataset $\mathcal{D}$, our goal is to find an $\epsilon$-**optimal policy** $\pi_{\text{out}}$, in the sense that $||V_1^{\pi^*} - V_1^{\pi_{\text{out}}}||_\infty < \epsilon$.

**Assumption on data coverage** Due to the curse of distributional shift, efficient offline RL is only possible under certain data coverage properties for the behavior policy $\mu$. Throughout this paper we assume the following:

**Assumption 2.1** (Weak coverage). *The behavior policy $\mu$ satisfies the following: There exists some optimal policy $\pi^\star$ such that $d_{t'}^\mu(s_{t'}, a_{t'}) > 0$ if there exists $t < t'$ such that $d_{t:t'}^{\pi^\star}(s_{t'}, a_{t'}|s_t, a_t) > 0$, where $d_{t:t'}^{\pi^\star}(s_{t'}, a_{t'}|s_t, a_t)$ is the conditional multi-step transition probability from step $t$ to $t'$.*

---

[2]This is commonly assumed in the RL literature. The randomness in the reward will only cause a lower order error (than the randomness in the transition) for learning.

Intuitively, Assumption 2.1 requires $\mu$ to "cover" certain optimal policy $\pi^\star$, in the sense that any $s_{t'}, a_{t'}$ is reachable by $\mu$ if it is attainable from a previous state-action pair by $\pi^\star$. It is similar to [Liu et al., 2019, Assumption 1]. Note that this is weaker than the standard "concentrability" assumption [Munos, 2003, Le et al., 2019, Chen and Jiang, 2019]: Concentrability defines $\beta_\mu := \sup_{\pi \in \Pi} ||d^\pi(s_t, a_t)/d^\mu(s_t, a_t)||_\infty < \infty$ (cf. [Le et al., 2019, Assumption 1 & Example 4.1]), which requires the sufficient exploration for tabular case[3] since we optimize over all policies (see Section F.2 for a discussion). In contrast, our assumption only requires $\mu$ to "trace" one single optimal policy.[4]

With Assumption 2.1, we define

$$d_m := \min_{t, s_t, a_t} \{ d_t^\mu(s_t, a_t) : d_t^\mu(s_t, a_t) > 0 \}, \tag{1}$$

which is decided by the behavior policy $\mu$ and is required by offline learning (see Theorem G.2 in Yin et al. [2021]). In general, $d_m$ is unknown yet we assume $d_m$ is *known* for the moment and will utilize $d_m$ in our algorithms. Indeed, in Lemma 5.1, we show that estimating $d_m$ up to a multiplicative factor only requires $\widetilde{O}(1/d_m)$ episodes of offline data; replacing the exact $d_m$ with this estimator suffices for our purpose and, importantly, will not affect our downstream sample complexities.

## 3 Variance reduction for offline RL

In this section, we introduce our main algorithm Off-Policy Double Variance Reduction (*OPDVR*), and present its theoretical guarantee in the finite-horizon non-stationary setting.

### 3.1 Review: variance reduction for RL

We begin by briefly reviewing the variance reduction algorithm for online reinforcement learning, where we have the interactive access to the environment.

Variance reduction (VR) initially emerged as a technique for obtaining fast convergence in large scale optimization problems, for example in the Stochastic Variance Reduction Gradient method (SVRG, [Johnson and Zhang, 2013, Zhang et al., 2013]). This technique is later brought into reinforcement learning for handling policy evaluation [Du et al., 2017] and policy optimization problems [Sidford et al., 2018b,a, Yang and Wang, 2019, Wainwright, 2019, Sidford et al., 2020, Li et al., 2020, Zhang et al., 2020].

In the case of policy optimization, VR is an algorithm that approximately iterating the Bellman optimality equation, using an inner loop that performs an approximate value (or Q-value) iteration using fresh interactive data to estimate $V^\star$, and an outer loop that performs multiple steps of such iterations to refine the estimates. Concretely, to obtain an reliable $Q_t(s, a)$ for some step $t \in [H]$, by the Bellman equation $Q_t(s, a) = r(s, a) + P_t^\top(\cdot|s, a)V_{t+1}$, we need to estimate $P_t^\top(\cdot|s, a)V_{t+1}$ with sufficient accuracy. VR handles this by decomposing:

$$P_t^\top(\cdot|s, a)V_{t+1} = P_t^\top(\cdot|s, a)(V_{t+1} - V_{t+1}^{\mathrm{in}}) + P_t^\top(\cdot|s, a)V_{t+1}^{\mathrm{in}}, \tag{2}$$

where $V_{t+1}^{\mathrm{in}}$ is a *reference* value function obtained from previous calculation (See line 4,13 in the inner loop of Algorithm 1) and $P_t^\top(\cdot|s, a)(V_{t+1} - V_{t+1}^{\mathrm{in}})$, $P_t^\top(\cdot|s, a)V_{t+1}^{\mathrm{in}}$ are estimated separately at different stages. This technique can help in reducing the "effective variance" along the learning process (see Wainwright [2019] Section 2 for a discussion).

In addition, in order to translate the guarantees from learning values to learning policies[5], we build on the following "monotonicity property": For any policy $\pi$ that satisfies the monotonicity condition $V_t \leq \mathcal{T}_{\pi_t} V_{t+1}$ for all $t \in [H]$, the performance of $\pi$ is sandwiched as $V_t \leq V_t^\pi \leq V_t^\star$, i.e. $\pi$ is guaranteed to perform the same or better than $V_t$. This property is first captured by [Sidford et al., 2018a] (for completeness we provide a proof in Lemma B.1), and later reused by Yang and Wang [2019], Sidford et al. [2020] under different settings. We rely on this property in our offline setting as well for providing policy optimization guarantees.

---

[3] Note Xie and Jiang [2020b] has a tighter concentration coefficient with $C_\mu := \max_{\pi \in \Pi} \left\| w_{d_\pi/\mu} \right\|_{2,\mu}^2$ but it still requires full exploration when $\Pi$ contains all policies.

[4] We admit that function approximation+concentrability assumption is powerful for handling realizability/agnostic case and related concepts (*e.g.* inherent Bellman error) and easier to scale up to general settings.

[5] Note in general, direct translation of learning a $\epsilon$-optimal value to $\epsilon$-optimal policy will cause additional suboptimal complexity dependency of $H$.

---

**Algorithm 1** OPVRT: A Prototypical Off-Policy Variance Reduction Template

---

1: **Functional input:** Integer valued function $\mathbf{m} : \mathbb{R}_+ \to \mathbb{N}$. Off-policy estimator $\mathbf{z}_t, \mathbf{g}_t$ in function forms that provides lower confidence bounds (LCB) of the two terms in the bootstrapped value function (2).

2: **Static input:** Initial value function $V_t^{(0)}$ and $\pi_t^{(0)}$ (which satisfy $V_t^{(0)} \leq \mathcal{T}_{\pi_t^{(0)}} V_{t+1}^{(0)}$ and $V_{H+1}^{(0)} \equiv 0$).

   A scalar $u^{(0)}$ satisfies $u^{(0)} \geq \sup_t \|V_t^\star - V_t^{(0)}\|_\infty$. Outer loop iterations $K$. Offline dataset $\mathcal{D} = \{\{s_t^{(i)}, a_t^{(i)}, r_t^{(i)}\}_{t=1}^H\}_{i=1}^n$ from $\mu$ as a data-stream where $n \geq \sum_{i=1}^K 2 \cdot \mathbf{m}(u^{(0)} \cdot 2^{-(i-1)})$.

3: ————————————INNER LOOP ————————————-

4: **function** QVI-VR $(\mathcal{D}_1, \mathcal{D}_2^{(1:H)}, V_t^{\text{in}}, \pi^{\text{in}}, \mathbf{z}_t, \mathbf{g}_t, u^{\text{in}})$

5:     $\diamond$ Computing reference with $\mathcal{D}_1$: Initialize $Q_t \leftarrow \mathbf{0} \in \mathbb{R}^{\mathcal{S} \times \mathcal{A}}$ for $t \in [H+1]$.

6:     **for** $t \in [H]$ and each pair $(s_t, a_t) \in \mathcal{S} \times \mathcal{A}$ **do**

7:       $\diamond$ Compute an LCB of $P_t^\top(\cdot|s_t, a_t)V_{t+1}^{\text{in}}$: $z_t \leftarrow \mathbf{z}_t(\mathcal{D}_1, V_{t+1}^{\text{in}}, u^{\text{in}})$

8:     **end for**

9:     $\diamond$ Value Iterations with $\mathcal{D}_2^{(1:H)}$:

10:     **for** $t = H+1, H, ..., 1$ **do**

11:       $\diamond$ Update value function: $V_t \leftarrow \max(V_{Q_t}, V_t^{\text{in}})$.

12:       $\diamond$ Update policy: $\forall s_t$, if $V_t(s_t) = V_t^{\text{in}}(s_t)$, set $\pi_t(s_t) \leftarrow \pi_t^{\text{in}}(s_t)$; else set $\pi_t(s_t) \leftarrow \pi_{Q_t}(s_t)$.

13:       **if** $t \geq 1$ **then**

14:         $\diamond$ LCB of $P^\top(\cdot|s_{t-1}, a_{t-1})[V_t - V_t^{\text{in}}]$: $g_{t-1} \leftarrow \mathbf{g}_{t-1}(\mathcal{D}_2^{(t-1)}, V_t, V_t^{\text{in}}, u^{\text{in}})$.

15:         $\diamond$ Update $Q$ function: $Q_{t-1} \leftarrow r + z_{t-1} + g_{t-1}$

16:       **end if**

17:     **end for**

18: **end function** **and Return:** $V_1, ..., V_H$ and $\pi$

19: ————————————OUTER LOOP ————————————-

20: **for** $i = 1, ..., K$ **do**

21:     Set $m^{(i)} \to \mathbf{m}(u^{(i-1)})$; Get $\mathcal{D}_1$ with size $m^{(i)}$ and $\mathcal{D}_2^{(t)}$ with size $\frac{m^{(i)}}{H}(u^{(i-1)})^2$ for all $t \in [H]$ from the stream $\mathcal{D}$.

22:     Call $V^{(i)}, \pi^{(i)} \leftarrow$ QVI-VR$(\mathcal{D}_1, \mathcal{D}_2, V^{(i-1)}, \pi^{(i-1)}, \mathbf{z}_t, \mathbf{g}_t, u^{(i-1)})$, Set $u^{(i)} \leftarrow u^{(i-1)}/2$.

23: **end for and Output:** $V^{(K)}, \pi^{(K)}$

---

## 3.2 OPDVR: variance reduction for offline RL

We now explain how we design the VR algorithm in the offline setting. Even though our primary novel contribution is for the stationary case (Theorem 4.1), we begin with non-stationary setting for the ease of explaining algorithmic design. We let $\iota := \log(HSA/\delta)$ as a short hand.

**Prototypical offline VR.** We first describe a prototypical version of our offline VR algorithm in Algorithm 1, which we will instantiate with different parameters *twice* (hence the name"Double") in each of the three settings of interest.

Algorithm 1 takes estimators $\mathbf{z}_t$ and $\mathbf{g}_t$ that produce lower confidence bounds (LCB) of the two terms in (2) using offline data. Specifically, we assume $\mathbf{z}_t, \mathbf{g}_t$ are both available in *function forms* in that they take an offline dataset (with an arbitrary size), fixed value function $V_{t+1}, V_{t+1}^{\text{in}}$ and an external scalar input $u$ then return $z_t, g_t \in \mathbb{R}^{\mathcal{S} \times \mathcal{A}}$. $z_t, g_t$ satisfies that

$$z_t(s_t, a_t) \leq P^\top(\cdot|s_t, a_t)V_{t+1}^{\text{in}}, \quad g_t(s_t, a_t) \leq P^\top(\cdot|s_t, a_t)[V_{t+1} - V_{t+1}^{\text{in}}],$$

uniformly for all $s_t, a_t$ with high probability.

Algorithm 1 then proceeds by taking the input offline dataset as a stream of iid sampled trajectories and use an exponentially increasing-sized batches of independent data to pass in $\mathbf{z}_t$ and $\mathbf{g}_t$ while updating the estimated $Q$ value function by applying the Bellman backup operator except that the update is based on a conservative and variance reduced estimated values. Each inner loop iteration backs up from the last time-step and update all $Q_t$ for $t = H, ..., 1$; and each outer loop iteration passes a new batch of data into the inner loop while ensuring reducing the suboptimality gap from the optimal policy by a factor of 2 in each outer loop iteration, provided that the estimators $\mathbf{z}_t + \mathbf{g}_t$ are increasingly more accurate estimates of (2) as the suboptimality gap gets smaller.

**Plug-in estimators and high-confidence LCBs.** The estimators $\mathbf{z}_t$ and $\mathbf{g}_t$ we use for the three different settings are provided in Figure 3.2. They are essentially the natural plug-in estimators of $P_t^\top(\cdot|s, a)(V_{t+1} - V_{t+1}^{\text{in}})$ and $P_t^\top(\cdot|s, a)V_{t+1}^{\text{in}}$ as well as their standard deviation by replacing $P_t$ with $\widehat{P}_t$ except that we use two disjoint splits $\mathcal{D}_1$ and $\mathcal{D}_2^{(1)}, \ldots, \mathcal{D}_2^{(H)}$ for $\mathbf{z}_t$ and $\mathbf{g}_t$ so they remain

Figure 1: The *implementable* "plug-in" lower confidence bound estimators $\mathbf{z}_t$ and $\mathbf{g}_t$.

| Setting | $\mathbf{z}_t(\mathcal{D}_1, V_{t+1}^{\text{in}}, u)$ | $\mathbf{g}_t(\mathcal{D}_2, V_{t+1}, V_{t+1}^{\text{in}}, u)$ |
|---|---|---|
| Non-stationary | $\frac{1}{n_{s_t,a_t}}\sum_{i=1}^m V_{t+1}^{\text{in}}(s_{t+1}^{(i)})\cdot\mathbf{1}_{[s_t^{(i)},a_t^{(i)}=s_t,a_t]} - e_t(s_t,a_t)$ | $\frac{1}{n'_{s_t,a_t}}\sum_{j=1}^l [V_{t+1}(s_{t+1}'^{(j)}) - V_{t+1}^{\text{in}}(s_{t+1}'^{(j)})]\cdot\mathbf{1}_{[s_t'^{(j)},a_t'^{(j)}=s_t,a_t]} - f_t(s_t,a_t,u)$ |
| Stationary | $\frac{1}{n_{s,a}}\sum_{i=1}^m\sum_{u=1}^H V_{t+1}^{\text{in}}(s_{u+1}^{(i)})\cdot\mathbf{1}_{[s_u^{(i)}=s,a_u^{(i)}=a]} - e_t(s,a)$ | $\frac{1}{n'_{s,a}}\sum_{j=1}^l\sum_{u=1}^H [V_{t+1}(s_{u+1}'^{(j)}) - V_{t+1}^{\text{in}}(s_{u+1}'^{(j)})]\cdot\mathbf{1}_{[s_u'^{(j)},a_u'^{(j)}=s,a]} - f_t(s,a,u)$ |
| ∞-Horizon | $\frac{1}{n_{s,a}}\sum_{i=1}^m V^{\text{in}}(s'^{(i)})\cdot\mathbf{1}_{[s^{(i)}=s,a^{(i)}=a]} - e(s,a)$ | $\frac{1}{n'_{s_t,a_t}}\sum_{j=1}^l [V^{(j)}(s'^{(j)}) - V^{\text{in}}(s'^{(j)})]\cdot\mathbf{1}_{[s'^{(j)},a'^{(j)}=s,a]} - f(s,a,u)$ |

| Setting | $\tilde{\sigma}(s_t,a_t)$ | $e_t(s_t,a_t)$ | $f_t(s_t,a_t,u)$ |
|---|---|---|---|
| Non-stationary | $\frac{1}{n_{s_t,a_t}}\sum_{i=1}^m [V_{t+1}^{\text{in}}(s_{t+1}^{(i)})]^2\cdot\mathbf{1}_{[s_t^{(i)},a_t^{(i)}=s_t,a_t]} - \tilde{z}_t^2(s_t,a_t)$ | $\sqrt{\frac{4\tilde{\sigma}_{V_{t+1}^{\text{in}}}\iota}{n_{s_t,a_t}}} + 2\sqrt{6}V_{\max}\left(\frac{\iota}{n_{s_t,a_t}}\right)^{3/4} + 16V_{\max}\frac{\iota}{n_{s_t,a_t}}$ | $4u\sqrt{\frac{\iota}{n'_{s_t,a_t}}}$ |
| Stationary | $\frac{1}{n_{s,a}}\sum_{i=1}^m\sum_{u=1}^H [V_{t+1}^{\text{in}}(s_{u+1}^{(i)})]^2\cdot\mathbf{1}_{[s_u^{(i)}=s,a_u^{(i)}=a]} - \tilde{z}_t^2(s,a)$ | $\sqrt{\frac{4\tilde{\sigma}_{V_{t+1}^{\text{in}}}\iota}{n_{s,a}}} + 2\sqrt{6}V_{\max}\left(\frac{\iota}{n_{s,a}}\right)^{3/4} + 16V_{\max}\frac{\iota}{n_{s,a}}$ | $4u\sqrt{\frac{\iota}{n'_{s,a}}}$ |
| ∞-Horizon | $\frac{1}{n_{s,a}}\sum_{i=1}^m [V^{\text{in}}(s'^{(i)})]^2\cdot\mathbf{1}_{[s^{(i)}=s,a^{(i)}=a]} - \tilde{z}^2(s,a)$ | $\sqrt{\frac{4\cdot\tilde{\sigma}_{V^{\text{in}}}\cdot\iota}{n_{s,a}}} + 2\sqrt{6}\cdot V_{\max}\cdot\left(\frac{\iota}{n_{s,a}}\right)^{3/4} + \frac{16V_{\max}\iota}{3n_{s,a}}$ | $4u\sqrt{\frac{\log(2RSA/\delta)}{n'_{s,a}}}$ |

\* $m, l$ are the number of episodes in $\mathcal{D}_1, \mathcal{D}_2^{(t)}$ respectively. $\iota$ is a logarithmic factor in $HSA/\delta$ in the finite horizon case and $SA/\delta$ in the infinite horizon cases. $n_{s_t,a_t}$ is the number of times $s_t, a_t$ appears at time $t$ in $\mathcal{D}_1$; and $n'(s_t,a_t)$ is the that for $\mathcal{D}_2^{(t)}$. In the case when $n_{s_t,a_t} = 0$, we simply output 0 for all quantities above.

---

**Algorithm 2** (OPDVR) Off-Policy Doubled Variance Reduction

**input** Offline Dataset $\mathcal{D}$ of size $n$ as a stream. Target accuracy $\epsilon, \delta$ such that the algorithm does not use up $\mathcal{D}$.
**input** Estimators $\mathbf{z}_t, \mathbf{g}_t$ in function forms, $m_1', m_2', K_1, K_2$.
1: ◇ Stage 1. coarse learning: a "warm-up" procedure
2: Set $V_t^{(0)} := \mathbf{0}$ and any policy $\pi^{(0)}$. Set initial $u^{(0)} := H$. Set $\mathbf{m}(u) = m_1' \log(16HSAK_1)/u^2$.
3: Run Algorithm 1 with $\mathbf{m}, \mathbf{z}_t, \mathbf{g}_t, V_t^{(0)}, \pi^{(0)}, u^{(0)}, K_1, \mathcal{D}$ and return $V_t^{\text{intermediate}}, \pi^{\text{intermediate}}$.
4: ◇ Stage 2. fine learning: reduce error to given accuracy
5: Reset initial values $V_t^{(0)} := V_t^{\text{intermediate}}$ and policy $\pi^{(0)} := \pi^{\text{intermediate}}$. Set $u^{(0)} := \sqrt{H}$.
6: Reset $\mathbf{m}(u)$ by replacing $m_1'$ with $m_2'$, $K_1$ with $K_2$.
7: Run Algorithm 1 with $\mathbf{m}, \mathbf{z}_t, \mathbf{g}_t, V_t^{(0)}, \pi^{(0)}, u^{(0)}, K_2, \mathcal{D}$ and return $V_t^{\text{final}}, \pi^{\text{final}}$.
**output** $V_t^{\text{final}}, \pi^{\text{final}}$

---

statistically independent. We also remark that $P_t$ is of dimension $S^2 A$, and we operates in the sparse regime where the number of observed samples could be drastically smaller than the total number of coordinates there are. In other words, $\widehat{P}_t$ will be *sparse* as most of its coordinates are 0. The saving grace is that when we fix $V_{t+1}$ and $V_{t+1}^{\text{in}}$, the output space collapsed to only $SA$ dimensional, which ensures that we have sufficient number of data points to produce accurate estimates.

A key difference from the generative model setting is that these estimators are dependent across $t$, thus it requires new technical steps to establish the convergence of these estimators as well as putting them together to show that Algorithm 1 works.

**The doubling procedure.** It turns out that Algorithm 1 alone does not yield a tight sample complexity guarantee, due to its *suboptimal dependence on the initial optimality gap* $u^{(0)} \geq \sup_t \|V_t^\star - V_t^{(0)}\|_\infty$ (recall $u^{(0)}$ is the initial parameter in the outer loop of Algorithm 1). This is captured in the following (for the non-stationary case):

**Proposition 3.1** (Informal version of Lemma B.10). *Suppose* $\epsilon \in (0, 1]$ *is the final target accuracy. Algorithm 1 outputs the $\epsilon$-optimal policy with episode complexity:*

$$\bullet\ \tilde{O}(H^4/d_m\epsilon^2), \quad \text{If } u^{(0)} > \sqrt{H}; \quad \bullet\ \tilde{O}(H^3/d_m\epsilon^2), \quad \text{If } u^{(0)} \leq \sqrt{H}.$$

Proposition 3.1 suggests that Algorithm 1 may have a suboptimal sample complexity when the initial optimality gap $u^{(0)} > \sqrt{H}$. Unfortunately, this is precisely the case for standard initializations such as $V_t^{(0)} := \mathbf{0}$, for which we must take $u^{(0)} = H$. We overcome this issue by designing a two-stage *doubling* procedure: At stage 1, we use Algorithm 1 to obtain $V_t^{\text{intermediate}}, \pi^{\text{intermediate}}$ that are $\epsilon' = \sqrt{H}\epsilon$ accurate; At stage 2, we then use Algorithm 1 again with $V_t^{\text{intermediate}}, \pi^{\text{intermediate}}$ as the input and further reduce the error from $\epsilon'$ to $\epsilon$. The main take-away of this doubling procedure is that the episode complexity of both stage is only $\tilde{O}(H^3/d_m\epsilon^2)$, therefore the total sample complexity optimality is preserved. The pseudo-code of the two-stage procedure **OPDVR** is summarized in Algorithm 2.

### 3.3 *OPDVR* for non-stationary transition settings

We now state our main theoretical guarantee for *OPDVR* in the finite-horizon non-stationary transition setting.

**Theorem 3.2** (*OPDVR* in episodic non-stationary setting). *For the $H$-horizon non-stationary setting, there exist universal constants $c_1, c_2, c_3 > 0$ such that if we set $m_1' = c_1 H^4/d_m$ for Stage 1, $m_2' = c_2 H^3/d_m$ for Stage 2, set $K_1 = K_2 = \log_2(\sqrt{H}/\epsilon)$, take $\mathbf{g}_t$ and $\mathbf{z}_t$ according to Figure 3.2, then* ***OPDVR*** *(Algorithm 2) with probability $1 - \delta$ outputs an $\epsilon$-optimal policy $\hat{\pi}$ provided that the number of episodes in the offline data $\mathcal{D}$ exceeds (below can be readily simplified as $\widetilde{O}\left(H^3/d_m\epsilon^2\right)$):*

$$\frac{c_3 \max[\frac{m_1'}{H}, m_2']}{\epsilon^2}\left(\iota + \log\log_2(\frac{\sqrt{H}}{\epsilon})\right)\log_2(\frac{\sqrt{H}}{\epsilon}),$$

**Optimality of sample complexity.** Theorem 3.2 shows that our *OPDVR* algorithm can find an $\epsilon$-optimal policy with $\widetilde{O}(H^3/d_m\epsilon^2)$ episodes of offline data. Compared with the sample complexity lower bound $\Omega(H^3/d_m\epsilon^2)$ for offline learning (Theorem G.2. in Yin et al. [2021]), we see that our *OPDVR* algorithm matches the lower bound up to logarithmic factors. The same rate was achieved previously by the local uniform convergence argument of Yin et al. [2021] under a stronger assumption of full data coverage.

*Proof sketch of Theorem 3.2.* One challenge in analyzing *OPDVR* is that the number of state-transition samples are random and dependent. The idea of the proof is to first construct *fictitious* versions of estimators $\mathbf{z}_t$ (or $\mathbf{g}_t$), that uses the empirical plug-in formula only if the event $E_{m,t} = \{n_{s_t,a_t} > \frac{1}{2}m \cdot d_t^\mu(s_t, a_t)\}$ (or $E_{l,t}$) are true, i.e., there are sufficient number of samples at $s_t, a_t$. When there aren't, we plug in the *ground truth transition kernel* $P_t$ instead. These fictitious estimators are not implementable in practice, but they simplify the analysis and are central to our extension of the Variance Reduction framework previously used in the generative model setting [Sidford et al., 2018a] to the offline setting. The following proposition shows that practical implementations (summarized in Figure 3.2) are *identical* to these *fictitious* estimators with high probability:

**Proposition 3.3** (Summary of Section B.4). *Under the condition of Theorem 3.2, we have*

$$\mathbb{P}\left[\bigcup_{\substack{i \in [K_1],\\ t \in [H]}}\left(E_{l,t}^{(i)c} \cup E_{m,t}^{(i)c}\right) \bigcup_{\substack{j \in [K_2],\\ t \in [H]}}\left(E_{l,t}^{(j)c} \cup E_{m,t}^{(j)c}\right)\right] \le \delta/2.$$

*This means with probability $1 - \delta/2$, fictitious estimators $\tilde{z}_t, \tilde{g}_t, \tilde{\sigma}$ are all identical to their practical versions (summarized in Figure 3.2).*

By Proposition 3.3, it suffices to analyze the performance of *OPDVR* instantiated with fictitious estimators (60) and (59). Theorem 3.2 then relies on analyzing the the prototypical *OPDVR* (Algorithm 1) with the fictitious estimators. In particular, both off-policy estimators $z_t$ and $g_t$ use lower confidence update to avoid over-optimism and the $\max$ operator in $V_t = \max(V_{Q_t}, V_t^{\text{in}})$ helps prevent pessimism. By doing so the update $V_t$ in Algorithm 1 always satisfies $0 \le V_t \le V_t^\star$, which is always within valid range. The doubling procedure of Algorithm 2 then first decreases the accuracy to a coarse level $\epsilon' = \sqrt{H}\epsilon$, and further lowers it to the given accuracy $\epsilon$. The key technical lemma for achieving optimal dependence in $H$ is Lemma G.5, which bounds the term $\sum_{u=t}^H \mathbb{E}_{s_u,a_u}^{\pi^\star}[\text{Var}[V_{u+1}^\star(s_{u+1})|s_u, a_u]]$ by $O(H^2)$ instead of the naive $O(H^3)$. The full proof of Theorem 3.2 can be found in Appendix B. ∎

## 4 *OPDVR* for stationary transition settings

In this section, we switch gears to the *stationary* transition setting, in which the transition probabilities are identical at all time steps: $P_t(s'|s, a) :\equiv P(s'|s, a)$. We will consider both the (a) finite-horizon case where each episode is consist of $H$ steps; and (b) the infinite-horizon case where the reward at the $t$-th step is discounted by $\gamma^t$, where $\gamma \in (0, 1)$ is a discount factor. These settings encompass additional challenges compared with the non-stationary case, as in theory the transition probabilities can now be estimated more accurately due to the shared information across time steps.

## 4.1 Finite-horizon stationary setting

We begin by considering the finite-horizon stationary setting. As this is a special case of the non-stationary setting, Theorem 3.2 implies that **OPDVR** achieves $\widetilde{O}(H^3/d_m\epsilon^2)$ sample complexity. However, similar as in online RL [Azar et al., 2017], this result may be potentially loose by an $O(H)$ factor, as the algorithm does not take into account the stationarity of the transitions. This motivates us to design an algorithm that better leverages the stationarity by aggregating state-action pairs across different time steps (see the second rows of the two tables in Figure 3.2).

**Theorem 4.1** (Sample complexity of **OPDVR** in finite-horizon stationary setting)**.** *In the $H$-horizon stationary transition setting, there exists universal constants $c_1', c_2', c_3'$ such that if we set $m_1' = c_1'H^3/d_m$, $m_2' = c_2'H^2/d_m$ for Stage 1 and 2, set $K_1 = K_2 = \log_2(\sqrt{H}/\epsilon)$, and take $\mathbf{z}_t$ and $\mathbf{g}_t$ according to Figure 3.2, then with probability $1 - \delta$, Practical **OPDVR** finds an $\epsilon$-optimal policy provided that the number of episodes in the offline data $\mathcal{D}$ exceeds (which is of order $\widetilde{O}\left(\frac{H^2}{d_m\epsilon^2}\right)$):*

$$\frac{c_3' \max[\frac{m_1'}{H}, m_2']}{\epsilon^2}\left(\iota + \log\log_2(\frac{\sqrt{H}}{\epsilon})\right)\log_2(\frac{\sqrt{H}}{\epsilon}),$$

*Proof sketch.* Similar to the proof of the non-stationary case, we start by reducing the problem to a fictitious version that replace the pathological events that happen with low-probability with ground truth. Then the key challenge is how to analyze the stronger fictitious estimators that pools over steps within each roll-out. We design a martingale $X_k = \sum_{i=1}^m \sum_{u=1}^{k-1} \left(V_{t+1}^{\text{in}}(s_{u+1}^{(i)}) - P^\top(\cdot|s,a)V_{t+1}^{\text{in}}\right) \cdot \mathbf{1}[s_u^{(i)} = s, a_u^{(i)} = a].$ under the filtration $\mathcal{F}_k := \{s_u^{(i)}, a_u^{(i)}\}_{i\in[m]}^{u\in[k]}$ for bounding $z_t(s_t, a_t) \leq P^\top(\cdot|s_t, a_t)V_{t+1}^{\text{in}}$. The conditional variance sum $\sum_{k=1}^H \text{Var}[X_{k+1} | \mathcal{F}_k] = \sum_{k=1}^H \sum_{i=1}^m \mathbf{1}\left[s_k^{(i)}, a_k^{(i)} = s, a\right] \text{Var}\left[V_{t+1}^{\text{in}}\left(s_{k+1}^{(i)}\right) | s_k^{(i)}, a_k^{(i)} = s, a\right]$. For stationary case, $s_{k+1}^{(i)} \sim P(\cdot|s_k^{(i)}, a_k^{(i)} = s, a)$ is irrelevant to time $k$ so above equals $\sum_{k=1}^H \sum_{i=1}^m \mathbf{1}\left[s_k^{(i)}, a_k^{(i)} = s, a\right]$ $\sigma_{V_{t+1}^{\text{in}}}(s, a) = n_{s,a} \cdot \sigma_{V_{t+1}^{\text{in}}}(s, a)$, where $V_{t+1}^{\text{in}}$ is later replaced by $V_{t+1}^\star$ and $\sum_{t=1}^H \mathbb{E}_{s,a}^{\pi^\star}[\sigma_{V_t^\star}(s, a)]$ can be bounded by $H^2$ which is tight. In contrast, in non-stationary regime $P_t$ is varying across time so we can only obtain $\sum_{k=1}^H \text{Var}[X_{k+1} | \mathcal{F}_k] \leq n_{s,a} \max_t \sigma_{V_t^{\text{in}}}(s, a)$, which is later translated into $\sum_{t=1}^H \mathbb{E}_{s,a}^{\pi^\star}[\max_t \sigma_{V_t^\star}(s, a)]$ and are of order $H^3$ general. To sum, the fact that $P$ is identical is carefully leveraged multiple times for obtaining $\widetilde{O}(H^2/d_m\epsilon^2)$ rate. The detailed proof of Theorem 4.1 can be found in Appendix C. ∎

Theorem 4.1 encompasses our main technical contribution, as the compact data aggregation among different time steps make analyzing the estimators (61) and (62) knotty due to data-dependence (unlike the non-stationary transition setting where estimators are designed using data at specific time). In particular, we need to fully exploit the property that transition $P$ is identical across different times in a pinpoint way to obtain the $H^2$ dependence in the sample complexity bound.

**Improved dependence on $H$.** Theorem 4.1 shows that **OPDVR** achieves a sample complexity upper bound $\widetilde{O}(H^2/d_m\epsilon^2)$ in the stationary setting. To the best of our knowledge, this is the first result that achieves an $H^2$ dependence for offline RL with stationary transitions, and improves over the $H^3$ dependence in either the (non-stationary) **OPDVR** (Theorem 3.2) or the "off-policy evaluation + uniform convergence" algorithm of Yin et al. [2021]. We exploit specific properties of **OPDVR** in our techniques for knocking off a factor of $H$ and there seems to be no direct ways in applying the same techniques in improving the uniform convergence-style results for the stationary-transition setting.

**Optimality of $\widetilde{O}(H^2/d_m\epsilon^2)$.** We accompany Theorem 4.1 by a establishing a sample complexity lower bound for this setting, showing that our algorithm achieves the optimal dependence of all parameters up to logarithmic factors. The proof of Theorem 4.2 builds on modifying the $\widetilde{O}(H^3/d_m\epsilon^2)$ sample complexity lower bound in Yin et al. [2021], and can be found in Appendix E.

**Theorem 4.2** (Information-theoretic lower bound)**.** *For all $0 < d_m \leq \frac{1}{SA}$, let the family of problem be $\mathcal{M}_{d_m} := \left\{(\mu, M) \mid \min_{t, s_t, a_t} d_t^\mu(s_t, a_t) \geq d_m\right\}$. There exists universal constants $c_1, c_2, c, p$ (with $H, S, A \geq c_1$ and $0 < \epsilon < c_2$) such that when $n \leq cH^2/d_m\epsilon^2$, $\inf_{v^{\pi_{alg}}} \sup_{(\mu, M)\in\mathcal{M}_{d_m}} \mathbb{P}_{\mu, M}\left(v^* - v^{\pi_{alg}} \geq \epsilon\right) \geq p$.*

## 4.2 Infinite-horizon discounted setting

Finally, we consider the infinite-horizon discounted setting. The setting is slightly different to the finite horizon case as we adopt the same assumption of Chen and Jiang [2019], Xie and Jiang [2020b] that data $\mathcal{D} = \{s^{(i)}, a^{(i)}, r^{(i)}, s'^{(i)}\}_{i \in [n]}$ are i.i.d off-policy pieces with $(s, a) \sim d^\mu$ and $s' \sim P(\cdot|s, a)$. The infinite horizon-versions of **OPDVR** (Algorithm 3 and 4) are stated in the Appendix due to the space limit.

**Theorem 4.3** (Sampe complexity of **OPDVR** in infinite-horizon discounted setting). *Consider Algorithm 4. There are constants $c_1', c_2', c_3'$, such that if we set $m_1' = O((1-\gamma)^{-4}/d_m), m_2' = O((1-\gamma)^{-3}/d_m)$ (see more precise expressions in Lemma D.7), $K_1 = \log_2((1-\gamma)^{-1}/\epsilon), K_2 = \log_2(\sqrt{(1-\gamma)^{-1}}/\epsilon)$, $R = \log(4/\epsilon(1-\gamma))$, and choose LCB estimators $\mathbf{z}$ and $\mathbf{g}$ as in Figure 3.2, then with probability $1 - \delta$, the infinite horizon version of **OPDVR** (Algorithm 4) outputs an $\epsilon$-optimal policy provided that in offline data $\mathcal{D}$ has number of samples exceeding*

$$\frac{c_3' \max[\frac{m_1'}{(1-\gamma)^{-1}}, m_2']}{\epsilon^2} \cdot \iota' = \widetilde{O}\left[(1-\gamma)^{-3}/d_m \epsilon^2\right].$$

*where $\iota' := R \cdot (\log(32(1-\gamma)^{-1}RSA/\delta) + \log\log_2(\sqrt{(1-\gamma)^{-1}}/\epsilon)) \cdot \log_2(\sqrt{(1-\gamma)^{-1}}/\epsilon)$.*

We note that for the infinite horizon case, the sample-complexity measures the number of steps, thus $(1-\gamma)^{-3}$ is comparable to the $H^2$ dependence. To the best of our knowledge, Theorem 4.1 and Theorem 4.3 are the first results that achieve $H^2$, $(1-\gamma)^{-3}$ dependence in the offline regime respectively for stationary transition and infinite horizon setting, see Table 1.

## 5 Discussions

**Estimating $d_m$.** It is worth mentioning that the input of **OPDVR** depends on unknown system quantity $d_m$. Nevertheless, $d_m$ is only one-dimensional scalar and thus it is plausible (from a statistical perspective) to leverage standard parameter-tuning tools (*e.g.* cross validation [Varma and Simon, 2006]) for obtaining a reliable estimate in practice. On the theoretical side, we provide the following result to show plug-in on-policy estimator $\widehat{d_t^\mu}(s_t, a_t) = n_{s_t, a_t}/n$ and $\widehat{d_m} := \min_{t, s_t, a_t}\{n_{s_t, a_t}/n : n_{s_t, a_t} > 0\}$, is sufficient for accurately estimating $d_t^\mu, d_m$ simultaneously.

**Lemma 5.1.** *For the finite-horizon setting (either stationary or non-stationary), there exists universal constant $c$, s.t. when $n \geq c \cdot 1/d_m \cdot \log(HSA/\delta)$, then w.p. $1 - \delta$, we have $\forall t, s_t, a_t, \frac{1}{2}d_t^\mu(s_t, a_t) \leq \widehat{d_t^\mu}(s_t, a_t) \leq \frac{3}{2}d_t^\mu(s_t, a_t)$ and, in particular, $\frac{1}{2}d_m \leq \widehat{d_m} \leq \frac{3}{2}d_m$. See Appendix F.1 for the proof.*

Lemma 5.1 ensures one can replace $d_t^\mu$ by $\widehat{d_t^\mu}$ ($d_m$ by $\widehat{d_m}$) in **OPDVR** and we obtain a fully data-adaptive algorithm. Note that the requirement on $n$ does not affect our near-minimax complexity bound in either Theorem 3.2 and 4.1—we only require $n \approx \widetilde{\Theta}(1/d_m)$ additional episodes to estimate $d_m$ and it is of lower order compared to our upper bound $\widetilde{O}(H^3/d_m)$ or $\widetilde{O}(H^2/d_m)$.

**Computational and memory cost. OPDVR** can be implemented as a streaming algorithm that uses only one pass of the dataset. Its computational cost is $\widetilde{O}(H^4/d_m \epsilon^2)$ — the same as its sample complexity in steps ($H$ steps is an episode), and the memory cost is $O(HSA)$ for the episodic case and $O(SA)$ for the stationary or infinite horizon case. In particular, the double variance reduction technique does not introduce additional overhead beyond constant factors.

**Improvement over VR in the generative model setting.** This work extends of the variance reduction framework for RL in the generative model setting [Sidford et al., 2018a, Yang and Wang, 2019]. However, we make two improvements to these works. First, the data collected in the offline case are highly dependent (in contrast in the generative model setting each simulator call is independent), and disentangling the dependent structure makes the offline setting inherently more challenging. Second, our doubling mechanism always guarantee the minimax rate with any initialization and the single VR procedure does not have this property (see Appendix F.4 for a more detailed discussion), which could be a critical issue when Sidford et al. [2018a] claims the optimality.

## 6 Conclusion

This paper proposes **OPDVR** (off-policy double variance reduction), a new variance reduction algorithm for offline reinforcement learning. We show that **OPDVR** achieves tight sample complexity

for offline RL in tabular MDPs; in particular, **OPDVR** is the first algorithm that achieves the optimal sample complexity for offline RL in the stationary transition setting. On the technical end, we present a sharp analysis under stationary transitions, and use the doubling technique to resolve the initialization dependence in variance reduction, both of which could be of broader interest. There are several interesting next directions. For example, can our analysis shed light on other commonly used algorithms (such as Q-Learning) for offline RL? How can we better deal with insufficient data coverage? Can we go beyond the tabular setting? We would like to leave these as future work.

**Acknowledgment**

The authors would like to thank Lin F. Yang for the discussions about [Sidford et al., 2018a]. Ming Yin and Yu-Xiang Wang are partially supported by NSF Awards #2007117 and #2003257. Yu Bai is partly funded through the employment with Salesforce.

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
