# Appendix

## A  More Discussion on Related Work

**Offline reinforcement learning**  There is a growing body of work on offline RL recently [Levine et al., 2020] in both offline policy evaluation (OPE) and offline policy optimization. OPE (also known as Off-Policy Evaluation [Li et al., 2015]) requires estimating the value of a *target policy* $\pi$ from an offline dataset that is often generated using another behavior policy $\mu$. A variety of algorithms and theoretical guarantees have been established for offline policy evaluation [Li et al., 2015, Jiang and Li, 2016, Liu et al., 2018, Kallus and Uehara, 2019a,b, Uehara and Jiang, 2019, Xie et al., 2019, Yin and Wang, 2020, Duan and Wang, 2020, Liu et al., 2020a,b, Feng et al., 2020]. The majority of these work uses (vanilla or more advanced versions of) importance sampling to correct for the distribution shift, or uses minimax formulation to approximate the task and solves the questions through convex/non-convex optimization.

Meanwhile, the offline policy optimization problem needs to find a near-optimal policy given the offline dataset. The study of offline policy optimization can be dated back to the classical Fitted Q-Iteration algorithm [Antos et al., 2008a,b]. The sample complexity for offline policy optimization is studied in a line of recent work [Chen and Jiang, 2019, Le et al., 2019, Xie and Jiang, 2020b,a, Liu et al., 2020b]. The focus on these work is on the combination of offline RL and function approximation; when specialized to the tabular setting, the sample complexities have a rather suboptimal dependence on the horizon $H$ (or $(1 - \gamma)^{-1}$ in the discounted setting). In particular, Chen and Jiang [2019], Le et al. [2019] first established finite sample guarantees with complexity $\widetilde{O}((1 - \gamma)^{-6}\beta_\mu/\epsilon^2)$ (where $\beta_\mu$ is the concentration coefficient) and it is later improved to $\widetilde{O}((1 - \gamma)^{-4}\beta_\mu/\epsilon^2)$ by Xie and Jiang [2020b] with a finer analysis. Later, Xie and Jiang [2020a] considers offline RL under weak *realizability* assumption and Liu et al. [2020b] considers offline RL *without* good exploration. Those are challenging offline settings but their dependence on horizon $(1 - \gamma)^{-1}$ (or $H$) is very suboptimal. The recent work of Yin et al. [2021] (OPE + uniform convergence) first achieves the sample complexity $\widetilde{O}(H^3/d_m\epsilon^2)$ in the finite-horizon non-stationary transition setting for tabular offline RL, and establishes a lower bound $\Omega(H^3/d_m\epsilon^2)$ (where $d_m$ is a constant related to the data coverage of the behavior policy in the given MDP that is similar to the concentration coefficient $\beta_\mu$) that matches the upper bound up to logarithmic factors. Compared with these work, we analyze a new variance reduction algorithm for offline policy learning, which is a more generic approach as it adapts to all typical settings (finite horizon stationary/non-stationary transition, infinite horizon setting) with optimal sample complexity while the technique in Yin et al. [2021] only works for non-stationary setting and cannot directly reduce to $\widetilde{O}(H^2/d_m\epsilon^2)$ when the transition becomes stationary. Concurrent to this work, Jin et al. [2020] study pessimism-based algorithms for offline policy optimization under insufficient coverage of the data and Wang et al. [2020], Zanette [2020] provide some negative results (exponential lower bound) for offline RL with linear MDP structure.

**Reinforcement learning in online settings**  In online RL (where one has interactive access to the environment), the model-based UCBVI algorithm achieves the minimax regret of $\tilde{O}(\sqrt{HSAT})$ [Azar et al., 2017] and is later improved by [Dann et al., 2018]. Later this minimax rate is also achieved by EULER with stronger problem-dependent expressions [Zanette and Brunskill, 2019, Simchowitz and Jamieson, 2019]. Model-free algorithms such a *Q-learning* is able to achieve a $\sqrt{H}$-suboptimal regret comparing to lower bound [Jin et al., 2018] and this gap is recently closed by an improved model-free algorithm in [Zhang et al., 2020].

In the generative model setting (where one has a simulator that samples $(r_t, s_{t+1})$ from any $(s_t, a_t)$, [Azar et al., 2013, Wainwright, 2019] prove sample complexity $\widetilde{O}((1 - \gamma)^{-3}SA/\epsilon^2)$ is sufficient for the output $Q$-function to be $\epsilon$-optimal, *i.e.* $||Q^\star - Q^{\text{out}}||_\infty < \epsilon$, however this does not imply $\epsilon$-optimal policy with the same sample complexity. The most related to our work among this line is [Sidford et al., 2018a], which designs an *variance reduction* algorithm that overcomes the above issue and obtains $\widetilde{O}((1 - \gamma)^{-3}SA/\epsilon^2)$ sample complexity or finding the optimal policy as well. Later [Yang and Wang, 2019] again uses VR to obtain the sample optimality under the linear transition models. The design of our algorithm builds upon the variance reduction technique; our

doubling technique and analysis in the offline setting can be seen as a generalization of [Sidford et al., 2018a]; see Section 5 and Appendix F.4 for more detailed discussions.

## B   Proofs for finite-horizon non-stationary setting

The roadmap of our analysis in this section consists of first doing concentration analysis, then iteratively reasoning using induction, analyzing the doubling procedure and proving from prototypical version to the practical version. Moreover, we also formally point out one defect in Sidford et al. [2018a] later (see Section F.4) to contrast that our doubling VR procedure is necessary.

Even before that, let us start with the simple *monotone preservation* lemma.

**Lemma B.1.** *Suppose $V$ and $\pi$ is any value and policy satisfy $V_t \leq \mathcal{T}_{\pi_t} V_{t+1}$ for all $t \in [H]$. Then it holds $V_t \leq V_t^\pi \leq V_t^\star$, for all $t \in [H]$.*

*Proof.* We only need to show $V_t \leq V_t^\pi$. Since $V_t \leq \mathcal{T}_{\pi_t} V_{t+1}$, we can use it repeatedly to obtain

$$V_t \leq \mathcal{T}_{\pi_t} V_{t+1} \leq \mathcal{T}_{\pi_t}(\mathcal{T}_{\pi_{t+1}} V_{t+2}) \leq ... \leq \mathcal{T}_{\pi_t} \circ \mathcal{T}_{\pi_{t+1}} \circ ... \circ \mathcal{T}_{\pi_H} V_{H+1} \tag{3}$$

where "$\circ$" denotes operator composition. Note by default $V_{H+1} = V_{H+1}^\pi = V_{H+1}^\star = \mathbf{0}$, therefore

$$\mathcal{T}_{\pi_{t+1}} \circ ... \circ \mathcal{T}_{\pi_H} V_{H+1} = \mathcal{T}_{\pi_{t+1}} \circ ... \circ \mathcal{T}_{\pi_H} V_{H+1}^\pi = \mathcal{T}_{\pi_{t+1}} \circ ... \circ \mathcal{T}_{\pi_{H-1}} V_H^\pi = ... = \mathcal{T}_{\pi_t} V_{t+1}^\pi = V_t^\pi \tag{4}$$

where we use the definition of Bellman equation that $V_t^\pi = \mathcal{T}_{\pi_t} V_{t+1}^\pi$ for all $t$. Combining (3) and (4) gives the stated result. ∎

### B.1   Concentration analysis for non-stationary transition setting

Recall $\tilde{z}_t, \tilde{\sigma}_{V_{t+1}^{\text{in}}}$ (59) and $g_t$ (60) are three quantities deployed in Algorithm 1 that use off-policy data $\mathcal{D}$. We restate their definition as follows:

$$\tilde{z}_t(s_t, a_t) = \begin{cases} P_t^\top(\cdot|s_t, a_t) V_{t+1}^{\text{in}}, & if \ n_{s_t,a_t} < \frac{1}{2} m \cdot d_t^\mu(s_t, a_t), \\ \frac{1}{n_{s_t,a_t}} \sum_{i=1}^m V_{t+1}^{\text{in}}(s_{t+1}^{(i)}) \cdot \mathbf{1}_{[s_t^{(i)}=s_t, a_t^{(i)}=a_t]}, & if \ n_{s_t,a_t} \geq \frac{1}{2} m \cdot d_t^\mu(s_t, a_t). \end{cases}$$

$$\tilde{\sigma}_{V_{t+1}^{\text{in}}}(s_t, a_t) = \begin{cases} \sigma_{V_{t+1}^{\text{in}}}(s_t, a_t), & if \ n_{s_t,a_t} < \frac{1}{2} m \cdot d_t^\mu(s_t, a_t), \\ \frac{1}{n_{s_t,a_t}} \sum_{i=1}^m [V_{t+1}^{\text{in}}(s_{t+1}^{(i)})]^2 \cdot \mathbf{1}_{[s_t^{(i)}=s_t, a_t^{(i)}=a_t]} - \tilde{z}_t^2(s_t, a_t), & otherwise. \end{cases}$$

$$g_t(s_t, a_t) = \begin{cases} P^\top(\cdot|s_t, a_t)[V_{t+1} - V_{t+1}^{\text{in}}] - f(s_t, a_t), & if \ n_{s_t,a_t} < \frac{1}{2} l \cdot d_t^\mu(s_t, a_t), \\ \frac{1}{n'_{s_t,a_t}} \sum_{j=1}^l [V_{t+1}(s_{t+1}'^{(j)}) - V_{t+1}^{\text{in}}(s_{t+1}'^{(j)})] \cdot \mathbf{1}_{[s_t'^{(j)}, a_t'^{(j)}=s_t, a_t]} - f(s_t, a_t), & o.w. \end{cases}$$

and recall $f(s_t, a_t) = 4 u^{\text{in}} \sqrt{\log(2HSA/\delta)/l d_t^\mu(s_t, a_t)}$. Also, we use bold letters to represent matrices, *e.g.* $\mathbf{P}_t \in \mathbb{R}^{SA \times S}$ satisfies $\mathbf{P}_t[(s_t, a_t), s_{t+1}] = P_t(s_{t+1}|s_t, a_t)$. The following Lemmas B.2,B.4,B.5 provide their concentration properties.

**Lemma B.2.** *Let $\tilde{z}_t$ be defined as (59) in Algorithm 1, where $\tilde{z}_t$ is the off-policy estimator of $P_t^\top(\cdot|s_t, a_t) V_{t+1}^{\text{in}}$ using $m$ episodic data. Then with probability $1 - \delta$, we have*

$$\left| \tilde{z}_t - \mathbf{P}_t V_{t+1}^{\text{in}} \right| \leq \sqrt{\frac{4 \cdot \sigma_{V_{t+1}^{\text{in}}} \cdot \log(HSA/\delta)}{m \cdot d_t^\mu}} + \frac{4 V_{\text{max}}}{3m \cdot d_t^\mu} \log(HSA/\delta), \quad \forall t \in [H] \tag{5}$$

*here $\tilde{z}_t, \mathbf{P}_t V_{t+1}^{\text{in}}, \sigma_{V_{t+1}^{\text{in}}}, d_t^\mu \in \mathbb{R}^{S \times A}$ are $S \times A$ column vectors and $\sqrt{\cdot}$ is elementwise operation.*

*Proof.* First fix $s_t, a_t$. Let $E_t := \{n_{s_t,a_t} \geq \frac{1}{2} m \cdot d_t^\mu(s_t, a_t)\}$, then by definition,

$$\tilde{z}_t(s_t, a_t) - P_t^\top(\cdot|s_t, a_t) V_{t+1}^{\text{in}} = \left( \frac{1}{n_{s_t,a_t}} \sum_{i=1}^m V_{t+1}^{\text{in}}(s_{t+1}^{(i)}) \cdot \mathbf{1}[s_t^{(i)} = s_t, a_t^{(i)} = a_t] - P_t^\top(\cdot|s_t, a_t) V_{t+1}^{\text{in}} \right) \cdot \mathbf{1}(E_t).$$

Next we conditional on $n_{s_t,a_t}$. Then from above expression and Bernstein inequality G.3 we have with probability at least $1 - \delta$

$$\left|\tilde{z}_t(s_t, a_t) - P_t^\top(\cdot|s_t, a_t)V_{t+1}^{\text{in}}\right|$$

$$= \left|\frac{1}{n_{s_t,a_t}} \sum_{i=1}^{n_{s_t,a_t}} V_{t+1}^{\text{in}}(s_{t+1}^{(i)}|s_t, a_t) - P_t^\top(\cdot|s_t, a_t)V_{t+1}^{\text{in}}\right| \cdot \mathbf{1}(E_t)$$

$$\leq \left(\sqrt{\frac{2 \cdot \sigma_{V_{t+1}^{\text{in}}}(s_t, a_t) \cdot \log(1/\delta)}{n_{s_t,a_t}}} + \frac{2V_{\max}}{3n_{s_t,a_t}}\log(1/\delta)\right) \cdot \mathbf{1}(E_t)$$

$$\leq \sqrt{\frac{4 \cdot \sigma_{V_{t+1}^{\text{in}}}(s_t, a_t) \cdot \log(1/\delta)}{m \cdot d_t^\mu(s_t, a_t)}} + \frac{4V_{\max}}{3m \cdot d_t^\mu(s_t, a_t)}\log(1/\delta)$$

where we use shorthand notation $V_{t+1}^{\text{in}}(s_{t+1}^{(i)}|s_t, a_t)$ to denote the value of $V_{t+1}^{\text{in}}(s_{t+1}^{(i)})$ given $s_t^{(i)} = s_t$ and $a^{(i)} = a_t$. The condition $V_t^{\text{in}} \leq V_{\max}$ is guaranteed by Lemma B.1. Now we get rid of the conditional on $n_{s_t,a_t}$. Denote $A = \{\tilde{z}_t(s_t, a_t) - P^\top(\cdot|s_t, a_t)V_{t+1}^{\text{in}} \leq \sqrt{4 \cdot \sigma_{V_{t+1}^{\text{in}}}(s_t, a_t) \cdot \log(1/\delta)/m \cdot d_t^\mu(s_t, a_t)} + \frac{4V_{\max}}{3m \cdot d_t^\mu(s_t,a_t)}\log(1/\delta)\}$, then equivalently we can rewrite above result as $\mathbb{P}(A|n_{s_t,a_t}) \geq 1 - \delta$. Note this is the same as $\mathbb{E}[\mathbf{1}(A)|n_{s_t,a_t}] \geq 1 - \delta$, therefore by law of total expectation we have

$$\mathbb{P}(A) = \mathbb{E}[\mathbf{1}(A)] = \mathbb{E}[\mathbb{E}[\mathbf{1}(A)|n_{s_t,a_t}]] \geq \mathbb{E}[1 - \delta] = 1 - \delta,$$

*i.e.* for fixed $(s_t, a_t)$ we have with probability at least $1 - \delta$,

$$\left|\tilde{z}_t(s_t, a_t) - P^\top(\cdot|s_t, a_t)V_{t+1}^{\text{in}}\right| \leq \sqrt{\frac{4 \cdot \sigma_{V_{t+1}^{\text{in}}}(s_t, a_t) \cdot \log(1/\delta)}{m \cdot d_t^\mu(s_t, a_t)}} + \frac{4V_{\max}}{3m \cdot d_t^\mu(s_t, a_t)}\log(1/\delta)$$

Apply the union bound over all $t, s_t, a_t$, we obtain

$$\left|\tilde{z}_t - \boldsymbol{P}_t V_{t+1}^{\text{in}}\right| \leq \sqrt{\frac{4 \cdot \sigma_{V_{t+1}^{\text{in}}} \cdot \log(HSA/\delta)}{m \cdot d_t^\mu}} + \frac{4V_{\max}}{3m \cdot d_t^\mu}\log(HSA/\delta),$$

where the inequality is element-wise and this is (5). ∎

**Remark B.3.** *Exquisite reader might notice under the Assumption 2.1 it is likely for some $(s_t, a_t)$ the corresponding $d_t^\mu(s_t, a_t) = 0$, then the result (5) may fail to be meaningful (since less than infinity is trivial). However, in fact for those entries it is legitimate to set the right hand side of (5) equal to 0. The reason comes from our construction in (59) that when $d_t^\mu(s_t, a_t) = 0$, it must holds $n_{s_t,a_t} = 0$, so in this case $\tilde{z}_t(s_t, a_t) = P^\top(\cdot|s_t, a_t)V_{t+1}^{\text{in}}$. Therefore, we keep writing in this fashion only for the ease of illustration.*

**Lemma B.4.** *Let $\tilde{\sigma}_{V_{t+1}^{\text{in}}}$ be defined as (59) in Algorithm 1, i.e. the off-policy estimator of $\sigma_{V_{t+1}^{\text{in}}}(s_t, a_t)$ using $m$ episodic data. Then with probability $1 - \delta$, we have*

$$\left|\tilde{\sigma}_{V_{t+1}^{\text{in}}} - \sigma_{V_{t+1}^{\text{in}}}\right| \leq 6V_{\max}^2\sqrt{\frac{\log(4HSA/\delta)}{m \cdot d_t^\mu}} + \frac{4V_{\max}^2 \log(4HSA/\delta)}{m \cdot d_t^\mu}, \quad \forall t = 1, ..., H. \quad (6)$$

*Proof.* From the definition we have for fixed $(s_t, a_t)$

$$\tilde{\sigma}_{V_{t+1}^{\text{in}}}(s_t, a_t) - \sigma_{V_{t+1}^{\text{in}}}(s_t, a_t) = \left(\frac{1}{n_{s_t,a_t}} \sum_{i=1}^{n_{s_t,a_t}} V_{t+1}^{\text{in}}(s_{t+1}^{(i)}|s_t, a_t)^2 - P^\top(\cdot|s_t, a_t)(V_{t+1}^{\text{in}})^2\right)\mathbf{1}(E_t)$$

$$+ \left(\left[\frac{1}{n_{s_t,a_t}} \sum_{i=1}^{n_{s_t,a_t}} V_{t+1}^{\text{in}}(s_{t+1}^{(i)}|s_t, a_t)\right]^2 - \left[P^\top(\cdot|s_t, a_t)V_{t+1}^{\text{in}}\right]^2\right)\mathbf{1}(E_t)$$

By using the same conditional on $n_{s_t,a_t}$ as in Lemma B.2, applying Hoeffding's inequality and law of total expectation, we obtain with probability $1 - \delta/2$, the first term in above is bounded by

$$\left( \frac{1}{n_{s_t,a_t}} \sum_{i=1}^{n_{s_t,a_t}} V_{t+1}^{\text{in}}(s_{t+1}^{(i)}|s_t,a_t)^2 - P^\top(\cdot|s_t,a_t)(V_{t+1}^{\text{in}})^2 \right) \mathbf{1}(E_t)$$

$$\leq V_{\max}^2 \sqrt{\frac{2\log(4/\delta)}{n_{s_t,a_t}}} \cdot \mathbf{1}(E_t) \leq 2V_{\max}^2 \sqrt{\frac{\log(4/\delta)}{m \cdot d_t^\mu(s_t,a_t)}}, \tag{7}$$

and similarly with probability $1 - \delta/2$,

$$\left( \frac{1}{n_{s_t,a_t}} \sum_{i=1}^{n_{s_t,a_t}} V_{t+1}^{\text{in}}(s_{t+1}^{(i)}|s_t,a_t) - P^\top(\cdot|s_t,a_t)V_{t+1}^{\text{in}} \right) \mathbf{1}(E_t) \leq 2V_{\max} \sqrt{\frac{\log(4/\delta)}{m \cdot d_t^\mu(s_t,a_t)}}. \tag{8}$$

Note for $a,b,c > 0$, if $|a - b| \leq c$, then $|a^2 - b^2| = |a - b| \cdot |a + b| \leq |a - b| \cdot (|a| + |b|) \leq |a - b| \cdot (2|b| + c) \leq c \cdot (2|b| + c) = 2bc + c^2$, therefore by (8) we have

$$\left( \left[ \frac{1}{n_{s_t,a_t}} \sum_{i=1}^{n_{s_t,a_t}} V_{t+1}^{\text{in}}(s_{t+1}^{(i)}|s_t,a_t) \right]^2 - \left[ P^\top(\cdot|s_t,a_t)V_{t+1}^{\text{in}} \right]^2 \right) \mathbf{1}(E_t)$$

$$\leq 4P^\top(\cdot|s_t,a_t)V_{t+1}^{\text{in}} \cdot V_{\max} \sqrt{\frac{\log(4/\delta)}{m \cdot d_t^\mu(s_t,a_t)}} + \frac{4V_{\max}^2 \log(4/\delta)}{m \cdot d_t^\mu(s_t,a_t)} \tag{9}$$

$$\leq 4V_{\max}^2 \sqrt{\frac{\log(4/\delta)}{m \cdot d_t^\mu(s_t,a_t)}} + \frac{4V_{\max}^2 \log(4/\delta)}{m \cdot d_t^\mu(s_t,a_t)}$$

where the last inequality comes from $|P^\top(\cdot|s_t,a_t)V_{t+1}^{\text{in}}| \leq ||P(\cdot|s_t,a_t)||_1 ||V_{t+1}^{\text{in}}||_\infty \leq V_{\max}$. Combining (7), (9) and a union bound, we have with probability $1 - \delta$,

$$\left| \tilde{\sigma}_{V_{t+1}^{\text{in}}}(s_t,a_t) - \sigma_{V_{t+1}^{\text{in}}}(s_t,a_t) \right| \leq 6V_{\max}^2 \sqrt{\frac{\log(4/\delta)}{m \cdot d_t^\mu(s_t,a_t)}} + \frac{4V_{\max}^2 \log(4/\delta)}{m \cdot d_t^\mu(s_t,a_t)},$$

apply again the union bound over $t, s_t, a_t$ gives the desired result. ∎

**Lemma B.5.** *Fix time $t \in [H]$. Let $g_t$ be the estimator in (60) in Algorithm 1. Then if $||V_{t+1} - V_{t+1}^{\text{in}}||_\infty \leq 2u^{\text{in}}$, then with probability $1 - \delta/H$,*

$$\mathbf{0} \leq \boldsymbol{P}_t[V_{t+1} - V_{t+1}^{\text{in}}] - g_t \leq 8u^{\text{in}} \sqrt{\frac{\log(2HSA/\delta)}{ld_t^\mu}}$$

*Proof.* Recall $g_t, d_t^\mu$ are vectors. By definition of $g_t(s_t,a_t)$, applying Hoeffding's inequality we obtain with probability $1 - \delta/H$

$$g_t(s_t,a_t) + f(s_t,a_t) - P^\top(\cdot|s_t,a_t)[V_{t+1} - V_{t+1}^{\text{in}}]$$

$$= \left( \frac{1}{n'_{s_t,a_t}} \sum_{j=1}^{l} \left[ V_{t+1}(s_{t+1}'^{(j)}|s_t,a_t) - V_{t+1}^{\text{in}}(s_{t+1}'^{(j)}|s_t,a_t) \right] - P^\top(\cdot|s_t,a_t)[V_{t+1} - V_{t+1}^{\text{in}}] \right) \cdot \mathbf{1}(E_t)$$

$$\leq \left( ||V_{t+1} - V_{t+1}^{\text{in}}||_\infty \sqrt{\frac{2\log(2H/\delta)}{n'_{s_t,a_t}}} \right) \cdot \mathbf{1}(E_t)$$

$$\leq ||V_{t+1} - V_{t+1}^{\text{in}}||_\infty \sqrt{\frac{4\log(2H/\delta)}{l \cdot d_t^\mu(s_t,a_t)}}$$

Now use assumption $||V_{t+1} - V_{t+1}^{\text{in}}||_\infty \leq 2u^{\text{in}}$ and a union bound over $s_t, a_t$, we have with probability $1 - \delta/H$,

$$\left| g_t + f - \boldsymbol{P}[V_{t+1} - V_{t+1}^{\text{in}}] \right| \leq 4u^{\text{in}}\sqrt{\frac{\log(2HSA/\delta)}{ld_t^\mu}} \tag{10}$$

use $f = 4u^{\text{in}}\sqrt{\log(2HSA/\delta)/ld_t^\mu}$, we obtain the stated result. $\blacksquare$

**Remark B.6.** *The marginal state-action distribution $d_t^\mu$ entails the hardness in off-policy setting. If the current logging policy $\mu$ satisfies there exists some $s_t, a_t$ such that $d_t^\mu(s_t, a_t)$ is very small, then learning the MDP using this off-policy data will be generically hard, unless $d_t^{\pi^*}(s_t, a_t)$ is also relatively small for this $s_t, a_t$, see analysis in the following sections.*

## B.2 Iterative update analysis

The goal of iterative update is to obtain the recursive relation: $Q_t^\star - Q_t \leq \boldsymbol{P}_t^{\pi^*}[Q_{t+1}^\star - Q_{t+1}] + \xi_t$, where $\boldsymbol{P}_t^{\pi^*} \in \mathbb{R}^{SA \times SA}$ is a matrix. We control the error propagation term $\xi_t$ to be small enough.

**Lemma B.7.** *Let $Q^\star$ be the optimal Q-value satisfying $Q_t^\star = r + \boldsymbol{P}_t V_{t+1}^\star$ and $\pi^\star$ is one optimal policy satisfying Assumption 2.1. Let $\pi$ and $V_t$ be the **Return** of inner loop in Algorithm 1, and recall $V_{H+1} = \mathbf{0} \in \mathbb{R}^S$, $Q_{H+1} = \mathbf{0} \in \mathbb{R}^{S \times A}$. We have with probability $1 - \delta$, for all $t \in [H]$,*

$$V_t^{in} \leq V_t \leq \mathcal{T}_{\pi_t} V_{t+1} \leq V_t^\star, \quad Q_t \leq r + \boldsymbol{P}_t V_{t+1}, \quad and \quad Q_t^\star - Q_t \leq \boldsymbol{P}_t^{\pi^*}[Q_{t+1}^\star - Q_{t+1}] + \xi_t,$$

*where*

$$\xi_t \leq 8u^{in}\sqrt{\frac{\log(2HSA/\delta)}{ld_t^\mu}} + \sqrt{\frac{16 \cdot \sigma_{V_{t+1}^\star} \cdot \log(4HSA/\delta)}{m \cdot d_t^\mu}} + \sqrt{\frac{16 \cdot \log(4HSA/\delta)}{m \cdot d_t^\mu}} \cdot u^{in}$$

$$+ V_{\max}\left[ 8\sqrt{6} \cdot \left( \frac{\log(16HSA/\delta)}{m \cdot d_t^\mu} \right)^{3/4} + \frac{56\log(16HSA/\delta)}{3m \cdot d_t^\mu} \right].$$

*Here $\boldsymbol{P}^{\pi^*} \in \mathbb{R}^{S \cdot A \times S \cdot A}$ with $\boldsymbol{P}_{(s_t, a_t),(s_{t+1}, a_{t+1})}^{\pi^*} = d^{\pi^*}(s_{t+1}, a_{t+1}|s_t, a_t)$.*

*Proof.* **Step1:** For any $a, b \geq 0$, we have the basic inequality $\sqrt{a + b} \leq \sqrt{a} + \sqrt{b}$, and apply to Lemma B.4 we have with probability $1 - \delta/4$,

$$\sqrt{\left| \tilde{\sigma}_{V_{t+1}^{\text{in}}} - \sigma_{V_{t+1}^{\text{in}}} \right|} \leq V_{\max} \cdot \left( \frac{36\log(16HSA/\delta)}{m \cdot d_t^\mu} \right)^{1/4} + 2V_{\max} \cdot \sqrt{\frac{\log(16HSA/\delta)}{m \cdot d_t^\mu}}, \quad \forall t = 1, ..., H. \tag{11}$$

Next, similarly for any $a, b \geq 0$, we have $\sqrt{a} \leq \sqrt{|a - b|} + \sqrt{b}$, conditional on above then apply to Lemma B.2 (with probability $1 - \delta/4$) and we obtain with probability $1 - \delta/2$,

$$\left| \tilde{z}_t - \boldsymbol{P}_t V_{t+1}^{\text{in}} \right|$$

$$\leq \sqrt{\frac{4 \cdot \sigma_{V_{t+1}^{\text{in}}} \cdot \log(4HSA/\delta)}{m \cdot d_t^\mu}} + \frac{4V_{\max}}{3m \cdot d_t^\mu}\log(4HSA/\delta)$$

$$\leq \left( \sqrt{\tilde{\sigma}_{V_{t+1}^{\text{in}}}} + \sqrt{\left| \tilde{\sigma}_{V_{t+1}^{\text{in}}} - \sigma_{V_{t+1}^{\text{in}}} \right|} \right) \sqrt{\frac{4 \cdot \log(4HSA/\delta)}{m \cdot d_t^\mu}} + \frac{4V_{\max}}{3m \cdot d_t^\mu}\log(4HSA/\delta)$$

$$= \sqrt{\frac{4 \cdot \tilde{\sigma}_{V_{t+1}^{\text{in}}} \cdot \log(4HSA/\delta)}{m \cdot d_t^\mu}} + \left( \sqrt{\left| \tilde{\sigma}_{V_{t+1}^{\text{in}}} - \sigma_{V_{t+1}^{\text{in}}} \right|} \right) \sqrt{\frac{4 \cdot \log(4HSA/\delta)}{m \cdot d_t^\mu}} + \frac{4V_{\max}}{3m \cdot d_t^\mu}\log(4HSA/\delta)$$

$$\leq \sqrt{\frac{4 \cdot \tilde{\sigma}_{V_{t+1}^{\text{in}}} \cdot \log(4HSA/\delta)}{m \cdot d_t^\mu}} + 2\sqrt{6} \cdot V_{\max} \cdot \left( \frac{\log(16HSA/\delta)}{m \cdot d_t^\mu} \right)^{3/4} + \frac{16V_{\max}}{3m \cdot d_t^\mu}\log(16HSA/\delta).$$

Since $e = \sqrt{4 \cdot \tilde{\sigma}_{V_{t+1}^{\text{in}}} \cdot \log(4HSA/\delta)/(m \cdot d_t^\mu)} + 2\sqrt{6} \cdot V_{\max} \cdot (\log(16HSA/\delta)/(m \cdot d_t^\mu))^{3/4} + 16V_{\max}\log(16HSA/\delta)/(3m \cdot d_t^\mu)$, from above we have

$$z_t = \tilde{z}_t - e \leq \boldsymbol{P}_t V_{t+1}^{\text{in}}, \tag{12}$$

and

$$z_t \geq \boldsymbol{P}_t V_{t+1}^{\text{in}} - 2e. \tag{13}$$

Next note $\sqrt{\sigma_{(\cdot)}}$ is a norm, so by norm triangle inequality (for the second inequality) and $\sqrt{a} \leq \sqrt{b} + \sqrt{|b-a|}$ with (11) (for the first inequality) we have

$$\sqrt{\tilde{\sigma}_{V_{t+1}^{\text{in}}}} \leq \sqrt{\sigma_{V_{t+1}^{\text{in}}}} + V_{\max} \left[ \left( \frac{36 \log(16HSA/\delta)}{m \cdot d_t^\mu} \right)^{1/4} + \sqrt{\frac{4 \log(16HSA/\delta)}{m \cdot d_t^\mu}} \right]$$

$$\leq \sqrt{\sigma_{V_{t+1}^\star}} + \sqrt{\sigma_{V_{t+1}^\star - V_{t+1}^{\text{in}}}} + V_{\max} \left[ \left( \frac{36 \log(16HSA/\delta)}{m \cdot d_t^\mu} \right)^{1/4} + \sqrt{\frac{4 \log(16HSA/\delta)}{m \cdot d_t^\mu}} \right]$$

$$\leq \sqrt{\sigma_{V_{t+1}^\star}} + \sqrt{\boldsymbol{P}_t (V_{t+1}^\star - V_{t+1}^{\text{in}})^2} + V_{\max} \left[ \left( \frac{36 \log(16HSA/\delta)}{m \cdot d_t^\mu} \right)^{1/4} + \sqrt{\frac{4 \log(16HSA/\delta)}{m \cdot d_t^\mu}} \right]$$

$$\leq \sqrt{\sigma_{V_{t+1}^\star}} + ||V_{t+1}^\star - V_{t+1}^{\text{in}}||_\infty \cdot \mathbf{1} + V_{\max} \left[ \left( \frac{36 \log(16HSA/\delta)}{m \cdot d_t^\mu} \right)^{1/4} + \sqrt{\frac{4 \log(16HSA/\delta)}{m \cdot d_t^\mu}} \right]$$

$$\leq \sqrt{\sigma_{V_{t+1}^\star}} + u^{\text{in}} \cdot \mathbf{1} + V_{\max} \left[ \left( \frac{36 \log(16HSA/\delta)}{m \cdot d_t^\mu} \right)^{1/4} + \sqrt{\frac{4 \log(16HSA/\delta)}{m \cdot d_t^\mu}} \right]$$

Plug this back to (13) we get

$$\begin{aligned} z_t \geq & \boldsymbol{P}_t V_{t+1}^{\text{in}} - \sqrt{\frac{16 \cdot \sigma_{V_{t+1}^\star} \cdot \log(4HSA/\delta)}{m \cdot d_t^\mu}} - \sqrt{\frac{16 \cdot \log(4HSA/\delta)}{m \cdot d_t^\mu}} \cdot u^{\text{in}} \\ & - V_{\max} \left[ 8\sqrt{6} \cdot \left( \frac{\log(16HSA/\delta)}{m \cdot d_t^\mu} \right)^{3/4} + \frac{56 \log(16HSA/\delta)}{3m \cdot d_t^\mu} \right]. \end{aligned} \tag{14}$$

To sum up, so far we have shown that (12), (14) hold with probability $1 - \delta/2$ and we condition on that.

**Step2:** Next we prove

$$Q_t \leq r + \boldsymbol{P}_t V_{t+1}, \quad V_t^{\text{in}} \leq V_t \leq V_t^\star, \quad \forall t \in [H] \tag{15}$$

using backward induction.

First of all, $V_{H+1}^\star = V_{H+1} = V_{H+1}^{\text{in}} = 0$ implies $V_{H+1}^\star \leq V_{H+1} \leq V_{H+1}^{\text{in}}$ and

$$Q_H = r + z_H + g_H = r + (\mathbf{0} - e) + (\mathbf{0} - f) \leq r = r + \boldsymbol{P}_H^\top \mathbf{0} = r + \boldsymbol{P}_H^\top V_{H+1},$$

so the results hold for the base case.

Now for certain $t$, using induction assumption we can assume with probability at least $1 - (H - t - 1)\delta/H$, for all $t' = t + 1, ..., H$,

$$V_{t'}^{\text{in}} \leq V_{t'} \leq V_{t'}^\star, \quad Q_{t'} \leq r + \boldsymbol{P}_{t'} V_{t'+1}. \tag{16}$$

In particular, since $V_{t+1}^{\text{in}} \leq V_{t+1}^\star \leq V_{t+1}^{\text{in}} + u^{\text{in}}\mathbf{1}$, so combine this and (16) for $t' = t + 1$ we get

$$V_{t+1}^\star - V_{t+1} \leq V_{t+1}^\star - V_{t+1}^{\text{in}} \leq u^{\text{in}}\mathbf{1}.$$

By Lemma B.5, with probability $1 - \delta/H$,

$$\boldsymbol{P}_t [V_{t+1} - V_{t+1}^{\text{in}}] - 8u^{\text{in}} \sqrt{\frac{\log(2HSA/\delta)}{l d_t^\mu}} \leq g_t \leq \boldsymbol{P}_t [V_{t+1} - V_{t+1}^{\text{in}}]. \tag{17}$$

By the right hand side of above and (12) we acquire with probability $1 - (H - t)\delta/H$,

$$Q_t = r + z_t + g_t \leq r + \boldsymbol{P}_t V_{t+1}^{\text{in}} + \boldsymbol{P}_t [V_{t+1} - V_{t+1}^{\text{in}}] = r + \boldsymbol{P}_t V_{t+1} \leq r + \boldsymbol{P}_t V_{t+1}^\star = Q_t^\star$$

where the second equality already gives the proof of the first part of claim (15) and the second inequality is by induction assumption. Moreover, above $Q_t \leq Q_t^\star$ also implies $V_{Q_t} \leq V_{Q_t^\star} = V_t^\star$, so together with Lemma B.1 (note $V_t^{\text{in}} \leq \mathcal{T}_{\pi_t^{\text{in}}} V_{t+1}^{\text{in}}$) we have

$$V_t = \max(V_{Q_t}, V_t^{\text{in}}) \leq V_t^\star,$$

this completes the proof of the second part of claim (15).

**Step3:** Next we prove $V_t \leq \mathcal{T}_{\pi_t} V_{t+1}$.

For a particular $s_t$, on one hand, if $\pi_t(s_t) = \text{argmax}_{a_t} Q_t(s_t, a_t)$, by $Q_t \leq r + \boldsymbol{P}_t V_{t+1}$ we have in this case:

$$V_t(s_t) = \max_{a_t} Q_t(s_t, a_t) = Q_t(s_t, \pi_t(s_t)) \leq r(s_t, \pi_t(s_t)) + P^\top(\cdot|s_t, \pi_t(s_t)) V_{t+1} = (\mathcal{T}_{\pi_t} V_{t+1})(s_t),$$

where the first equal sign comes from the definition of $V_t$ when $V_{Q_t}(s_t) \geq V_t^{\text{in}}(s_t)$ and the first inequality is from Step2.

On the other hand, if $\pi_t(s_t) = \pi^{\text{in}}(s_t)$, then

$$V_t(s_t) = V_t^{\text{in}}(s_t) \leq (\mathcal{T}_{\pi_t^{\text{in}}} V_{t+1}^{\text{in}})(s_t) \leq (\mathcal{T}_{\pi_t^{\text{in}}} V_{t+1})(s_t) = (\mathcal{T}_{\pi_t} V_{t+1})(s_t)$$

where the first inequality is the property of input $V^{\text{in}}$, $\pi^{\text{in}}$ and the second inequality is from Step2.

**Step4:** It remains to prove $Q_t^\star - Q_t \leq \boldsymbol{P}_t^{\pi^\star}[Q_{t+1}^\star - Q_{t+1}] + \xi_t$. Indeed, using the construction of $Q_t$, we have

$$\begin{aligned}
Q_t^\star - Q_t &= Q_t^\star - r - z_t - g_t = \boldsymbol{P}_t V_{t+1}^\star - z_t - g_t \\
&= \boldsymbol{P}_t V_{t+1}^\star - \boldsymbol{P}_t(V_{t+1} - V_{t+1}^{\text{in}}) - \boldsymbol{P}_t V_{t+1}^{\text{in}} + \xi_t = \boldsymbol{P}_t V_{t+1}^\star - \boldsymbol{P}_t V_{t+1} + \xi_t,
\end{aligned} \tag{18}$$

where the second equation uses Bellman optimality equation and the third equation uses the definition of $\xi_t = \boldsymbol{P}_t(V_{t+1} - V_{t+1}^{\text{in}}) - g_t + \boldsymbol{P}_t V_{t+1}^{\text{in}} - z_t$. By (14) and (17),

$$\begin{aligned}
\xi_t \leq &8u^{\text{in}} \sqrt{\frac{\log(2HSA/\delta)}{l d_t^\mu}} + \sqrt{\frac{16 \cdot \sigma_{V_{t+1}^\star} \cdot \log(4HSA/\delta)}{m \cdot d_t^\mu}} + \sqrt{\frac{16 \cdot \log(4HSA/\delta)}{m \cdot d_t^\mu}} \cdot u^{\text{in}} \\
&+ V_{\max} \left[ 8\sqrt{6} \cdot \left( \frac{\log(16HSA/\delta)}{m \cdot d_t^\mu} \right)^{3/4} + \frac{56 \log(16HSA/\delta)}{3m \cdot d_t^\mu} \right].
\end{aligned}$$

Lastly, note $\boldsymbol{P}_t V_{t+1}^\star = \boldsymbol{P}_t^{\pi^\star} Q_{t+1}^\star$ and by definition $V_{t+1} \geq V_{Q_{t+1}}$, so we have $\boldsymbol{P}_t V_{t+1} \geq \boldsymbol{P}_t V_{Q_{t+1}} = \boldsymbol{P}_t^{\pi_{Q_{t+1}}} Q_{t+1} \geq \boldsymbol{P}_t^{\pi^\star} Q_{t+1}$, the last inequality holds true since $\pi_{Q_{t+1}}$ is the greedy policy over $Q_{t+1}$. Threfore (18) becomes $Q_t^\star - Q_t = \boldsymbol{P}_t V_{t+1}^\star - \boldsymbol{P}_t V_{t+1} + \xi_t \leq \boldsymbol{P}_t^{\pi^\star} Q_{t+1}^\star - \boldsymbol{P}_t^{\pi^\star} Q_{t+1} + \xi_t$. This completes the proof. ∎

**Lemma B.8.** *Suppose the input $V_t^{in}$, $t \in [H]$ of Algorithm 1 satisfies $V_t^{in} \leq \mathcal{T}_{\pi_t^{in}} V_{t+1}^{in}$ and $V_t^{in} \leq V_t^\star \leq V_t^{in} + u^{in} \boldsymbol{1}$. Let $V_t$, $\pi$ be the return of inner loop of Algorithm 1 and choose $m = l := m' \cdot \log(16HSA)/(u^{in})^2$, where $m'$ is a parameter will be decided later. Then in addition to the results of Lemma B.7, we have with probability $1 - \delta$,*

- *if $u^{in} \in [\sqrt{H}, H]$, then:*

$$\begin{aligned}
\boldsymbol{0} \leq V_t^\star - V_t \leq & \\
\leq \Bigg( \frac{12H^2}{\sqrt{m'}} &\left\| \boldsymbol{d}_{t:t'}^{\pi^\star} \sqrt{\frac{1}{d_{t'}^\mu}} \right\|_{\infty, H} + \frac{4}{\sqrt{m'}} \left\| \sum_{t'=t}^{H} \boldsymbol{d}_{t:t'}^{\pi^\star} \sqrt{\frac{\sigma_{V_{t'+1}^\star}}{d_{t'}^\mu}} \right\|_\infty + \frac{8\sqrt{6} H^{\frac{10}{4}}}{(m')^{3/4}} \left\| \boldsymbol{d}_{t:t'}^{\pi^\star} \sqrt{\frac{1}{d_{t'}^\mu}} \right\|_{\infty, H} \\
&+ \frac{56H^3}{3m'} \left\| \boldsymbol{d}_{t:t'}^{\pi^\star} \frac{1}{d_{t'}^\mu} \right\|_{\infty, H} \Bigg) u^{in} \cdot \boldsymbol{1}.
\end{aligned}$$

- *if $u^{in} \leq \sqrt{H}$, then*

$$0 \leq V_t^\star - V_t \leq$$

$$\leq \left( \frac{12\sqrt{H^3}}{\sqrt{m'}} \left\| \boldsymbol{d}_{t:t'}^{\pi^\star} \sqrt{\frac{1}{d_{t'}^\mu}} \right\|_{\infty,H} + \frac{4}{\sqrt{m'}} \left\| \sum_{t'=t}^{H} \boldsymbol{d}_{t:t'}^{\pi^\star} \sqrt{\frac{\sigma_{V_{t'+1}^\star}}{d_{t'}^\mu}} \right\|_\infty + \frac{8\sqrt{6}H^{\frac{9}{4}}}{(m')^{3/4}} \left\| \boldsymbol{d}_{t:t'}^{\pi^\star} \sqrt{\frac{1}{d_{t'}^\mu}} \right\|_{\infty,H} \right.$$

$$\left. + \frac{56H^{\frac{5}{2}}}{3m'} \left\| \boldsymbol{d}_{t:t'}^{\pi^\star} \frac{1}{d_{t'}^\mu} \right\|_{\infty,H} \right) u^{in} \cdot \mathbf{1}.$$

*where $\boldsymbol{d}_{t:t'}^{\pi^\star} \in \mathbb{R}^{S\cdot A \times S\cdot A}$ is a matrix represents the multi-step transition from time $t$ to $t'$, i.e.*
*$\boldsymbol{d}_{(s_t,a_t),(s_{t'},a_{t'})}^{\pi^\star} = d_{t:t'}^\star(s_{t'},a_{t'}|s_t,a_t)$ and recall $1/d_{t'}^\mu$ is a vector. $\boldsymbol{d}_{t:t'}^{\pi^\star} \frac{1}{d_{t'}^\mu}$ is a matrix-vector multi-*
*plication. For a vector $d_t \in \mathbb{R}^{S \times A}$, norm $\|\cdot\|_{\infty,H}$ is defined as $\|d_t\|_{\infty,H} = \max_{t,s_t,a_t} d_t(s_t,a_t)$.*

**Remark B.9.** *Note if $u^{in} \geq \sqrt{H}$, the first term in $V_t - V_t^\star$ requires sample $m'$ of order $O(H^4)$, which is suboptimal. This is the main reason why we need the doubling procedure in Algorithm 2 to keep the whole algorithm optimal.*

*Proof.* By Lemma B.7, we have with probability $1 - \delta$, for all $t \in [H]$,

$$V_t^{in} \leq V_t \leq \mathcal{T}_{\pi_t} V_{t+1} \leq V_t^\star, \quad Q_t \leq r + \boldsymbol{P}_t V_{t+1}, \quad \text{and} \quad Q_t^\star - Q_t \leq \boldsymbol{P}_t^{\pi^\star}[Q_{t+1}^\star - Q_{t+1}] + \xi_t,$$

where

$$\xi_t \leq 8u^{in} \sqrt{\frac{\log(2HSA/\delta)}{l d_t^\mu}} + \sqrt{\frac{16 \cdot \sigma_{V_{t+1}^\star} \cdot \log(4HSA/\delta)}{m \cdot d_t^\mu}} + \sqrt{\frac{16 \cdot \log(4HSA/\delta)}{m \cdot d_t^\mu}} \cdot u^{in}$$

$$+ V_{\max} \left[ 8\sqrt{6} \cdot \left( \frac{\log(16HSA/\delta)}{m \cdot d_t^\mu} \right)^{3/4} + \frac{56\log(16HSA/\delta)}{3m \cdot d_t^\mu} \right].$$

Applying the recursion repeatedly, we obtain

$$Q_t^\star - Q_t \leq \sum_{t'=t}^{H} \left[ \prod_{i=t}^{t'-1} \boldsymbol{P}_i^{\pi^\star} \right] \xi_{t'}$$

Note $\prod_{i=t}^{t'-1} \boldsymbol{P}_i^{\pi^\star} \in \mathbb{R}^{S\cdot A \times S\cdot A}$ represents the multi-step transition from time $t$ to $t'$, *i.e.* $(\prod_{i=t}^{t'-1} \boldsymbol{P}_i^{\pi^\star})_{(s_t,a_t),(s_{t'},a_{t'})} = d_{t:t'}^{\pi^\star}(s_{t'},a_{t'}|s_t,a_t)$. Therefore

$$Q_t^\star - Q_t \leq \sum_{t'=t}^{H} \left[ \prod_{i=t}^{t'-1} \boldsymbol{P}_i^{\pi^\star} \right] \xi_{t'} = \sum_{t'=t}^{H} \boldsymbol{d}_{t:t'}^{\pi^\star} \xi_{t'}$$

$$\leq \sum_{t'=t}^{H} \boldsymbol{d}_{t:t'}^{\pi^\star} \left( 8u^{in} \sqrt{\frac{\log(2HSA/\delta)}{l d_{t'}^\mu}} + \sqrt{\frac{16 \cdot \sigma_{V_{t'+1}^\star} \cdot \log(4HSA/\delta)}{m \cdot d_{t'}^\mu}} + \sqrt{\frac{16 \cdot \log(4HSA/\delta)}{m \cdot d_{t'}^\mu}} \cdot u^{in} \right.$$

$$\left. + V_{\max} \left[ 8\sqrt{6} \cdot \left( \frac{\log(16HSA/\delta)}{m \cdot d_{t'}^\mu} \right)^{3/4} + \frac{56\log(16HSA/\delta)}{3m \cdot d_{t'}^\mu} \right] \right)$$

$$(19)$$

Now by our choice of $m := m' \cdot \log(16HSA/\delta)/(u^{in})^2$ and $l := m'/H \cdot \log(16HSA/\delta)$, then (19) further less than

$$\leq \sum_{t'=t}^{H} \boldsymbol{d}_{t:t'}^{\pi^\star} \left( \frac{8\sqrt{H} + 4u^{in}}{\sqrt{m' d_{t'}^\mu}} u^{in} + \sqrt{\frac{16 \cdot \sigma_{V_{t'+1}^\star}}{m' \cdot d_{t'}^\mu}} u^{in} + V_{\max} \left[ 8\sqrt{6} \cdot \left( \frac{(u^{in})^{2/3}}{m' \cdot d_{t'}^\mu} \right)^{3/4} + \frac{56 u^{in}}{3m' \cdot d_{t'}^\mu} \right] \cdot u^{in} \right)$$

$$(20)$$

**Case1.** If $u^{\text{in}} \leq \sqrt{H}$, then (20) is less than

$$\leq \sum_{t'=t}^{H} \boldsymbol{d}_{t:t'}^{\pi^\star} \left( \frac{12\sqrt{H}}{\sqrt{m'd_{t'}^{\mu}}} + \sqrt{\frac{16 \cdot \sigma_{V_{t'+1}^\star}}{m' \cdot d_{t'}^{\mu}}} + V_{\max}\left[8\sqrt{6} \cdot \left(\frac{H^{1/3}}{m' \cdot d_{t'}^{\mu}}\right)^{3/4} + \frac{56H^{1/2}}{3m' \cdot d_{t'}^{\mu}}\right]\right) u^{\text{in}}$$

$$\leq \left(\frac{12\sqrt{H^3}}{\sqrt{m'}} \left\|\boldsymbol{d}_{t:t'}^{\pi^\star}\sqrt{\frac{1}{d_{t'}^{\mu}}}\right\|_{\infty,H} + \frac{4}{\sqrt{m'}}\left\|\sum_{t'=t}^{H} \boldsymbol{d}_{t:t'}^{\pi^\star}\sqrt{\frac{\sigma_{V_{t'+1}^\star}}{d_{t'}^{\mu}}}\right\|_{\infty} + \frac{8\sqrt{6}H^{\frac{9}{4}}}{(m')^{3/4}}\left\|\boldsymbol{d}_{t:t'}^{\pi^\star}\left[\frac{1}{d_{t'}^{\mu}}\right]^{\frac{3}{4}}\right\|_{\infty,H}\right.$$

$$\left.+ \frac{56H^{\frac{5}{2}}}{3m'}\left\|\boldsymbol{d}_{t:t'}^{\pi^\star}\frac{1}{d_{t'}^{\mu}}\right\|_{\infty,H}\right) u^{\text{in}} \cdot \boldsymbol{1}.$$

$$(21)$$

**Case2.** If $u^{\text{in}} \geq \sqrt{H}$, then (20) is less than

$$\leq \sum_{t'=t}^{H} \boldsymbol{d}_{t:t'}^{\pi^\star} \left( \frac{12H}{\sqrt{m'd_{t'}^{\mu}}} + \sqrt{\frac{16 \cdot \sigma_{V_{t'+1}^\star}}{m' \cdot d_{t'}^{\mu}}} + V_{\max}\left[8\sqrt{6} \cdot \left(\frac{H^{2/3}}{m' \cdot d_{t'}^{\mu}}\right)^{3/4} + \frac{56H}{3m' \cdot d_{t'}^{\mu}}\right]\right) u^{\text{in}}$$

$$\leq \left(\frac{12H^2}{\sqrt{m'}} \left\|\boldsymbol{d}_{t:t'}^{\pi^\star}\sqrt{\frac{1}{d_{t'}^{\mu}}}\right\|_{\infty,H} + \frac{4}{\sqrt{m'}}\left\|\sum_{t'=t}^{H} \boldsymbol{d}_{t:t'}^{\pi^\star}\sqrt{\frac{\sigma_{V_{t'+1}^\star}}{d_{t'}^{\mu}}}\right\|_{\infty} + \frac{8\sqrt{6}H^{\frac{10}{4}}}{(m')^{3/4}}\left\|\boldsymbol{d}_{t:t'}^{\pi^\star}\left[\frac{1}{d_{t'}^{\mu}}\right]^{\frac{3}{4}}\right\|_{\infty,H}\right.$$

$$\left.+ \frac{56H^3}{3m'}\left\|\boldsymbol{d}_{t:t'}^{\pi^\star}\frac{1}{d_{t'}^{\mu}}\right\|_{\infty,H}\right) u^{\text{in}} \cdot \boldsymbol{1}.$$

$$(22)$$

$\blacksquare$

## B.3 The doubling procedure

Before we explain the doubling procedure, let us first finish the proof the Algorithm 1.

**Lemma B.10.** *For convenience, define:*

$$A_{\frac{1}{2}} = \left\|\boldsymbol{d}_{t:t'}^{\pi^\star}\sqrt{\frac{1}{d_{t'}^{\mu}}}\right\|_{\infty,H}, \quad A_2 = \left\|\sum_{t'=t}^{H} \boldsymbol{d}_{t:t'}^{\pi^\star}\sqrt{\frac{\sigma_{V_{t'+1}^\star}}{d_{t'}^{\mu}}}\right\|_{\infty}, \quad A_{\frac{3}{4}} = \left\|\boldsymbol{d}_{t:t'}^{\pi^\star}\left[\frac{1}{d_{t'}^{\mu}}\right]^{\frac{3}{4}}\right\|_{\infty,H}, \quad A_1 = \left\|\boldsymbol{d}_{t:t'}^{\pi^\star}\frac{1}{d_{t'}^{\mu}}\right\|_{\infty,H}.$$

*Recall $\epsilon$ is the target accuracy in the outer loop of Algorithm 1. Then:*

- *If $u^{(0)} \leq \sqrt{H}$, then choose $m^{(i)} = B\log(16HSAK/\delta)/(u^{(i-1)})^2$ and $l^{(i)} = B/H\log(16HSAK/\delta)$, where*

$$B = \max\left[96^2 H^3 A_{\frac{1}{2}}^2, 32^2 A_2^2, \left(64\sqrt{6}A_{\frac{3}{4}}\right)^{\frac{4}{3}} H^3, \frac{448}{3}H^{5/2}A_1\right], \quad K = \log_2(\sqrt{H}/\epsilon),$$

- *If $u^{(0)} > \sqrt{H}$, then choose $m^{(i)} = B\log(16HSAK/\delta)/(u^{(i-1)})^2$ and $l^{(i)} = B/H\log(16HSAK/\delta)$, where*

$$B = \max\left[96^2 H^4 A_{\frac{1}{2}}^2, 32^2 A_2^2, \left(64\sqrt{6}A_{\frac{3}{4}}\right)^{\frac{4}{3}} H^{\frac{10}{3}}, \frac{448}{3}H^3 A_1\right], \quad K = \log_2(H/\epsilon),$$

*Then Algorithm 1 guarantees with probability $1 - \delta$, the output $\pi^{(K)}$ is a $\epsilon$-optimal policy, i.e. $\|V_1^\star - V_1^{\pi^{(K)}}\|_\infty < \epsilon$ with total episode complexity:*

$$\frac{2B\log(16HSAK/\delta)}{\epsilon^2}K$$

*for both cases. Moreover, B can be simplified as:*

- *If $u^{(0)} \leq \sqrt{H}$, then $B \leq cH^3/d_m$;*

- If $u^{(0)} > \sqrt{H}$, then $B \leq cH^4/d_m$.

*Proof of Lemma B.10.* **Step1: proof in general.** First, using induction it is easy to show for all $0 < a_1, ..., a_n < 1$, it follows

$$(1 - a_1) \cdot (1 - a_2) \cdot ... \cdot (1 - a_n) \geq 1 - (a_1 + ... + a_n).$$

and this directly implies $(1 - \frac{\delta}{K})^K \geq 1 - \delta$. By the choice of $m'$ and $K$, for both situation by Lemma B.8 we always have $||V_t^\star - V_t^{\pi^{(i)}}||_\infty < u^{(i-1)}/2 = u^{(i)}$ with probability $1 - \delta/K$ (this is because we choose $m^{(i)} = l^{(i)} = B\log(16HSAK/\delta)/(u^{(i-1)})^2$ instead of $B\log(16HSA/\delta)/(u^{(i-1)})^2$).

Therefore by chain rule of probability,

$$\mathbb{P}\left(\forall i \in [K], t \in [H], \ V_t^\star - V_t^{\pi^{(i)}} \leq u^{(i)}\right)$$

$$= \prod_{j=2}^{K} \mathbb{P}\left(\forall t \in [H], \ V_t^\star - V_t^{\pi^{(j)}} \leq u^{(j)} \Big| \forall i \in [j-1], t \in [H], \ V_t^\star - V_t^{\pi^{(i)}} \leq u^{(i)}\right)$$

$$\times \mathbb{P}\left(\forall t \in [H], \ V_1^\star - V_1^{\pi^{(1)}} \leq u^{(1)}\right)$$

$$\geq (1 - \frac{\delta}{K})^K \geq 1 - \delta.$$

In particular, in both situation[6]

$$\forall t \in [H], \ V_t^\star - V_t^{\pi^{(K)}} \leq u^{(K)} = u^{(0)} \cdot 2^{-K} = \epsilon,$$

with total number of budget to be

$$\sum_{i=1}^{K}(m^{(i)} + Hl^{(i)}) = \sum_{i=1}^{K}\frac{B\log(16HSAK/\delta)}{(u^{(i-1)})^2} + \sum_{i=1}^{K}Hl^{(i)}$$

$$\leq \sum_{i=1}^{K}\frac{B\log(16HSAK/\delta)}{(u^{(0)} \cdot 2^{-K})^2} + \sum_{i=1}^{K}Hl^{(i)} = \frac{B\log(16HSAK/\delta)}{\epsilon^2}K + \sum_{i=1}^{K}Hl^{(i)} \leq 2\frac{B\log(16HSAK/\delta)}{\epsilon^2}K,$$

where the last step uses $\epsilon \leq 1$.

**Step2: simplified expression for $m'$.** Indeed,

$$\boldsymbol{d}_{t:t'}^{\pi^\star}\sqrt{\frac{1}{d_{t'}^\mu}} \leq \boldsymbol{d}_{t:t'}^{\pi^\star}\sqrt{\frac{1}{d_m} \cdot \boldsymbol{1}} \leq \sqrt{\frac{1}{d_m}}||\boldsymbol{d}_{t:t'}^{\pi^\star}||_1 \cdot ||\boldsymbol{1}||_\infty \cdot \boldsymbol{1} = \sqrt{\frac{1}{d_m}} \cdot \boldsymbol{1} \Rightarrow A_{\frac{1}{2}} \leq \sqrt{\frac{1}{d_m}};$$

$$\boldsymbol{d}_{t:t'}^{\pi^\star}\left[\frac{1}{d_{t'}^\mu}\right]^{\frac{3}{4}} \leq \boldsymbol{d}_{t:t'}^{\pi^\star}\left[\frac{1}{d_m}\right]^{\frac{3}{4}} \cdot \boldsymbol{1} \leq \left[\frac{1}{d_m}\right]^{\frac{3}{4}}||\boldsymbol{d}_{t:t'}^{\pi^\star}||_1 \cdot ||\boldsymbol{1}||_\infty \cdot \boldsymbol{1} = \left[\frac{1}{d_m}\right]^{\frac{3}{4}} \cdot \boldsymbol{1} \Rightarrow A_{\frac{3}{4}} \leq \left[\frac{1}{d_m}\right]^{\frac{3}{4}};$$

$$\boldsymbol{d}_{t:t'}^{\pi^\star}\frac{1}{d_{t'}^\mu} \leq \boldsymbol{d}_{t:t'}^{\pi^\star}\frac{1}{d_m} \cdot \boldsymbol{1} \leq \frac{1}{d_m}||\boldsymbol{d}_{t:t'}^{\pi^\star}||_1 \cdot ||\boldsymbol{1}||_\infty \cdot \boldsymbol{1} = \frac{1}{d_m} \cdot \boldsymbol{1} \Rightarrow A_1 \leq \frac{1}{d_m};$$

and

$$\sum_{t'=t}^{H} \sum_{s_{t'},a_{t'}} d_{t:t'}^{\pi^\star}(s_{t'}, a_{t'}|s_t, a_t)\sqrt{\frac{\sigma_{V_{t'+1}^\star}(s_{t'}, a_{t'})}{d_{t'}^\mu(s_{t'}, a_{t'})}}$$

$$= \sum_{t'=t}^{H} \sum_{s_{t'},a_{t'}} \sqrt{d_{t:t'}^{\pi^\star}(s_{t'}, a_{t'}|s_t, a_t)}\sqrt{\frac{\sigma_{V_{t'+1}^\star}(s_{t'}, a_{t'})d_{t:t'}^{\pi^\star}(s_{t'}, a_{t'}|s_t, a_t)}{d_{t'}^\mu(s_{t'}, a_{t'})}}$$

---

[6]The last equal sign holds since if $u^{(0)} \leq \sqrt{H}$ (or $u^{(0)} \leq H$), you can always reset $u^{(0)} = \sqrt{H}$ (or $u^{(0)} = H$).

$$\leq \sqrt{\frac{1}{d_m}} \sum_{t'=t}^{H} \sum_{s_{t'},a_{t'}} \sqrt{d_{t:t'}^{\pi^\star}(s_{t'},a_{t'}|s_t,a_t)} \sqrt{\sigma_{V_{t'+1}^\star}(s_{t'},a_{t'})d_{t:t'}^{\pi^\star}(s_{t'},a_{t'}|s_t,a_t)}$$

$$\underset{CS\ Ineq}{\leq} \sqrt{\frac{1}{d_m}} \sum_{t'=t}^{H} \sqrt{\sum_{s_{t'},a_{t'}} d_{t:t'}^{\pi^\star}(s_{t'},a_{t'}|s_t,a_t) \sum_{s_{t'},a_{t'}} \sigma_{V_{t'+1}^\star}(s_{t'},a_{t'})d_{t:t'}^{\pi^\star}(s_{t'},a_{t'}|s_t,a_t)}$$

$$= \sqrt{\frac{1}{d_m}} \sum_{t'=t}^{H} \sqrt{1 \cdot \mathbb{E}_{s_{t'},a_{t'}}^{\pi^\star} \left[ \sigma_{V_{t'+1}^\star}(s_{t'},a_{t'}) \Big| s_t,a_t \right]}$$

$$\underset{CS\ Ineq}{\leq} \sqrt{\frac{1}{d_m}} \sqrt{\sum_{t'=t}^{H} 1 \cdot \sum_{t'=t}^{H} \mathbb{E}_{s_{t'},a_{t'}}^{\pi^\star} \left[ \sigma_{V_{t'+1}^\star}(s_{t'},a_{t'}) \Big| s_t,a_t \right]}$$

$$\underset{lem\ G.5}{\leq} \sqrt{\frac{1}{d_m}} \sqrt{\sum_{t'=t}^{H} 1 \cdot \mathrm{Var}_{\pi^\star} \left[ \sum_{t'=t}^{H} r_{t'} \Big| s_t,a_t \right]} \leq \sqrt{\frac{H^3}{d_m}} \Rightarrow A_2 \leq \sqrt{\frac{H^3}{d_m}},$$

Plug all these numbers back, we have the simplified bound for $B$. ∎

**Remark B.11.** *The Assumption 2.1 comes into picture for the validity of the bound for $A_2$ since when $d_{t:t'}^{\pi^\star}(s_t',a_t'|s_t,a_t) > 0$, by Assumption 2.1 we always have $d_{t'}^{\mu}(s_t',a_t') > 0$ so the bound will never be the trivial $+\infty$.*

**Corollary B.12.** *Note choose any $m' > B$ (in Lemma B.10) yields the similar complexity bound of*

$$\frac{2m' \log(16HSAK/\delta)}{\epsilon^2} K,$$

*therefore by the simplified bound of B, we choose $m' = O(H^4/d_m)$ for stage1 and $m' = O(H^3/d_m)$ for stage2.*

**The doubling procedure.** As we can see in Lemma B.10, if the initial input $V_t^{(0)}$ in Algorithm 2 has $\sup_t ||V_t^{(0)} - V_t^\star||_\infty \geq \sqrt{H}$ (i.e. $u^{(0)} > \sqrt{H}$), then it requires total of $\tilde{O}(H^4/d_m\epsilon^2)$ episodes to obtain $\epsilon$ accuracy, which is suboptimal. The doubling procedure helps resolve the problem. Concretely, for any final accuracy $0 < \epsilon \leq 1$:

- **Stage1.** Denote $\epsilon' = \sqrt{H}\epsilon$ and $u^{(0)} = H$, then by the choice of $K$ and $m'_H = cH^4/d_m$ for the case of $u^{(0)} \geq \sqrt{H}$ in Lemma B.10, it outputs $V_t^{\text{intermediate}}$, $\pi^{\text{intermediate}}$ which is $\epsilon'$ optimal with complexity:

$$\frac{2m'_H \log(16HSAK_{\epsilon'}/\delta)}{\epsilon'^2} K_{\epsilon'}$$

  where $K_{\epsilon'} = \log_2(H/\epsilon')$;

- **Stage2.** Use $V_t^{\text{intermediate}}$, $\pi^{\text{intermediate}}$ as input, since $\epsilon' = \sqrt{H}\epsilon \leq \sqrt{H}$, we can set $u^{(0)} = \sqrt{H}$. Now by Lemma B.10 again (with $m'_{\sqrt{H}} = cH^3/d_m$), Algorithm 2 has the final output $V_t^{\text{final}}$, $\pi^{\text{final}}$ that is $\epsilon$ optimal with complexity

$$\frac{2m'_{\sqrt{H}} \log(16HSAK_\epsilon/\delta)}{\epsilon^2} K_\epsilon.$$

  where $K_\epsilon = \log_2(\sqrt{H}/\epsilon)$.

Plug back $\epsilon' = \sqrt{H}\epsilon$, Algorithm 2 guarantees $\epsilon$-optimal policy with probability $1 - \delta$ using total complexity

$$\frac{2m'_H \log(16HSAK_{\epsilon'}/\delta)}{\epsilon'^2} K_{\epsilon'} + \frac{2m'_{\sqrt{H}} \log(16HSAK_\epsilon/\delta)}{\epsilon^2} K_\epsilon$$

$$\leq \frac{4 \max[\frac{m'_H}{H}, m'_{\sqrt{H}}] \log(16HSA \log_2(\sqrt{H}/\epsilon)/\delta)}{\epsilon^2} \log_2(\sqrt{H}/\epsilon) \tag{23}$$

$$\leq O\left( \frac{H^3 \log(16HSA \log_2(\sqrt{H}/\epsilon)/\delta)}{d_m\epsilon^2} \log_2(\sqrt{H}/\epsilon) \right)$$

where the last inequality uses $m'_{\sqrt{H}} \leq cH^3/d_m$ and $m'_H \leq cH^4/d_m$ in Lemma B.10 and above holds with probability $1 - \delta$.

This provides the minimax optimality of $\tilde{O}(H^3/d_m\epsilon^2)$ for non-stationary setting.

## B.4    Practical OPDVR

To go from non-implementable version to the practical version, the idea is to bound the event $\{n_{s_t,a_t} \leq \frac{1}{2}m \cdot d_t^\mu(s_t, a_t)\}$ and $\{n'_{s_t,a_t} \leq \frac{1}{2}l \cdot d_t^\mu(s_t, a_t)\}$ so that with high probability, the non-implementable version is identical to the practical OPDVR in Algorithm 2. Specifically, when $m' \geq 8H^2/d_m$ (this is satisfied since for each stage we set $m'$ to be at least $O(H^3/d_m)$), then

$$d_m \min_i m^{(i)} = d_m \min_i \frac{m' \log(16KHSA/\delta)}{(u^{(i-1)})^2} \geq d_m \frac{m' \log(16KHSA/\delta)}{H^2} \geq 8 \log(16KHSA/\delta),$$

so by Lemma G.2 and a union bound

$$\mathbb{P}\left( \bigcup_{i \in [K]} \bigcup_{\{t,s_t,a_t \,:\, d_t^\mu(s_t,a_t) > 0\}} \{n_{s_t,a_t}^{(i)} \leq \frac{1}{2}m^{(i)} \cdot d_t^\mu(s_t, a_t)\} \cup \{n_{s_t,a_t}'^{(i)} \leq \frac{1}{2}l^{(i)} \cdot d_t^\mu(s_t, a_t)\} \right)$$

$$\leq 2\mathbb{P}\left( \bigcup_{i \in [K]} \bigcup_{\{t,s_t,a_t \,:\, d_t^\mu(s_t,a_t) > 0\}} \{n_{s_t,a_t}^{(i)} \leq \frac{1}{2}m^{(i)} \cdot d_t^\mu(s_t, a_t)\} \right)$$

$$\leq 2KHSA \cdot \max_{\{i,t,s_t,a_t \,:\, d_t^\mu(s_t,a_t) > 0\}} \mathbb{P}\left( n_{s_t,a_t}^{(i)} \leq \frac{1}{2}m^{(i)} \cdot d_t^\mu(s_t, a_t) \right)$$

$$\leq 2KHSA \cdot e^{-d_m \min_i m^{(i)}/8} \leq \frac{2KHSA}{16KHSA/\delta} = \delta/8 < \delta/4,$$

(24)

and repeat this analysis for both stages, we have with probability $1 - \delta/2$, Practical OPDVR is identical to the non-implementable version.

## B.5    Proof of Theorem 3.2

*Proof.* The proof consists of two parts. The first part is to use (23) to show OPDVR in Algorithm 2 outputs $\epsilon$-optimal policy using episode complexity

$$\frac{2\max[\frac{m'_H}{H}, m'_{\sqrt{H}}] \log(32HSA \log_2(\sqrt{H}/\epsilon)/\delta)}{\epsilon^2} \log_2(\sqrt{H}/\epsilon)$$

with probability $1 - \delta/2$, and the second part is to use (24) to let Practical OPDVR is identical to the non-implementable version with probability $1 - \delta/2$. Apply a union bound of these two gives the stated results in Theorem 3.2 with probability $1 - \delta$. ∎

## C    Proofs for finite-horizon stationary setting

Again, recall $\tilde{z}_t(s, a)$, $\tilde{\sigma}_{V_{t+1}^{\text{in}}}(s, a)$ (61) and $g_t$ (62) are three quantities deployed in Algorithm 1 that use off-policy data $\mathcal{D}$. We restate their definition as follows:

$$\tilde{z}_t(s, a) = \begin{cases} P^\top(\cdot|s, a)V_{t+1}^{\text{in}}, & if \; n_{s,a} \leq \frac{1}{2}m \cdot \sum_{t=1}^H d_t^\mu(s, a), \\ \frac{1}{n_{s,a}} \sum_{i=1}^m \sum_{u=1}^H V_{t+1}^{\text{in}}(s_{u+1}^{(i)}) \cdot \mathbf{1}_{[s_u^{(i)}=s,a_u^{(i)}=a]}, & if \; n_{s,a} > \frac{1}{2}m \cdot \sum_{t=1}^H d_t^\mu(s, a). \end{cases}$$

$$\tilde{\sigma}_{V_{t+1}^{\text{in}}}(s, a) = \begin{cases} \sigma_{V_{t+1}^{\text{in}}}(s, a), & if \; n_{s,a} \leq \frac{1}{2}m \cdot \sum_{t=1}^H d_t^\mu(s, a), \\ \frac{1}{n_{s,a}} \sum_{i=1}^m \sum_{u=1}^H [V_{t+1}^{\text{in}}(s_{u+1}^{(i)})]^2 \cdot \mathbf{1}_{[s_u^{(i)}=s,a_u^{(i)}=a]} - \tilde{z}_t^2(s, a), & otherwise. \end{cases}$$

$$g_t(s, a) = \begin{cases} P^\top(\cdot|s, a)[V_{t+1} - V_{t+1}^{\text{in}}] - f(s, a), & if \; n'_{s,a} \leq \frac{1}{2}l \cdot \sum_{t=1}^H d_t^\mu(s, a), \\ \frac{1}{n'_{s,a}} \sum_{j=1}^l \sum_{u=1}^H [V_{t+1}(s_{u+1}'^{(j)}) - V_t^{\text{in}}(s_{u+1}'^{(j)})] \cdot \mathbf{1}_{[s_u'^{(j)},a_u'^{(j)}=s,a]} - f(s, a), & o.w. \end{cases}$$

where

$$n_{s,a} = \sum_{i=1}^{m} \sum_{t=1}^{H} \mathbf{1}[s_t^{(i)} = s, a_t^{(i)} = a], \tag{25}$$

and recall $f(s,a) = 4u^{\text{in}} \sqrt{\log(2HSA/\delta)/l \sum_{t=1}^{H} d_t^{\mu}(s,a)}$.

**Lemma C.1.** *Let $\tilde{z}_t$ be defined as (61) in Algorithm 1, where $\tilde{z}_t$ is the off-policy estimator of $P^{\top}(\cdot|s,a)V_{t+1}^{in}$ using $m$ episodic data. Then with probability $1 - \delta$, we have*

$$\left| \tilde{z}_t - \boldsymbol{P}_t V_{t+1}^{in} \right| \leq \left( \sqrt{\frac{16 \cdot \sigma_{V_{t+1}^{in}} \cdot \log(HSA/\delta)}{m \sum_{t=1}^{H} d_t^{\mu}}} + \sqrt{\frac{16 V_{\max} \cdot \log(2HSA/\delta)}{9m \sum_{t=1}^{H} d_t^{\mu}}} \cdot \log(HSA/\delta) \right), \quad \forall t \in [H] \tag{26}$$

*here $\tilde{z}_t, \boldsymbol{P}_t V_{t+1}^{in}, \sigma_{V_{t+1}^{in}}, d_t^{\mu} \in \mathbb{R}^{S \times A}$ are $S \times A$ column vectors and $\sqrt{\cdot}$ is elementwise operation.*

*Proof.* Consdier fixed $s, a$. Let $E_{s,a} := \{n_{s,a} \geq \frac{1}{2}m \cdot \sum_{t=1}^{H} d_t^{\mu}(s,a)\}$, then by definition,

$$\tilde{z}_t(s,a) - P^{\top}(\cdot|s,a)V_{t+1}^{in} = \left( \frac{1}{n_{s,a}} \sum_{i=1}^{m} \sum_{u=1}^{H} V_{t+1}^{in}(s_{u+1}^{(i)}) \cdot \mathbf{1}[s_u^{(i)} = s, a_u^{(i)} = a] - P^{\top}(\cdot|s,a)V_{t+1}^{in} \right) \cdot \mathbf{1}(E_{s,a}).$$

First note by (25)

$$\mathbb{E}[n_{s,a}] = \sum_{i=1}^{m} \sum_{t=1}^{H} \mathbb{E}\left[ \mathbf{1}[s_t^{(i)} = s, a_t^{(i)} = a] \right] = \sum_{i=1}^{m} \sum_{t=1}^{H} d_t^{\mu}(s,a) = m \sum_{t=1}^{H} d_t^{\mu}(s,a).$$

Next we conditional on $n_{s,a}$. Define $\mathcal{F}_k := \{s_u^{(i)}, a_u^{(i)}\}_{i \in [m]}^{u \in [k]}$ is an increasing filtration and denote

$$X := \sum_{i=1}^{m} \sum_{u=1}^{H} \left( V_{t+1}^{in}(s_{u+1}^{(i)}) - P^{\top}(\cdot|s,a)V_{t+1}^{in} \right) \cdot \mathbf{1}[s_u^{(i)} = s, a_u^{(i)} = a],$$

then by tower property $\mathbb{E}[X|\mathcal{F}(Y)] = \mathbb{E}[\mathbb{E}[X|\mathcal{F}(Y,Z)]|\mathcal{F}(Y)]$ (the fourth equal sign in below)

$$\begin{aligned}
X_k := \mathbb{E}[X|\mathcal{F}_k] &= \mathbb{E}\left[ \sum_{i=1}^{m} \sum_{u=1}^{H} \left( V_{t+1}^{in}(s_{u+1}^{(i)}) - P^{\top}(\cdot|s,a)V_{t+1}^{in} \right) \cdot \mathbf{1}[s_u^{(i)} = s, a_u^{(i)} = a] \middle| \mathcal{F}_k \right] \\
&= \sum_{i=1}^{m} \mathbb{E}\left[ \sum_{u=1}^{H} \left( V_{t+1}^{in}(s_{u+1}^{(i)}) - P^{\top}(\cdot|s,a)V_{t+1}^{in} \right) \cdot \mathbf{1}[s_u^{(i)} = s, a_u^{(i)} = a] \middle| \mathcal{F}_k \right] \\
&= \sum_{i=1}^{m} \mathbb{E}\left[ \sum_{u=k}^{H} \left( V_{t+1}^{in}(s_{u+1}^{(i)}) - P^{\top}(\cdot|s,a)V_{t+1}^{in} \right) \cdot \mathbf{1}[s_u^{(i)} = s, a_u^{(i)} = a] \middle| \mathcal{F}_k \right] \\
&\quad + \sum_{i=1}^{m} \sum_{u=1}^{k-1} \left( V_{t+1}^{in}(s_{u+1}^{(i)}) - P^{\top}(\cdot|s,a)V_{t+1}^{in} \right) \cdot \mathbf{1}[s_u^{(i)} = s, a_u^{(i)} = a] \\
&= \sum_{i=1}^{m} \sum_{u=k}^{H} \mathbb{E}\left[ \mathbb{E}\left[ \left( V_{t+1}^{in}(s_{u+1}^{(i)}) - P^{\top}(\cdot|s,a)V_{t+1}^{in} \right) \cdot \mathbf{1}[s_u^{(i)} = s, a_u^{(i)} = a] \middle| \mathcal{F}_u \right] \middle| \mathcal{F}_k \right] \\
&\quad + \sum_{i=1}^{m} \sum_{u=1}^{k-1} \left( V_{t+1}^{in}(s_{u+1}^{(i)}) - P^{\top}(\cdot|s,a)V_{t+1}^{in} \right) \cdot \mathbf{1}[s_u^{(i)} = s, a_u^{(i)} = a] \\
&= \sum_{i=1}^{m} \sum_{u=k}^{H} \mathbb{E}\left[ \mathbf{1}[s_u^{(i)} = s, a_u^{(i)} = a] \mathbb{E}\left[ \left( V_{t+1}^{in}(s_{u+1}^{(i)}) - P^{\top}(\cdot|s,a)V_{t+1}^{in} \right) \middle| s_u^{(i)}, a_u^{(i)} \right] \middle| \mathcal{F}_u \right] \\
&\quad + \sum_{i=1}^{m} \sum_{u=1}^{k-1} \left( V_{t+1}^{in}(s_{u+1}^{(i)}) - P^{\top}(\cdot|s,a)V_{t+1}^{in} \right) \cdot \mathbf{1}[s_u^{(i)} = s, a_u^{(i)} = a]
\end{aligned}$$

Note if $\mathbf{1}[s_u^{(i)} = s, a_u^{(i)} = a] = 1$, then

$$\mathbf{1}[s_u^{(i)} = s, a_u^{(i)} = a]\mathbb{E}\big[\big(V_{t+1}^{\text{in}}(s_{u+1}^{(i)}) - P^\top(\cdot|s,a)V_{t+1}^{\text{in}}\big)\big|s_u^{(i)}, a_u^{(i)}\big]$$

$$=1 \cdot \mathbb{E}\big[\big(V_{t+1}^{\text{in}}(s_{u+1}^{(i)}) - P^\top(\cdot|s,a)V_{t+1}^{\text{in}}\big)\big|s_u^{(i)} = s, a_u^{(i)} = a\big]$$

$$=\mathbb{E}\big[V_{t+1}^{\text{in}}(s_{u+1}^{(i)})\big|s_u^{(i)} = s, a_u^{(i)} = a\big] - P^\top(\cdot|s,a)V_{t+1}^{\text{in}}$$

$$=P^\top(\cdot|s,a)V_{t+1}^{\text{in}} - P^\top(\cdot|s,a)V_{t+1}^{\text{in}} = 0$$

if $\mathbf{1}[s_u^{(i)} = s, a_u^{(i)} = a] = 0$, then still

$$\mathbf{1}[s_u^{(i)} = s, a_u^{(i)} = a]\mathbb{E}\big[\big(V_{t+1}^{\text{in}}(s_{u+1}^{(i)}) - P^\top(\cdot|s,a)V_{t+1}^{\text{in}}\big)\big|s_u^{(i)}, a_u^{(i)}\big] = 0$$

So plug back to obtain

$$X_k = \sum_{i=1}^{m}\sum_{u=1}^{k-1}\big(V_{t+1}^{\text{in}}(s_{u+1}^{(i)}) - P^\top(\cdot|s,a)V_{t+1}^{\text{in}}\big) \cdot \mathbf{1}[s_u^{(i)} = s, a_u^{(i)} = a].$$

is a martingale.

First of all by Hoeffding's inequality, we have the martingale difference satisfies with probability $1 - \delta/2$,

$$|X_{k+1} - X_k| = \left|\sum_{i=1}^{m}\big(V_{t+1}^{\text{in}}(s_{k+1}^{(i)}) - P^\top(\cdot|s,a)V_{t+1}^{\text{in}}\big) \cdot \mathbf{1}[s_k^{(i)} = s, a_k^{(i)} = a]\right|$$

$$= \left|\sum_{i=1}^{n_{k,s,a}}\big(V_{t+1}^{\text{in}}(s_{k+1}^{(i)}|s,a) - P^\top(\cdot|s,a)V_{t+1}^{\text{in}}\big)\right| \qquad (27)$$

$$\leq \sqrt{2n_{k,s,a} \cdot V_{\max}\log(2/\delta)} \leq \sqrt{2n_{s,a} \cdot V_{\max}\log(2/\delta)}$$

where we use shorthand notation $V_{t+1}^{\text{in}}(s_{k+1}^{(i)}|s,a)$ to denote the value of $V_{k+1}^{\text{in}}(s_{k+1}^{(i)})$ given $s_k^{(i)} = s$ and $a_k^{(i)} = a$ and $n_{k,s,a} = \sum_{i=1}^{m}\mathbf{1}[s_k^{(i)} = s, a_k^{(i)} = a] \leq n_{s,a}$.

Second,

$$\text{Var}\,[X_{k+1}|\mathcal{F}_k] = \text{Var}\left[\sum_{i=1}^{m}\big(V_{t+1}^{\text{in}}(s_{k+1}^{(i)}) - P^\top(\cdot|s,a)V_{t+1}^{\text{in}}\big) \cdot \mathbf{1}[s_k^{(i)} = s, a_k^{(i)} = a]\bigg|\mathcal{F}_k\right]$$

$$= \sum_{i=1}^{m}\text{Var}\left[\big(V_{t+1}^{\text{in}}(s_{k+1}^{(i)}) - P^\top(\cdot|s,a)V_{t+1}^{\text{in}}\big) \cdot \mathbf{1}[s_k^{(i)} = s, a_k^{(i)} = a]\bigg|\mathcal{F}_k\right]$$

$$= \sum_{i=1}^{m}\text{Var}\left[\big(V_{t+1}^{\text{in}}(s_{k+1}^{(i)}) - P^\top(\cdot|s,a)V_{t+1}^{\text{in}}\big) \cdot \mathbf{1}[s_k^{(i)} = s, a_k^{(i)} = a]\bigg|s_k^{(i)}, a_k^{(i)}\right]$$

$$= \sum_{i=1}^{m}\mathbf{1}[s_k^{(i)} = s, a_k^{(i)} = a]\text{Var}\left[V_{t+1}^{\text{in}}(s_{k+1}^{(i)})\bigg|s_k^{(i)}, a_k^{(i)}\right]$$

$$= \sum_{i=1}^{m}\mathbf{1}[s_k^{(i)} = s, a_k^{(i)} = a]\text{Var}\left[V_{t+1}^{\text{in}}(s_{k+1}^{(i)})\bigg|s_k^{(i)} = s, a_k^{(i)} = a\right]$$

$$= \sum_{i=1}^{m}\mathbf{1}[s_k^{(i)} = s, a_k^{(i)} = a] \cdot \sigma_{V_{t+1}^{\text{in}}}(s,a).$$

where the second equal sign uses episodes are independent, the third equal sign uses Markov property, the fourth uses $\mathbf{1}[s_k^{(i)} = s, a_k^{(i)} = a]$ is measurable w.r.t $s_k^{(i)}, a_k^{(i)}$ and $P^\top(\cdot|s,a)V_{t+1}^{\text{in}}$ is constant, the fifth equal sign uses the identity

$$\mathbf{1}[s_k^{(i)} = s, a_k^{(i)} = a]\text{Var}\left[V_{t+1}^{\text{in}}(s_{k+1}^{(i)})\bigg|s_k^{(i)}, a_k^{(i)}\right] = \mathbf{1}[s_k^{(i)} = s, a_k^{(i)} = a]\text{Var}\left[V_{t+1}^{\text{in}}(s_{k+1}^{(i)})\bigg|s_k^{(i)} = s, a_k^{(i)} = a\right]$$

and sixth line is true since we have stationary transition, the underlying transition is always $P(\cdot|s,a)$ regardless of time step. This is the key for further reducing the dependence on $H$ and is **NOT** shared by non-stationary transition setting!

Therefore finally,

$$\sum_{k=1}^{H} \text{Var}\left[X_{k+1}|\mathcal{F}_k\right] = \sum_{k=1}^{H}\sum_{i=1}^{m} \mathbf{1}[s_k^{(i)} = s, a_k^{(i)} = a] \cdot \sigma_{V_{t+1}^{\text{in}}}(s,a) = n_{s,a} \cdot \sigma_{V_{t+1}^{\text{in}}}(s,a). \quad (28)$$

Recall that $s,a$ is fixed and we conditional on $n_{s,a}$. Also note by tower property $\mathbb{E}[X] = 0$. Therefore by (27), (28) Freedman's inequality (Lemma G.4) with probability[7] $1-\delta$

$$|X| = \left|\sum_{i=1}^{m}\sum_{u=1}^{H}\left(V_{t+1}^{\text{in}}(s_{u+1}^{(i)}) - P^\top(\cdot|s,a)V_{t+1}^{\text{in}}\right) \cdot \mathbf{1}[s_u^{(i)} = s, a_u^{(i)} = a]\right|$$

$$\leq \sqrt{8 n_{s,a} \cdot \sigma_{V_{t+1}^{\text{in}}}(s,a) \cdot \log(1/\delta)} + \frac{2\sqrt{2 n_{s,a} \cdot V_{\max}\log(2/\delta)}}{3} \cdot \log(1/\delta).$$

which means with probability at least $1-\delta$

$$\left|\tilde{z}_t(s,a) - P^\top(\cdot|s,a)V_{t+1}^{\text{in}}\right|$$

$$= \left|\frac{X}{n_{s,a}}\right| \cdot \mathbf{1}(E_{s,a})$$

$$\leq \left(\frac{\sqrt{8 n_{s,a} \cdot \sigma_{V_{t+1}^{\text{in}}}(s,a) \cdot \log(1/\delta)} + \frac{2\sqrt{2 n_{s,a} \cdot V_{\max}\log(2/\delta)}}{3} \cdot \log(1/\delta)}{n_{s,a}}\right) \cdot \mathbf{1}(E_{s,a})$$

$$= \left(\sqrt{\frac{8 \cdot \sigma_{V_{t+1}^{\text{in}}}(s,a) \cdot \log(1/\delta)}{n_{s,a}}} + \sqrt{\frac{8 \cdot V_{\max}}{9 n_{s,a}} \cdot \log(2/\delta) \cdot \log(1/\delta)}\right) \cdot \mathbf{1}(E_{s,a})$$

$$\leq \left(\sqrt{\frac{16 \cdot \sigma_{V_{t+1}^{\text{in}}}(s,a) \cdot \log(1/\delta)}{m\sum_{t=1}^{H} d_t^\mu(s,a)}} + \sqrt{\frac{16 V_{\max} \cdot \log(2/\delta)}{9m\sum_{t=1}^{H} d_t^\mu(s,a)} \cdot \log(1/\delta)}\right)$$

Now we get rid of the conditional on $n_{s_t, a_t}$. Denote

$$A = \left\{\left|\tilde{z}_t(s,a) - P^\top(\cdot|s,a)V_{t+1}^{\text{in}}\right| \leq \left(\sqrt{\frac{16 \cdot \sigma_{V_{t+1}^{\text{in}}}(s,a) \cdot \log(1/\delta)}{m\sum_{t=1}^{H} d_t^\mu(s,a)}} + \sqrt{\frac{16 V_{\max} \cdot \log(2/\delta)}{9m\sum_{t=1}^{H} d_t^\mu(s,a)} \cdot \log(1/\delta)}\right)\right\},$$

then equivalently we can rewrite above result as $\mathbb{P}(A|n_{s,a}) \geq 1-\delta$. Note this is the same as $\mathbb{E}[\mathbf{1}(A)|n_{s,a}] \geq 1-\delta$, therefore by law of total expectation we have

$$\mathbb{P}(A) = \mathbb{E}[\mathbf{1}(A)] = \mathbb{E}[\mathbb{E}[\mathbf{1}(A)|n_{s,a}]] \geq \mathbb{E}[1-\delta] = 1-\delta,$$

Finally, apply the union bound over all $t,s,a$, we obtain

$$\left|\tilde{z}_t - \boldsymbol{P}_t V_{t+1}^{\text{in}}\right| \leq \left(\sqrt{\frac{16 \cdot \sigma_{V_{t+1}^{\text{in}}} \cdot \log(HSA/\delta)}{m\sum_{t=1}^{H} d_t^\mu}} + \sqrt{\frac{16 V_{\max} \cdot \log(2HSA/\delta)}{9m\sum_{t=1}^{H} d_t^\mu} \cdot \log(HSA/\delta)}\right),$$

where the inequality is element-wise and this is (5). $\blacksquare$

---

[7]To be mathematically rigorous, the difference bound is not with probability 1 but in the high probability sense. Therefore essentially we are using a weaker version of freedman's inequality that with high probability bounded difference, *e.g.* see Chung and Lu [2006] Theorem 34,37. We do not present our result by explicitly writing in that way in order to prevent over-technicality and make the readers easier to understand.

**Lemma C.2.** *Let $\tilde{\sigma}_{V_{t+1}^{in}}$ be defined as* (61) *in Algorithm 1, the off-policy estimator of $\sigma_{V_{t+1}^{in}}(s,a)$ using $m$ episodic data. Then with probability $1-\delta$, we have*

$$\left|\tilde{\sigma}_{V_{t+1}^{in}} - \sigma_{V_{t+1}^{in}}\right| \leq 6V_{\max}^2 \sqrt{\frac{\log(4HSA/\delta)}{m \cdot \sum_{t=1}^H d_t^\mu}} + \frac{4V_{\max}^2 \log(4HSA/\delta)}{m \cdot \sum_{t=1}^H d_t^\mu}, \quad \forall t = 1, ..., H. \quad (29)$$

*Proof.* From the definition we have for fixed $(s,a)$

$$\tilde{\sigma}_{V_{t+1}^{in}}(s,a) - \sigma_{V_{t+1}^{in}}(s,a)$$

$$= \left(\frac{1}{n_{s,a}} \sum_{i=1}^m \sum_{u=1}^H \left[V_{t+1}^{in}(s_{u+1}^{(i)})^2 - P^\top(\cdot|s,a)(V_{t+1}^{in})^2\right] \mathbf{1}[s_u^{(i)} = s, a^{(i)} = a]\right) \mathbf{1}(E_{s,a})$$

$$+ \left(\left[\frac{1}{n_{s,a}} \sum_{i=1}^m \sum_{u=1}^H V_{t+1}^{in}(s_{u+1}^{(i)})\mathbf{1}[s_u^{(i)} = s, a^{(i)} = a]\right]^2 - \left[P^\top(\cdot|s,a)V_{t+1}^{in}\right]^2\right) \mathbf{1}(E_{s,a})$$

Now we conditional on $n_{s,a}$. The key point is we can regroup $m$ episodic data into $mH$ data pieces, in order. (This is valid since within each episode data is generated by time and between different episodes are independent, so we can concatenate one episode after another and end up with $mH$ pieces that comes in sequentially.) This key reformulation allows us to apply Azuma Hoeffding's inequality and obtain with probability $1 - \delta/2$,

$$\left(\frac{1}{n_{s,a}} \sum_{i=1}^m \sum_{u=1}^H \left[V_{t+1}^{in}(s_{u+1}^{(i)})^2 - P^\top(\cdot|s,a)(V_{t+1}^{in})^2\right] \mathbf{1}[s_u^{(i)} = s, a_u^{(i)} = a]\right) \mathbf{1}(E_{s,a})$$

$$= \left(\frac{1}{n_{s,a}} \sum_{u'=1}^{n_{s,a}} \left[V_{t+1}^{in}(s_{u'+1}^{(i)})^2 - P^\top(\cdot|s,a)(V_{t+1}^{in})^2\right] \mathbf{1}[s_{u'}^{(i)} = s, a_{u'}^{(i)} = a]\right) \mathbf{1}(E_{s,a}) \quad (30)$$

$$\leq V_{\max}^2 \sqrt{\frac{2\log(4/\delta)}{n_{s,a}}} \cdot \mathbf{1}(E_{s,a}) \leq 2V_{\max}^2 \sqrt{\frac{\log(4/\delta)}{m \cdot \sum_{t=1}^H d_t^\mu(s,a)}},$$

where the first equal sign comes from the reformulation trick and the first inequality is by $X_k := \sum_{u'=1}^k \left[V_{t+1}^{in}(s_{u'+1}^{(i)})^2 - P^\top(\cdot|s_t,a_t)(V_{t+1}^{in})^2\right] \mathbf{1}[s_{u'}^{(i)} = s, a_{u'}^{(i)} = a]$ is martingale. Similarly with probability $1 - \delta/2$,

$$\left(\frac{1}{n_{s,a}} \sum_{u'=1}^{n_{s_t,a_t}} V_{t+1}^{in}(s_{u'+1}^{(i)}|s,a) - P^\top(\cdot|s,a)V_{t+1}^{in}\right) \mathbf{1}(E_{s,a}) \leq 2V_{\max} \sqrt{\frac{\log(4/\delta)}{m \cdot \sum_{t=1}^H d_t^\mu(s,a)}}. \quad (31)$$

Note for $a, b, c > 0$, if $|a - b| \leq c$, then $|a^2 - b^2| = |a - b| \cdot |a + b| \leq |a - b| \cdot (|a| + |b|) \leq |a - b| \cdot (2|b| + c) \leq c \cdot (2|b| + c) = 2bc + c^2$, therefore by (31) we have

$$\left(\left[\frac{1}{n_{s,a}} \sum_{u'=1}^{n_{s,a}} V_{t+1}^{in}(s_{u'+1}^{(i)}|s,a)\right]^2 - \left[P^\top(\cdot|s,a)V_{t+1}^{in}\right]^2\right) \mathbf{1}(E_{s,a})$$

$$\leq 4P^\top(\cdot|s,a)V_{t+1}^{in} \cdot V_{\max} \sqrt{\frac{\log(4/\delta)}{m \cdot \sum_{t=1}^H d_t^\mu(s,a)}} + \frac{4V_{\max}^2 \log(4/\delta)}{m \cdot \sum_{t=1}^H d_t^\mu(s,a)} \quad (32)$$

$$\leq 4V_{\max}^2 \sqrt{\frac{\log(4/\delta)}{m \cdot \sum_{t=1}^H d_t^\mu(s,a)}} + \frac{4V_{\max}^2 \log(4/\delta)}{m \cdot \sum_{t=1}^H d_t^\mu(s,a)}$$

where the last inequality comes from $|P^\top(\cdot|s,a)V_{t+1}^{in}| \leq ||P(\cdot|s,a)||_1 ||V_{t+1}^{in}||_\infty \leq V_{\max}$. Combining (30), (32) and a union bound, we have with probability $1 - \delta$,

$$\left|\tilde{\sigma}_{V_{t+1}^{in}}(s,a) - \sigma_{V_{t+1}^{in}}(s,a)\right| \leq 6V_{\max}^2 \sqrt{\frac{\log(4/\delta)}{m \cdot \sum_{t=1}^H d_t^\mu(s,a)}} + \frac{4V_{\max}^2 \log(4/\delta)}{m \cdot \sum_{t=1}^H d_t^\mu(s,a)},$$

apply again the union bound over $t, s, a$ gives the desired result.

∎

**Lemma C.3.** *Fix time $t \in [H]$. Let $g_t$ be the estimator in (60) in Algorithm 1. Then if $||V_{t+1} - V_{t+1}^{in}||_\infty \leq 2u^{in}$, then with probability $1 - \delta/H$,*

$$\mathbf{0} \leq \mathbf{P}[V_{t+1} - V_{t+1}^{in}] - g_t \leq 8u^{in}\sqrt{\frac{\log(2HSA/\delta)}{l \sum_{t=1}^{H} d_t^\mu}}$$

*Proof.* Recall $g_t, d_t^\mu$ are vectors. By definition of $g_t(s, a)$, use similar regrouping trick and apply Azuma Hoeffding's inequality we obtain with probability $1 - \delta/H$

$$g_t(s, a) + f(s, a) - P^\top(\cdot|s, a)[V_{t+1} - V_{t+1}^{in}]$$

$$= \left( \frac{1}{n'_{s,a}} \sum_{u'=1}^{n'_{s,a}} \left[ V_{t+1}(s'^{(j)}_{u'+1}|s, a) - V_{t+1}^{in}(s'^{(j)}_{u'+1}|s, a) \right] - P^\top(\cdot|s, a)[V_{t+1} - V_{t+1}^{in}] \right) \cdot \mathbf{1}(E_t)$$

$$\leq \left( ||V_{t+1} - V_{t+1}^{in}||_\infty \sqrt{\frac{2\log(2H/\delta)}{n'_{s,a}}} \right) \cdot \mathbf{1}(E_t)$$

$$\leq ||V_{t+1} - V_{t+1}^{in}||_\infty \sqrt{\frac{4\log(2H/\delta)}{l \cdot \sum_{t=1}^{H} d_t^\mu(s, a)}}$$

Now use assumption $||V_{t+1} - V_{t+1}^{in}||_\infty \leq 2u^{in}$ and a union bound over $s, a$, we have with probability $1 - \delta/H$,

$$\left| g_t + f - \mathbf{P}[V_{t+1} - V_{t+1}^{(0)}] \right| \leq 4u^{in}\sqrt{\frac{\log(2HSA/\delta)}{l \sum_{t=1}^{H} d_t^\mu}} \tag{33}$$

use $f = 4u^{in}\sqrt{\log(2HSA/\delta)/l \sum_{t=1}^{H} d_t^\mu}$, we obtain the stated result. ∎

**Proof of Theorem 4.1** Note that Lemma C.1,C.2,C.3 updates Lemma B.2,B.4,B.5 by replacing $d_t^\mu$ with $\sum_{t=1}^{H} d_t^\mu$ and keeping the rest the same except the second order term $\sqrt{\frac{16V_{\max} \cdot \log(2HSA/\delta)}{9m \sum_{t=1}^{H} d_t^\mu}} \cdot \log(HSA/\delta)$ in Lemma C.1 is different from Lemma C.1. However, this is still lower order term since it is of order $\widetilde{O}(\sqrt{\frac{H}{m \sum_{t=1}^{H} d_t^\mu}})$. To avoid redundant reasoning, by following the identical logic as Section B.2 we have a similar expression of (14) as follows:

$$z_t \geq \mathbf{P}V_{t+1}^{in} - \sqrt{\frac{16 \cdot \sigma_{V_{t+1}^\star} \cdot \log(4HSA/\delta)}{m \cdot \sum_{t=1}^{H} d_t^\mu}} - \sqrt{\frac{16 \cdot \log(4HSA/\delta)}{m \cdot \sum_{t=1}^{H} d_t^\mu}} \cdot u^{in}$$

$$- V_{\max} \left[ 8\sqrt{6} \cdot \left( \frac{\log(16HSA/\delta)}{m \cdot \sum_{t=1}^{H} d_t^\mu} \right)^{3/4} + \frac{56\log(16HSA/\delta)}{3m \cdot \sum_{t=1}^{H} d_t^\mu} \right] - c\sqrt{\frac{V_{\max} \cdot \log(16HSA/\delta)}{m \cdot \sum_{t=1}^{H} d_t^\mu}} \log(HSA/\delta). \tag{34}$$

where the last term is additional. However, note when $u^{in} \leq \sqrt{H}$, then

$$\sqrt{\frac{16 \cdot \log(4HSA/\delta)}{m \cdot \sum_{t=1}^{H} d_t^\mu}} \cdot u^{in} \leq \tilde{O}\left( \sqrt{\frac{H}{m \sum_{t=1}^{H} d_t^\mu}} \right), \qquad c\sqrt{\frac{V_{\max} \cdot \log(16HSA/\delta)}{m \cdot \sum_{t=1}^{H} d_t^\mu}} \leq \tilde{O}\left( \sqrt{\frac{H}{m \sum_{t=1}^{H} d_t^\mu}} \right)$$

so the last term $c\sqrt{\frac{V_{\max} \cdot \log(16HSA/\delta)}{m \cdot \sum_{t=1}^{H} d_t^\mu}}$ can be assimilated by previous one. If $u^{in} > \sqrt{H}$, it is of even lower order. Therefore following the same reasoning we can complete the proof for Algorithm 1.

From non-implementable version to the practical version, we need to bound the event of $\{n_{s,a} \leq \frac{1}{2}m \cdot \sum_{t=1}^{H} d_t^\mu(s, a)\}$, where $n_{s,a} = \sum_{i=1}^{m} \sum_{t=1}^{H} \mathbf{1}[s_t^{(i)} = s, a_t^{(i)} = a]$. In this case, $n_{s,a}$ is no longer

binomial random variable so Lemma G.2 cannot be applied. However, the trick we use for resolving this issue is the following decomposition

$$\left\{ n_{s,a} \le \frac{1}{2}m \cdot \sum_{t=1}^{H} d_t^\mu(s,a) \right\} \subset \bigcup_{t=1}^{H} \left\{ n_{t,s,a} \le \frac{1}{2}m d_t^\mu(s,a) \right\},$$

where $n_{s,a} = \sum_{t=1}^{H} n_{t,s,a}$ and $n_{t,s,a} = \sum_{i=1}^{m} \mathbf{1}[s_t^{(i)} = s, a_t^{(i)} = a]$ are binomial random variables. Lemma G.2 can then be used together with union bounds to finish the proof.

## D  Proofs for infinite-horizon discounted setting

First recall data $\mathcal{D} = \{s^{(i)}, a^{(i)}, r^{(i)}, s'^{(i)}\}_{i \in [n]}$ are i.i.d off-policy pieces with $(s^{(i)}, a^{(i)}) \sim d^\mu$ and $s'^{(i)} \sim P(\cdot|s^{(i)}, a^{(i)})$. Moreover, $d^\mu$ is defined as:

$$d^\mu(s) = (1 - \gamma) \sum_{t=0}^{\infty} \gamma^t \mathbb{P}[s_t = s | s_0 \sim d_0, \mu], \quad d^\mu(s,a) = d^\mu(s)\mu(a|s).$$

The corresponding estimators in Algorithm 3 are defined as:

$$\tilde{z}(s,a) = \begin{cases} P^\top(\cdot|s,a)V^{\text{in}}, & \text{if } n_{s,a} \le \frac{1}{2}m \cdot d^\mu(s,a), \\ \frac{1}{n_{s,a}} \sum_{i=1}^{m} V^{\text{in}}(s'^{(i)}) \cdot \mathbf{1}_{[s^{(i)}=s, a^{(i)}=a]}, & \text{if } n_{s,a} > \frac{1}{2}m \cdot d^\mu(s,a). \end{cases}$$

$$\tilde{\sigma}_{V^{\text{in}}}(s,a) = \begin{cases} \sigma_{V^{\text{in}}}(s,a), & \text{if } n_{s,a} \le \frac{1}{2}m \cdot d^\mu(s,a), \\ \frac{1}{n_{s,a}} \sum_{i=1}^{m} [V^{\text{in}}(s'^{(i)})]^2 \cdot \mathbf{1}_{[s^{(i)}=s, a^{(i)}=a]} - \tilde{z}^2(s,a), & \text{otherwise}. \end{cases}$$
(35)

where $n_{s,a} := \sum_{i=1}^{n} \mathbf{1}[s^{(i)} = s, a^{(i)} = a]$ is the number of samples start at $(s,a)$. Similarly, $P^\top(\cdot|s,a)[V - V^{\text{in}}]$ is later updated using different $l$ episodes ($n'_{s,a}$ is the number count from $l$ episodes):

$$g^{(i)}(s,a) = \begin{cases} P^\top(\cdot|s,a)[V^{(i)} - V^{\text{in}}] - f(s,a), & \text{if } n'_{s,a} \le \frac{1}{2}l \cdot d^\mu(s,a), \\ \frac{1}{n'_{s,a}} \sum_{j=1}^{l} [V^{(i)}(s'^{(j)}) - V^{\text{in}}(s'^{(j)})] \cdot \mathbf{1}_{[s'^{(j)}, a'^{(j)}=s,a]} - f(s,a), & \text{o.w.} \end{cases}$$
(36)

where $f = 4u^{\text{in}}\sqrt{\log(2RSA/\delta)/ld^\mu}$ and $R = \ln(4/u^{\text{in}}(1-\gamma))$.

**Lemma D.1.** *Suppose $V$ and $\pi$ is any value and policy satisfy $V \le \mathcal{T}_\pi V$. Then it holds $V \le V^\pi \le V^\star$.*

*Proof.* This is similar to Lemma B.1 and the key is to use Bellman equation $V^\pi = \mathcal{T}_\pi V^\pi$. ∎

**Lemma D.2.** *Let $\tilde{z}$ be defined as (35) in Algorithm 3, where $\tilde{z}$ is the off-policy estimator of $P^\top(\cdot|s,a)V^{in}$ using $m$ episodic data. Then with probability $1 - \delta$, we have*

$$\left| \tilde{z} - \boldsymbol{P}V^{in} \right| \le \sqrt{\frac{4 \cdot \sigma_{V^{in}} \cdot \log(SA/\delta)}{m \cdot d^\mu}} + \frac{4V_{\max}}{3m \cdot d^\mu} \log(SA/\delta).$$
(37)

*here $\tilde{z}, \boldsymbol{P}V^{in}, \sigma_{V^{in}}, d^\mu \in \mathbb{R}^{S \times A}$ are $S \times A$ column vectors and $\sqrt{\cdot}$ is elementwise operation.*

*Proof.* First fix $s,a$. Let $E_{s,a} := \{n_{s,a} \ge \frac{1}{2}m \cdot d^\mu(s,a)\}$, then by definition,

$$\tilde{z}(s,a) - P^\top(\cdot|s,a)V_{t+1}^{\text{in}} = \left( \frac{1}{n_{s,a}} \sum_{i=1}^{m} V^{\text{in}}(s'^{(i)}) \cdot \mathbf{1}[s^{(i)} = s, a^{(i)} = a] - P^\top(\cdot|s,a)V^{\text{in}} \right) \cdot \mathbf{1}(E_{s,a}).$$

Next we conditional on $n_{s,a}$. Then from above expression and Bernstein inequality G.3 we have with probability at least $1 - \delta$

$$\left| \tilde{z}(s,a) - P^\top(\cdot|s,a)V^{\text{in}} \right|$$

$$= \left| \frac{1}{n_{s,a}} \sum_{i=1}^{n_{s,a}} V^{\text{in}}(s'^{(i)}|s,a) - P^\top(\cdot|s,a)V^{\text{in}} \right| \cdot \mathbf{1}(E_{s,a})$$

---

**Algorithm 3** OPVRT: A Prototypical Off-Policy Variance Reduction Template ($\infty$-horizon)

---

1: **Functional input:** Integer valued function $\mathbf{m} : \mathbb{R}_+ \to \mathbb{N}$. Off-policy estimator $\mathbf{z}_t, \mathbf{g}_t$ in function forms that provides lower confidence bounds (LCB) of the two terms in the bootstrapped value function (2).
2: **Static input:** Initial value function $V^{(0)}$ and $\pi^{(0)}$ (which satisfy $V^{(0)} \leq \mathcal{T}_{\pi^{(0)}} V^{(0)}$). A scalar $u^{(0)}$ satisfies $u^{(0)} \geq ||V^\star - V^{(0)}||_\infty$. Outer loop iterations $K$. Offline dataset $\mathcal{D} = \{s^{(i)}, a^{(i)}, r^{(i)}, s'^{(i)}\}_{i=1}^n$ from the behavior policy $\mu$ as a data-stream where $n \geq \sum_{i=1}^K (1 + R) \cdot \mathbf{m}(u^{(0)} \cdot 2^{-(i-1)})$.
3: ――――――――INNER LOOP――――――――
4: **function** QVI-VR-INF $(\mathcal{D}_1, [\mathcal{D}_2^{(i)}]_{i=1}^K, V_t^{\text{in}}, \pi^{\text{in}}, \mathbf{z}_t, \mathbf{g}_t, u^{\text{in}})$
5:    $\diamond$ Computing reference with $\mathcal{D}_1$:
6:    Initialize $Q^{(0)} \leftarrow \mathbf{0} \in \mathbb{R}^{\mathcal{S} \times \mathcal{A}}$ and $V^{(0)} = V_{\text{in}}$.
7:    **for** each pair $(s, a) \in \mathcal{S} \times \mathcal{A}$ **do**
8:      $\diamond$ Compute an LCB of $P^\top(\cdot|s, a)V^{in}$:
9:      $z \leftarrow \mathbf{z}(\mathcal{D}_1, V^{\text{in}}, u^{\text{in}})$
10:    **end for**
11:    $\diamond$ Value Iterations with $\mathcal{D}_2$:
12:    **for** $i = 1, ..., R$ **do**
13:      $\diamond$ Update value function: $V^{(i)} = \max(V_{Q^{(i-1)}}, V^{\text{in}})$,
14:      $\diamond$ Update policy according to value function:
15:      $\forall s$, if $V^{(i)}(s) = V^{(i-1)}(s)$ set $\pi(s) = \pi^{\text{in}}(s)$; else set $\pi(s) = \pi_{Q^{(i-1)}}(s)$.
16:      **if** $t \geq 1$ **then**
17:        $\diamond$ LCB of $P^\top(\cdot|s_{t-1}, a_{t-1})[V_t - V_t^{\text{in}}]$:
18:        $g^{(i)} \leftarrow \mathbf{g}(\mathcal{D}_2^{(i)}, V^{(i)}, V^{\text{in}}, u^{\text{in}})$.
19:        $\diamond$ Update $Q$ function: $Q^{(i)} \leftarrow r + \gamma z + \gamma g^{(i)}$
20:      **end if**
21:    **end for**
22:    **Return:** $V^{(R)}$ and $\pi^{(R)}$.
23: **end function**
24: ――――――――OUTER LOOP――――――――
25: **for** $j = 1, ..., K$ **do**
26:    $m^{(j)} \to \mathbf{m}(u^{(j-1)})$
27:    Get $\mathcal{D}_1$ and $\mathcal{D}_2^{(i)}$ for $i = 1, ..., K$ each with size $m^{(j)}$ from the stream $\mathcal{D}$.
28:    $V^{(j)}, \pi^{(j)} \leftarrow$ QVI-VR-INF $(\mathcal{D}_1, [\mathcal{D}_2^{(i)}]_{i=1}^K, V^{(i-1)}, \pi^{(i-1)}, \mathbf{z}, \mathbf{g}, u^{(i-1)})$.
29:    $u^{(j)} \leftarrow u^{(j-1)}/2$.
30: **end for**
31: **Output:** $V^{(K)}, \pi^{(K)}$

---

---

**Algorithm 4** OPDVR: Off-Policy Doubled Variance Reduction ($\infty$-horizon)

---

**input** Offline Dataset $\mathcal{D}$ of size $n$ as a stream. Target accuracy $\epsilon, \delta$ such that the algorithm does not use up $\mathcal{D}$.
**input** Estimators $\mathbf{z}, \mathbf{g}$ in function forms, $m_1', m_2', K_1, K_2$.
1: $\diamond$ Stage 1. coarse learning: a "warm-up" procedure
2: Set initial values $V^{(0)} := \mathbf{0}$ and any policy $\pi^{(0)}$.
3: Set initial $u^{(0)} := (1 - \gamma)^{-1}$.
4: Set $\mathbf{m}(u) = m_1' \log(16(1 - \gamma)^{-1} RSA)/u^2$.
5: Run Algorithm 3 with $\mathbf{m}, \mathbf{z}, \mathbf{g}, V^{(0)}, \pi^{(0)}, u^{(0)}, K_1, \mathcal{D}$ and return $V^{\text{intermediate}}, \pi^{\text{intermediate}}$.
6: $\diamond$ Stage 2. fine learning: reduce error to given accuracy
7: Reset initial values $V^{(0)} := V^{\text{intermediate}}$ and policy $\pi^{(0)} := \pi^{\text{intermediate}}$. Set $u^{(0)} := \sqrt{(1 - \gamma)^{-1}}$.
8: Reset $\mathbf{m}(u)$ by replacing $m_1'$ with $m_2'$, $K_1$ with $K_2$.
9: Run Algorithm 3 with $\mathbf{m}, \mathbf{z}, \mathbf{g}, V_t^{(0)}, \pi^{(0)}, u^{(0)}, K_2, \mathcal{D}$ and return $V_t^{\text{final}}, \pi^{\text{final}}$.
**output** $V_t^{\text{final}}, \pi^{\text{final}}$

---

$$\leq \left( \sqrt{\frac{2 \cdot \sigma_{V^{\text{in}}}(s, a) \cdot \log(1/\delta)}{n_{s,a}}} + \frac{2V_{\max}}{3n_{s,a}} \log(1/\delta) \right) \cdot \mathbf{1}(E_{s,a})$$

$$\leq \sqrt{\frac{4 \cdot \sigma_{V^{\text{in}}}(s, a) \cdot \log(1/\delta)}{m \cdot d^\mu(s, a)}} + \frac{4V_{\max}}{3m \cdot d^\mu(s, a)} \log(1/\delta)$$

where again notation $V^{\text{in}}(s'^{(i)}|s, a)$ denotes the value of $V^{\text{in}}(s'^{(i)})$ given $s^{(i)} = s$ and $a^{(i)} = a$. The condition $V^{\text{in}} \leq V_{\max}$ is guaranteed by Lemma B.1. Now we get rid of the conditional

on $n_{s,a}$. Denote $A = \{\tilde{z}(s,a) - P^\top(\cdot|s,a)V^{\text{in}} \leq \sqrt{4 \cdot \sigma_{V^{\text{in}}}(s,a) \cdot \log(1/\delta)/m \cdot d^\mu(s,a)} + \frac{4V_{\max}}{3m \cdot d^\mu(s,a)} \log(1/\delta)\}$, then equivalently we can rewrite above result as $\mathbb{P}(A|n_{s,a}) \geq 1 - \delta$. Note this is the same as $\mathbb{E}[\mathbf{1}(A)|n_{s,a}] \geq 1 - \delta$, therefore by law of total expectation we have

$$\mathbb{P}(A) = \mathbb{E}[\mathbf{1}(A)] = \mathbb{E}[\mathbb{E}[\mathbf{1}(A)|n_{s,a}]] \geq \mathbb{E}[1 - \delta] = 1 - \delta,$$

*i.e.* for fixed $(s,a)$ we have with probability at least $1 - \delta$,

$$\left|\tilde{z}(s,a) - P^\top(\cdot|s,a)V^{\text{in}}\right| \leq \sqrt{\frac{4 \cdot \sigma_{V^{\text{in}}}(s,a) \cdot \log(1/\delta)}{m \cdot d^\mu(s,a)}} + \frac{4V_{\max}}{3m \cdot d^\mu(s,a)} \log(1/\delta)$$

Apply the union bound over all $s, a$, we obtain

$$\left|\tilde{z} - \boldsymbol{P}V^{\text{in}}\right| \leq \sqrt{\frac{4 \cdot \sigma_{V^{\text{in}}} \cdot \log(SA/\delta)}{m \cdot d^\mu}} + \frac{4V_{\max}}{3m \cdot d^\mu} \log(SA/\delta),$$

where the inequality is element-wise and this is (37). ∎

**Lemma D.3.** *Let $\tilde{\sigma}_{V^{\text{in}}}$ be defined as* (35) *in Algorithm 3, the off-policy estimator of $\sigma_{V^{\text{in}}}(s,a)$ using $m$ episodic data. Then with probability $1 - \delta$, we have*

$$|\tilde{\sigma}_{V^{\text{in}}} - \sigma_{V^{\text{in}}}| \leq 6V_{\max}^2 \sqrt{\frac{\log(4SA/\delta)}{m \cdot d^\mu}} + \frac{4V_{\max}^2 \log(4SA/\delta)}{m \cdot d^\mu}. \tag{38}$$

*Proof.* From the definition we have for fixed $(s,a)$

$$\tilde{\sigma}_{V^{\text{in}}}(s,a) - \sigma_{V^{\text{in}}}(s,a) = \left(\frac{1}{n_{s,a}} \sum_{i=1}^{n_{s,a}} V^{\text{in}}(s'^{(i)}|s,a)^2 - P^\top(\cdot|s,a)(V^{\text{in}})^2\right) \mathbf{1}(E_{s,a})$$

$$+ \left(\left[\frac{1}{n_{s,a}} \sum_{i=1}^{n_{s,a}} V^{\text{in}}(s'^{(i)}|s,a)\right]^2 - \left[P^\top(\cdot|s,a)V^{\text{in}}\right]^2\right) \mathbf{1}(E_{s,a})$$

By using the same conditional on $n_{s,a}$ as in Lemma D.2, applying Hoeffding's inequality and law of total expectation, we obtain with probability $1 - \delta/2$,

$$\left(\frac{1}{n_{s,a}} \sum_{i=1}^{n_{s,a}} V^{\text{in}}(s'^{(i)}|s,a)^2 - P^\top(\cdot|s,a)(V^{\text{in}})^2\right) \mathbf{1}(E_{s,a})$$

$$\leq V_{\max}^2 \sqrt{\frac{2\log(4/\delta)}{n_{s,a}}} \cdot \mathbf{1}(E_{s,a}) \leq 2V_{\max}^2 \sqrt{\frac{\log(4/\delta)}{m \cdot d^\mu(s,a)}}, \tag{39}$$

and similarly with probability $1 - \delta/2$,

$$\left(\frac{1}{n_{s,a}} \sum_{i=1}^{n_{s,a}} V^{\text{in}}(s'^{(i)}|s,a) - P^\top(\cdot|s,a)V^{\text{in}}\right) \mathbf{1}(E_{s,a}) \leq 2V_{\max} \sqrt{\frac{\log(4/\delta)}{m \cdot d^\mu(s,a)}}. \tag{40}$$

Again note for $a, b, c > 0$, if $|a - b| \leq c$, then $|a^2 - b^2| = |a - b| \cdot |a + b| \leq |a - b| \cdot (|a| + |b|) \leq |a - b| \cdot (2|b| + c) \leq c \cdot (2|b| + c) = 2bc + c^2$, therefore by (40) we have

$$\left(\left[\frac{1}{n_{s,a}} \sum_{i=1}^{n_{s,a}} V^{\text{in}}(s^{(i)}|s,a)\right]^2 - \left[P^\top(\cdot|s,a)V^{\text{in}}\right]^2\right) \mathbf{1}(E_{s,a})$$

$$\leq 4P^\top(\cdot|s,a)V^{\text{in}} \cdot V_{\max} \sqrt{\frac{\log(4/\delta)}{m \cdot d^\mu(s,a)}} + \frac{4V_{\max}^2 \log(4/\delta)}{m \cdot d^\mu(s,a)} \tag{41}$$

$$\leq 4V_{\max}^2 \sqrt{\frac{\log(4/\delta)}{m \cdot d^\mu(s,a)}} + \frac{4V_{\max}^2 \log(4/\delta)}{m \cdot d^\mu(s,a)}$$

where the last inequality comes from $|P^\top(\cdot|s,a)V^{\text{in}}| \le ||P(\cdot|s,a)||_1||V^{\text{in}}||_\infty \le V_{\max}$. Combining (39), (41) and a union bound, we have with probability $1 - \delta$,

$$|\tilde{\sigma}_{V^{\text{in}}}(s,a) - \sigma_{V^{\text{in}}}(s,a)| \le 6V_{\max}^2\sqrt{\frac{\log(4/\delta)}{m \cdot d^\mu(s,a)}} + \frac{4V_{\max}^2\log(4/\delta)}{m \cdot d^\mu(s,a)},$$

apply again the union bound over $s, a$ gives the desired result.

∎

**Lemma D.4.** *Fix $i \in [R]$. Let $g^{(i)}$ be the estimator in (36) in Algorithm 3. Then if $||V^{(i)} - V^{in}||_\infty \le 2u^{in}$, then with probability $1 - \delta/R$,*

$$\mathbf{0} \le \boldsymbol{P}[V^{(i)} - V^{in}] - g^{(i)} \le 8u^{in}\sqrt{\frac{\log(2RSA/\delta)}{ld^\mu}}$$

*Proof.* Recall $g^{(i)}, d^\mu$ are vectors. By definition of $g^{(i)}(s,a)$, applying Hoeffding's inequality we obtain with probability $1 - \delta/R$,

$$g^{(i)}(s,a) + f(s,a) - P^\top(\cdot|s,a)[V^{(i)} - V^{\text{in}}]$$

$$= \left(\frac{1}{n'_{s,a}}\sum_{j=1}^{l}\left[V^{(i)}(s'^{(j)}|s,a) - V^{\text{in}}(s'^{(j)}|s,a)\right] - P^\top(\cdot|s,a)[V^{(i)} - V^{\text{in}}]\right) \cdot \mathbf{1}(E_{s,a})$$

$$\le \left(||V^{(i)} - V^{\text{in}}||_\infty\sqrt{\frac{2\log(2R/\delta)}{n'_{s,a}}}\right) \cdot \mathbf{1}(E_{s,a})$$

$$\le ||V^{(i)} - V^{\text{in}}||_\infty\sqrt{\frac{4\log(2R/\delta)}{l \cdot d^\mu(s,a)}}$$

Now use assumption $||V^{(i)} - V^{\text{in}}||_\infty \le 2u^{\text{in}}$ and a union bound over $s, a$, we have with probability $1 - \delta/R$,

$$\left|g^{(i)} + f - \boldsymbol{P}[V^{(i)} - V^{\text{in}}]\right| \le 4u^{\text{in}}\sqrt{\frac{\log(2RSA/\delta)}{ld^\mu}} \tag{42}$$

use $f = 4u^{\text{in}}\sqrt{\log(2RSA/\delta)/ld^\mu}$, we obtain the stated result. ∎

## D.1 Iterative update analysis for infinite horizon discounted setting

The goal of iterative update is to obtain the recursive relation: $Q^\star - Q^{(i)} \le \gamma\boldsymbol{P}^{\pi^\star}[Q^\star - Q^{(i-1)}] + \xi$.

**Lemma D.5.** *Let $Q^\star$ be the optimal Q-value satisfying $Q^\star = r + \gamma\boldsymbol{P}V^\star$ and $\pi^\star$ is one optimal policy satisfying Assumption 2.1. Let $\pi$ and $V_t$ be the **Return** of inner loop in Algorithm 3. We have with probability $1 - \delta$, for all $i \in [R]$,*

$$V^{in} \le V^{(i)} \le \mathcal{T}_{\pi^{(i)}}V^{(i)} \le V^\star, \quad Q^{(i)} \le r + \gamma\boldsymbol{P}V^{(i)}, \quad \text{and} \quad Q^\star - Q^{(i)} \le \gamma\boldsymbol{P}^{\pi^\star}[Q^\star - Q^{(i-1)}] + \xi,$$

*where*

$$\xi \le 8u^{in}\sqrt{\frac{\log(2RSA/\delta)}{ld^\mu}} + \sqrt{\frac{16 \cdot \sigma_{V^\star} \cdot \log(4SA/\delta)}{m \cdot d^\mu}} + \sqrt{\frac{16 \cdot \log(4SA/\delta)}{m \cdot d^\mu}} \cdot u^{in}$$

$$+ V_{\max}\left[8\sqrt{6} \cdot \left(\frac{\log(16SA/\delta)}{m \cdot d^\mu}\right)^{3/4} + \frac{56\log(16SA/\delta)}{3m \cdot d^\mu}\right].$$

*Here $\boldsymbol{P}^{\pi^\star} \in \mathbb{R}^{S \cdot A \times S \cdot A}$ with $\boldsymbol{P}^{\pi^\star}_{(s,a),(s',a')} = d^{\pi^\star}(s',a'|s,a)$.*

*Proof.* **Step1:** For any $a, b \ge 0$, we have the basic inequality $\sqrt{a + b} \le \sqrt{a} + \sqrt{b}$, and apply to Lemma D.3 we have with probability $1 - \delta/4$,

$$\sqrt{|\tilde{\sigma}_{V^{\text{in}}} - \sigma_{V^{\text{in}}}|} \le V_{\max} \cdot \left(\frac{36\log(16SA/\delta)}{m \cdot d^\mu}\right)^{1/4} + 2V_{\max} \cdot \sqrt{\frac{\log(16SA/\delta)}{m \cdot d^\mu}}. \tag{43}$$

Next, similarly for any $a, b \geq 0$, we have $\sqrt{a} \leq \sqrt{|a-b|} + \sqrt{b}$, conditional on above then apply to Lemma D.2 (with probability $1 - \delta/4$) and we obtain with probability $1 - \delta/2$,

$$
\left|\tilde{z} - \boldsymbol{P}V^{\mathrm{in}}\right|
$$
$$
\leq \sqrt{\frac{4 \cdot \sigma_{V^{\mathrm{in}}} \cdot \log(4SA/\delta)}{m \cdot d^{\mu}}} + \frac{4V_{\max}}{3m \cdot d^{\mu}} \log(4SA/\delta)
$$
$$
\leq \left(\sqrt{\tilde{\sigma}_{V^{\mathrm{in}}}} + \sqrt{|\tilde{\sigma}_{V^{\mathrm{in}}} - \sigma_{V^{\mathrm{in}}}|}\right) \sqrt{\frac{4 \cdot \log(4SA/\delta)}{m \cdot d^{\mu}}} + \frac{4V_{\max}}{3m \cdot d^{\mu}} \log(4SA/\delta)
$$
$$
= \sqrt{\frac{4 \cdot \tilde{\sigma}_{V^{\mathrm{in}}} \cdot \log(4SA/\delta)}{m \cdot d^{\mu}}} + \left(\sqrt{|\tilde{\sigma}_{V^{\mathrm{in}}} - \sigma_{V^{\mathrm{in}}}|}\right) \sqrt{\frac{4 \cdot \log(4SA/\delta)}{m \cdot d^{\mu}}} + \frac{4V_{\max}}{3m \cdot d^{\mu}} \log(4SA/\delta)
$$
$$
\leq \sqrt{\frac{4 \cdot \tilde{\sigma}_{V^{\mathrm{in}}} \cdot \log(4SA/\delta)}{m \cdot d^{\mu}}} + 2\sqrt{6} \cdot V_{\max} \cdot \left(\frac{\log(16SA/\delta)}{m \cdot d^{\mu}}\right)^{3/4} + \frac{16V_{\max}}{3m \cdot d^{\mu}} \log(16SA/\delta).
$$

Since $e = \sqrt{4 \cdot \tilde{\sigma}_{V^{\mathrm{in}}} \cdot \log(4SA/\delta)/(m \cdot d^{\mu})} + 2\sqrt{6} \cdot V_{\max} \cdot (\log(16SA/\delta)/(m \cdot d^{\mu}))^{3/4} + 16V_{\max} \log(16SA/\delta)/(3m \cdot d^{\mu})$, from above we have

$$
z = \tilde{z} - e \leq \boldsymbol{P}V^{\mathrm{in}}, \tag{44}
$$

and

$$
z \geq \boldsymbol{P}V^{\mathrm{in}} - 2e. \tag{45}
$$

Next note $\sqrt{\sigma_{(\cdot)}}$ is a norm, so by norm triangle inequality (for the second inequality) and $\sqrt{a} \leq \sqrt{b} + \sqrt{|b-a|}$ with (43) (for the first inequality) we have

$$
\sqrt{\tilde{\sigma}_{V^{\mathrm{in}}}} \leq \sqrt{\sigma_{V^{\mathrm{in}}}} + V_{\max}\left[\left(\frac{36 \log(16SA/\delta)}{m \cdot d^{\mu}}\right)^{1/4} + \sqrt{\frac{4 \log(16SA/\delta)}{m \cdot d^{\mu}}}\right]
$$
$$
\leq \sqrt{\sigma_{V^{\star}}} + \sqrt{\sigma_{V^{\star} - V^{\mathrm{in}}}} + V_{\max}\left[\left(\frac{36 \log(16SA/\delta)}{m \cdot d^{\mu}}\right)^{1/4} + \sqrt{\frac{4 \log(16SA/\delta)}{m \cdot d^{\mu}}}\right]
$$
$$
\leq \sqrt{\sigma_{V^{\star}}} + \sqrt{\boldsymbol{P}(V^{\star} - V^{\mathrm{in}})^2} + V_{\max}\left[\left(\frac{36 \log(16SA/\delta)}{m \cdot d^{\mu}}\right)^{1/4} + \sqrt{\frac{4 \log(16SA/\delta)}{m \cdot d^{\mu}}}\right]
$$
$$
\leq \sqrt{\sigma_{V^{\star}}} + ||V^{\star} - V^{\mathrm{in}}||_{\infty} \cdot \mathbf{1} + V_{\max}\left[\left(\frac{36 \log(16SA/\delta)}{m \cdot d^{\mu}}\right)^{1/4} + \sqrt{\frac{4 \log(16SA/\delta)}{m \cdot d^{\mu}}}\right]
$$
$$
\leq \sqrt{\sigma_{V^{\star}}} + u^{\mathrm{in}} \cdot \mathbf{1} + V_{\max}\left[\left(\frac{36 \log(16SA/\delta)}{m \cdot d^{\mu}}\right)^{1/4} + \sqrt{\frac{4 \log(16SA/\delta)}{m \cdot d^{\mu}}}\right]
$$

Plug this back to (45) we get

$$
\begin{aligned}
z \geq &\boldsymbol{P}V^{\mathrm{in}} - \sqrt{\frac{16 \cdot \sigma_{V^{\star}} \cdot \log(4SA/\delta)}{m \cdot d^{\mu}}} - \sqrt{\frac{16 \cdot \log(4SA/\delta)}{m \cdot d^{\mu}}} \cdot u^{\mathrm{in}} \\
&- V_{\max}\left[8\sqrt{6} \cdot \left(\frac{\log(16SA/\delta)}{m \cdot d^{\mu}}\right)^{3/4} + \frac{56 \log(16SA/\delta)}{3m \cdot d^{\mu}}\right].
\end{aligned} \tag{46}
$$

To sum up, so far we have shown that (44), (46) hold with probability $1 - \delta/2$ and we condition on that.

**Step2:** Next we prove

$$
Q^{(i)} \leq r + \gamma \boldsymbol{P}V^{(i)}, \quad V^{\mathrm{in}} \leq V^{(i)} \leq V^{\star}, \quad \forall i \in [R] \tag{47}
$$

using backward induction.

First of all, $V^{(0)} = V^{\mathrm{in}}$ implies $V^{\mathrm{in}} \leq V^{(0)} \leq V^{\star}$ and $Q^{(0)} := \mathbf{0} \leq r + \gamma \boldsymbol{P}V^{(0)}$ so the results hold for the base case.

Now for certain $i$, using induction assumption we can assume with probability at least $1 - (i-1)\delta/R$, for all $i' = 0, ..., i-1$,

$$Q^{(i')} \le r + \gamma \boldsymbol{P}V^{(i')} \qquad V^{\text{in}} \le V^{(i')} \le V^{\star} \tag{48}$$

In particular, since $V^{\text{in}} \le V^{\star} \le V^{\text{in}} + u^{\text{in}}\boldsymbol{1}$, so combine this and (48) for $i' = i - 1$ we get

$$V^{\star} - V^{(i-1)} \le V^{\star} - V^{\text{in}} \le u\boldsymbol{1}.$$

By Lemma D.4, with probability $1 - \delta/R$,

$$\boldsymbol{P}[V^{(i)} - V^{\text{in}}] - 8u^{\text{in}}\sqrt{\frac{\log(2RSA/\delta)}{ld^{\mu}}} \le g^{(i)} \le \boldsymbol{P}[V^{(i)} - V^{\text{in}}]. \tag{49}$$

By the right hand side of this and (44) we acquire with probability $1 - i\delta/R$,

$$Q^{(i)} = r + \gamma z + \gamma g^{(i)} \le r + \gamma \boldsymbol{P}V^{\text{in}} + \gamma \boldsymbol{P}[V^{(i)} - V^{\text{in}}] = r + \gamma \boldsymbol{P}V^{(i)}$$

where the second equality already gives the proof of the first part of claim (47). Moreover, by induction assumption $V^{(i-1)} \le V^{\star}$ we have

$$Q^{(i-1)} \le r + \gamma \boldsymbol{P}V^{(i-1)} \le r + \gamma \boldsymbol{P}V^{\star} = Q^{\star},$$

which implies $V_{Q^{(i-1)}} \le V_{Q^{\star}} = V^{\star}$, therefore we have

$$V^{(i)} = \max(V_{Q^{(i-1)}}, V^{(i-1)}) \le V_t^{\star},$$

this completes the proof of the second part of claim (47).

**Step3:** Next we prove $V^{(i)} \le \mathcal{T}_{\pi^{(i)}} V^{(i)}$.

For a particular $s$, on one hand, if $\pi^{(i)}(s) = \text{argmax}_a Q^{(i-1)}(s, a)$, by $Q^{(i-1)} \le r + \gamma \boldsymbol{P}V^{(i-1)}$ we have in this case:

$$V^{(i)}(s) = \max_a Q^{(i-1)}(s, a) = Q^{(i-1)}(s, \pi^{(i)}(s)) \le r(s, \pi^{(i)}(s)) + \gamma P^{\top}(\cdot|s, \pi^{(i)}(s))V^{(i-1)}$$

$$\le r(s, \pi^{(i)}(s)) + \gamma P^{\top}(\cdot|s, \pi^{(i)}(s))V^{(i)} = (\mathcal{T}_{\pi^{(i)}} V^{(i)})(s),$$

where the first equal sign comes from the definition of $V^{(i)}$ when $V_{Q^{(i-1)}}(s) \ge V^{\text{in}}(s)$ and the first inequality is from Step2.

On the other hand, if $\pi^{(i)}(s) = \pi^{(i-1)}(s)$, then

$$V^{(i)}(s) = V^{(i-1)}(s) \le (\mathcal{T}_{\pi^{(i-1)}} V^{(i-1)})(s) \le (\mathcal{T}_{\pi^{(i-1)}} V^{(i)})(s) = (\mathcal{T}_{\pi^{(i)}} V^{(i)})(s).$$

**Step4:** It remains to check $Q^{\star} - Q^{(i)} \le \gamma \boldsymbol{P}^{\pi^{\star}}[Q^{\star} - Q^{(i-1)}] + \xi$. Indeed, using the construction of $Q^{(i)}$, we have

$$Q^{\star} - Q^{(i)} = Q^{\star} - r - \gamma z - \gamma g^{(i)} = \gamma \boldsymbol{P}V^{\star} - \gamma z - \gamma g^{(i)}$$
$$= \gamma[\boldsymbol{P}V^{\star} - \boldsymbol{P}(V^{(i)} - V^{\text{in}}) - \boldsymbol{P}V^{\text{in}}] + \xi = \gamma \boldsymbol{P}V^{\star} - \gamma \boldsymbol{P}V^{(i)} + \xi, \tag{50}$$

where the second equation uses Bellman optimality equation and the third equation uses the definition of $\xi = \gamma[\boldsymbol{P}(V^{(i)} - V^{\text{in}}) - g^{(i)} + \boldsymbol{P}V^{\text{in}} - z]$. By (46) and (49),

$$\xi \le 8u^{\text{in}}\sqrt{\frac{\log(2RSA/\delta)}{ld^{\mu}}} + \sqrt{\frac{16 \cdot \sigma_{V^{\star}} \cdot \log(4SA/\delta)}{m \cdot d^{\mu}}} + \sqrt{\frac{16 \cdot \log(4SA/\delta)}{m \cdot d^{\mu}}} \cdot u^{\text{in}}$$
$$+ V_{\max}\left[8\sqrt{6} \cdot \left(\frac{\log(16SA/\delta)}{m \cdot d^{\mu}}\right)^{3/4} + \frac{56\log(16SA/\delta)}{3m \cdot d^{\mu}}\right].$$

Lastly, note $\boldsymbol{P}V^{\star} = \boldsymbol{P}^{\pi^{\star}}Q^{\star}$ and from $V^{(i)} \ge V_{Q^{(i-1)}}$, we have $\boldsymbol{P}V^{(i)} \ge \boldsymbol{P}V_{Q^{(i-1)}} = \boldsymbol{P}^{\pi_{Q^{i-1}}}Q^{(i-1)} \ge \boldsymbol{P}^{\pi^{\star}}Q^{(i-1)}$, the last inequality holds true since $\pi_{Q^{(i-1)}}$ is the greedy policy over $Q^{(i-1)}$. Therefore (50) becomes $Q^{\star} - Q^{(i)} = \gamma \boldsymbol{P}V^{\star} - \gamma \boldsymbol{P}V^{(i)} + \xi \le \gamma \boldsymbol{P}^{\pi^{\star}}Q^{\star} - \gamma \boldsymbol{P}^{\pi^{\star}}Q^{(i-1)} + \xi$. This completes the proof.

$\blacksquare$

**Lemma D.6.** *Suppose the input $V^{in}$ of Algorithm 3 satisfies $V^{in} \leq \mathcal{T}_{\pi^{in}} V^{in}$ and $V^{in} \leq V^\star \leq V^{in} + u^{in}\mathbf{1}$. Let $V^{out}$, $\pi^{out}$ be the return of inner loop of Algorithm 3 and choose $m = l^{(i)} := m' \cdot \log(16RSA)/(u^{in})^2$, where $m'$ is a parameter will be decided later. Then in addition to the results of Lemma D.5, we have with probability $1 - \delta$,*

- *if $u^{in} \in [\sqrt{1/(1-\gamma)}, 1/(1-\gamma)]$, then:*

$$\mathbf{0} \leq V^\star - V^{out} \leq$$

$$\leq \left( \frac{12/(1-\gamma)^2}{\sqrt{m'}} \left\| \boldsymbol{d}_t^{\pi^\star} \sqrt{\frac{1}{d^\mu}} \right\|_\infty + \frac{4}{\sqrt{m'}} \left\| \sum_{t=0}^\infty \gamma^t \boldsymbol{d}_t^{\pi^\star} \sqrt{\frac{\sigma_{V^\star}}{d^\mu}} \right\|_\infty + \frac{8\sqrt{6}(1/(1-\gamma))^{\frac{10}{4}}}{(m')^{3/4}} \left\| \boldsymbol{d}_t^{\pi^\star} \left[ \frac{1}{d^\mu} \right]^{\frac{3}{4}} \right\|_\infty \right.$$

$$\left. + \frac{56/(1-\gamma)^3}{3m'} \left\| \boldsymbol{d}_t^{\pi^\star} \frac{1}{d^\mu} \right\|_\infty \right) u^{in} \cdot \mathbf{1} + \frac{u^{in}}{4}\mathbf{1}.$$

- *if $u^{in} \leq \sqrt{1/(1-\gamma)}$, then*

$$\mathbf{0} \leq V^\star - V^{out} \leq$$

$$\leq \left( \frac{12\sqrt{(1/(1-\gamma))^3}}{\sqrt{m'}} \left\| \boldsymbol{d}_t^{\pi^\star} \sqrt{\frac{1}{d^\mu}} \right\|_\infty + \frac{4}{\sqrt{m'}} \left\| \sum_{t=0}^\infty \gamma^t \boldsymbol{d}_t^{\pi^\star} \sqrt{\frac{\sigma_{V^\star}}{d^\mu}} \right\|_\infty + \frac{8\sqrt{6}(1/(1-\gamma))^{\frac{9}{4}}}{(m')^{3/4}} \left\| \boldsymbol{d}_t^{\pi^\star} \left[ \frac{1}{d^\mu} \right]^{\frac{3}{4}} \right\|_\infty \right.$$

$$\left. + \frac{56(1/(1-\gamma))^{\frac{5}{2}}}{3m'} \left\| \boldsymbol{d}_t^{\pi^\star} \frac{1}{d^\mu} \right\|_\infty \right) u^{in} \cdot \mathbf{1} + \frac{u^{in}}{4}\mathbf{1}.$$

*where $\boldsymbol{d}_t^{\pi^\star} \in \mathbb{R}^{S \cdot A \times S \cdot A}$ is a matrix represents the multi-step transition from time $0$ to $t$, i.e. $\boldsymbol{d}_{(s,a),(s',a')}^{\pi^\star} = d_{0:t}^{\pi^\star}(s', a'|s, a)$ and recall $1/d^\mu$ is a vector. $\boldsymbol{d}_t^{\pi^\star} \frac{1}{d^\mu}$ is a matrix-vector multiplication. For a vector $d_t \in \mathbb{R}^{S \times A}$, norm $|| \cdot ||_\infty$ is defined as $||d_t||_\infty = \max_{t,s,a} d_t(s, a)$.*

*Proof.* By Lemma D.5, we have with probability $1 - \delta$, for all $t \in [H]$,

$$V^{in} \leq V^{(i)} \leq \mathcal{T}_{\pi^{(i)}} V^{(i)} \leq V^\star, \quad Q^{(i)} \leq r + \gamma \boldsymbol{P} V^{(i)}, \quad \text{and} \quad Q^\star - Q^{(i)} \leq \gamma \boldsymbol{P}^{\pi^\star}[Q^\star - Q^{(i-1)}] + \xi,$$

where

$$\xi \leq 8u^{in}\sqrt{\frac{\log(2RSA/\delta)}{ld^\mu}} + \sqrt{\frac{16 \cdot \sigma_{V^\star} \cdot \log(4SA/\delta)}{m \cdot d^\mu}} + \sqrt{\frac{16 \cdot \log(4SA/\delta)}{m \cdot d^\mu}} \cdot u^{in}$$

$$+ V_{\max}\left[ 8\sqrt{6} \cdot \left( \frac{\log(16SA/\delta)}{m \cdot d^\mu} \right)^{3/4} + \frac{56\log(16SA/\delta)}{3m \cdot d^\mu} \right].$$

Applying the recursion repeatedly, we obtain

$$Q^\star - Q^{(R)} \leq \gamma^R \boldsymbol{P}^{\pi^\star}[Q^\star - Q^{(0)}] + \sum_{i=0}^R \gamma^i \left( \boldsymbol{P}^{\pi^\star} \right)^i \xi \leq \gamma^R \boldsymbol{P}^{\pi^\star}[Q^\star - Q^{(0)}] + \sum_{i=0}^\infty \gamma^i \left( \boldsymbol{P}^{\pi^\star} \right)^i \xi$$

Note $(\boldsymbol{P}^{\pi^\star})^i \in \mathbb{R}^{S \cdot A \times S \cdot A}$ represents the multi-step transition from time $0$ to $i$, i.e. $(\boldsymbol{P}^{\pi^\star})_{(s,a),(s',a')}^i = d_i^{\pi^\star}(s', a'|s, a)$. Recall $R = \ln(4/u^{in}(1-\gamma))$, then

$$\gamma^R \boldsymbol{P}^{\pi^\star}[Q^\star - Q^{(0)}] \leq \gamma^R ||Q^\star - Q^{(0)}||_\infty \leq \gamma^R V_{\max} = \gamma^R/(1-\gamma) \leq u^{in}/4.$$

Therefore

$$Q^\star - Q^{(R)} \leq \frac{u^{in}}{4} + \sum_{t=0}^\infty \gamma^t \boldsymbol{d}_t^{\pi^\star} \xi$$

$$\leq \sum_{t=0}^\infty \gamma^t \boldsymbol{d}_t^{\pi^\star} \left( 8u^{in}\sqrt{\frac{\log(2RSA/\delta)}{ld^\mu}} + \sqrt{\frac{16 \cdot \sigma_{V^\star} \cdot \log(4SA/\delta)}{m \cdot d^\mu}} + \sqrt{\frac{16 \cdot \log(4SA/\delta)}{m \cdot d^\mu}} \cdot u^{in} \right.$$

$$\left. + V_{\max}\left[ 8\sqrt{6} \cdot \left( \frac{\log(16SA/\delta)}{m \cdot d^\mu} \right)^{3/4} + \frac{56\log(16SA/\delta)}{3m \cdot d^\mu} \right] \right) + \frac{u^{in}}{4},$$

$$\tag{51}$$

Now by our choice of $m = l^{(i)} := m' \cdot \log(16RSA/\delta)/(u^{\text{in}})^2$, then the first term of (51) is further less than

$$\leq \sum_{t=0}^{\infty} \gamma^t \boldsymbol{d}_t^{\pi^\star} \left( \frac{12u^{\text{in}}}{\sqrt{m'd^\mu}} u^{\text{in}} + \sqrt{\frac{16 \cdot \sigma_{V^\star}}{m' \cdot d^\mu}} u^{\text{in}} + V_{\max} \left[ 8\sqrt{6} \cdot \left( \frac{(u^{\text{in}})^{2/3}}{m' \cdot d^\mu} \right)^{3/4} + \frac{56u^{\text{in}}}{3m' \cdot d^\mu} \right] \cdot u^{\text{in}} \right) \tag{52}$$

**Case1.** If $u^{\text{in}} \leq \sqrt{1/(1-\gamma)}$, then (52) is less than

$$\leq \sum_{t=0}^{\infty} \gamma^t \boldsymbol{d}_t^{\pi^\star} \left( \frac{12\sqrt{1/(1-\gamma)}}{\sqrt{m'd^\mu}} + \sqrt{\frac{16 \cdot \sigma_{V^\star}}{m' \cdot d^\mu}} + V_{\max} \left[ 8\sqrt{6} \cdot \left( \frac{(1/(1-\gamma))^{1/3}}{m' \cdot d_{t'}^\mu} \right)^{3/4} + \frac{56(1/(1-\gamma))^{1/2}}{3m' \cdot d_{t'}^\mu} \right] \right) u^{\text{in}}$$

$$\leq \left( \frac{12\sqrt{(1/(1-\gamma))^3}}{\sqrt{m'}} \left\| \boldsymbol{d}_t^{\pi^\star} \sqrt{\frac{1}{d^\mu}} \right\|_\infty + \frac{4}{\sqrt{m'}} \left\| \sum_{t=0}^{\infty} \gamma^t \boldsymbol{d}_t^{\pi^\star} \sqrt{\frac{\sigma_{V^\star}}{d^\mu}} \right\|_\infty + \frac{8\sqrt{6}(1/(1-\gamma))^{\frac{9}{4}}}{(m')^{3/4}} \left\| \boldsymbol{d}_t^{\pi^\star} \left[ \frac{1}{d^\mu} \right]^{\frac{3}{4}} \right\|_\infty \right.$$

$$\left. + \frac{56(1/(1-\gamma))^{\frac{5}{2}}}{3m'} \left\| \boldsymbol{d}_t^{\pi^\star} \frac{1}{d^\mu} \right\|_\infty \right) u^{\text{in}} \cdot \mathbf{1}. \tag{53}$$

**Case2.** If $u^{\text{in}} \geq \sqrt{1/(1-\gamma)}$, then (52) is less than

$$\leq \sum_{t=0}^{\infty} \gamma^t \boldsymbol{d}_t^{\pi^\star} \left( \frac{12/(1-\gamma)}{\sqrt{m'd^\mu}} + \sqrt{\frac{16 \cdot \sigma_{V^\star}}{m' \cdot d^\mu}} + V_{\max} \left[ 8\sqrt{6} \cdot \left( \frac{(1/(1-\gamma))^{2/3}}{m' \cdot d_{t'}^\mu} \right)^{3/4} + \frac{56/(1-\gamma)}{3m' \cdot d_{t'}^\mu} \right] \right) u^{\text{in}}$$

$$\leq \left( \frac{12/(1-\gamma)^2}{\sqrt{m'}} \left\| \boldsymbol{d}_t^{\pi^\star} \sqrt{\frac{1}{d^\mu}} \right\|_\infty + \frac{4}{\sqrt{m'}} \left\| \sum_{t=0}^{\infty} \gamma^t \boldsymbol{d}_t^{\pi^\star} \sqrt{\frac{\sigma_{V^\star}}{d^\mu}} \right\|_\infty + \frac{8\sqrt{6}(1/(1-\gamma))^{\frac{10}{4}}}{(m')^{3/4}} \left\| \boldsymbol{d}_t^{\pi^\star} \left[ \frac{1}{d^\mu} \right]^{\frac{3}{4}} \right\|_\infty \right.$$

$$\left. + \frac{56/(1-\gamma)^3}{3m'} \left\| \boldsymbol{d}_t^{\pi^\star} \frac{1}{d^\mu} \right\|_\infty \right) u^{\text{in}} \cdot \mathbf{1}. \tag{54}$$

∎

Next, let us first finish the proof the Algorithm 3.

**Lemma D.7.** *For convenience, define:*

$$A_{\frac{1}{2}} = \sup_t \left\| \boldsymbol{d}_{0:t}^{\pi^\star} \sqrt{\frac{1}{d^\mu}} \right\|_\infty , \quad A_2 = \sup_t \left\| \sum_{t=0}^{\infty} \gamma^t \boldsymbol{d}_{0:t}^{\pi^\star} \sqrt{\frac{\sigma_{V^\star}}{d^\mu}} \right\|_\infty , \quad A_{\frac{3}{4}} = \sup_t \left\| \boldsymbol{d}_{0:t}^{\pi^\star} \left[ \frac{1}{d^\mu} \right]^{\frac{3}{4}} \right\|_\infty , \quad A_1 = \left\| \boldsymbol{d}_{0:t}^{\pi^\star} \frac{1}{d^\mu} \right\|_\infty .$$

*Recall $\epsilon$ is the target accuracy in the outer loop of Algorithm 3 and $R = \ln(4/\epsilon(1-\gamma))$. Then:*

- *If $u^{(0)} > \sqrt{(1-\gamma)^{-1}}$, then let $m^{(j)} = l^{(i,j)} = m' \log(16(1-\gamma)^{-1}SARK)/(u^{(i-1)})^2$, where*

$$m_1' = \max \left[ 96^2(1-\gamma)^{-4} A_{\frac{1}{2}}^2, 32^2 A_2^2, \left( 64\sqrt{6} A_{\frac{3}{4}} \right)^{\frac{4}{3}} (1-\gamma)^{-\frac{10}{3}}, \frac{448}{3}(1-\gamma)^{-3} A_1 \right],$$

$$K_1 = \log_2((1-\gamma)^{-1}/\epsilon),$$

- *If $u^{(0)} \leq \sqrt{(1-\gamma)^{-1}}$, then let $m^{(j)} = l^{(i,j)} = m' \log(16(1-\gamma)^{-1}SARK/\delta)/(u^{(j-1)})^2$, where*

$$m_2' = \max \left[ 96^2(1-\gamma)^{-3} A_{\frac{1}{2}}^2, 32^2 A_2^2, \left( 64\sqrt{6} A_{\frac{3}{4}} \right)^{\frac{4}{3}} (1-\gamma)^{-3}, \frac{448}{3}(1-\gamma)^{-5/2} A_1 \right],$$

$$K_2 = \log_2(\sqrt{(1-\gamma)^{-1}}/\epsilon),$$

*Algorithm 3 obeys that, with probability $1-\delta$, the output $\pi^{(K)}$ is an $\epsilon$-optimal policy, i.e. $||V_1^\star - V_1^{\pi^{(K)}}||_\infty < \epsilon$ with total sample complexity:*

$$O\left( \frac{m' \log(16(1-\gamma)^{-1}SARK/\delta)}{\epsilon^2} RK \right)$$

*for both cases. Moreover, $m'$ can be simplified as:*

- *If $u^{(0)} \leq \sqrt{(1-\gamma)^{-1}}$, then $m' \leq c(1-\gamma)^{-3}/d_m$;*
- *If $u^{(0)} > \sqrt{(1-\gamma)^{-1}}$, then $m' \leq c(1-\gamma)^{-4}/d_m$.*

*Proof.* The proof of this lemma follows the same logic as Lemma B.10. Note there is additional logarithmic factor $R$ since the Inner loop of Algorithm 3 has an extra **For** loop. Also, $A_2$ can be bounded by $O((1-\gamma)^{-3/2})$ due to the following counterpart result of Lemma G.5:

$$\sum_{t'=t}^{\infty} \gamma^{t'} \mathbb{E}_{s_{t'}, a_{t'}}^{\pi^\star} \left[ \sigma_{V^\star}(s_{t'}, a_{t'}) | s_t, a_t \right] \leq \mathrm{Var}_{\pi^\star} \left[ \sum_{t'=t}^{\infty} \gamma^{t'} r_{t'} \middle| s_t, a_t \right]$$

which reduces the dependence from $(1-\gamma)^{-3}$ to $(1-\gamma)^{-2}$. ∎

### D.2 Proof of Theorem 4.3

*Proof.* Again the proof relies on the two stages of Algorithm 4 where the first stage reduces the error to the level below $\sqrt{(1-\gamma)^{-1}}$ and the next stage decrease the error to given accuracy. Moreover, bounding the event $\{n_{s,a} \leq \frac{1}{2} m d^\mu(s,a)\}$ using Lemma G.2 is valid since data $\mathcal{D}$ is i.i.d. and $n_{s,a} = \sum_{i=1}^{m} \mathbf{1}[s^{(i)} = s, a^{(i)} = a]$ follows binomial distribution with $\mathbb{E}[n_{s,a}] = m d^\mu(s,a)$. ∎

## E   Proof of Theorem 4.2

We prove the offline learning lower bound (best policy identification in the offline regime) of $\Omega(H^2/d_m\epsilon^2)$ for stationary transition case. Our proof consists of two steps: we will first show a minimax lower bound (over all MDP instances) for learning $\epsilon$-optimal policy is $\Omega(H^2 SA/\epsilon^2)$; next we can further improve the lower bound (over problem class $\mathcal{M}_{d_m}$) for learning $\epsilon$-optimal policy to $\Omega(H^2/d_m\epsilon^2)$ by a reduction of the first result.

There are numerous literature that provide information theoretical lower bounds under different setting, *e.g.* Dann and Brunskill [2015], Jiang et al. [2017], Krishnamurthy et al. [2016], Jin et al. [2018], Sidford et al. [2018a], Domingues et al. [2020], Yin et al. [2021], Zanette [2020], Duan and Wang [2020], Wang et al. [2020], Jin et al. [2020]. However, to the best of our knowledge, Yin et al. [2021] is the only one that gives the lower bound for explicit parameter dependence in offline case. Concretely, their lower bound $\Omega(H^3/d_m\epsilon^2)$ (for non-stationary setting) includes $d_m$ which is an inherent measure of offline problems. In the stationary transition setting, by a modification of their construction (which again originated from Jiang et al. [2017]) we can prove the lower bound of $\Omega(H^2/d_m\epsilon^2)$.

### E.1 Information theoretical lower sample complexity bound over all MDP instances for identifying $\epsilon$-optimal policy.

**Theorem E.1.** *Given $H \geq 2$, $A \geq 2$, $0 < \epsilon < \frac{1}{48\sqrt{8}}$ and $S \geq c_1$ where $c_1$ is a universal constant. Then for any algorithm and any $n \leq cH^2 SA/\epsilon^2$, there exists a non-stationary $H$ horizon MDP with probability at least $p$, the algorithm outputs a policy $\widehat{\pi}$ with $v^\star - v^{\widehat{\pi}} \geq \epsilon$.*

The proof relies on embedding $\Theta(S)$ independent multi-arm bandit problems into a family of hard-to-learn MDP instances so that any algorithm that wants to output a near-optimal policy needs to identify the best action in $\Omega(S)$ problems. By standard multi-arm bandit identification result Lemma G.1 we need $O(SA)$ episodes. To recover the $H^2$ factor, we only assign reward 1 to "good" states in the latter half of the MDP and all other states have reward 0.

*Proof of Theorem E.1.* We construct a non-stationary MDP with $S$ states per level, $A$ actions per state and has horizon $2H$. States are categorized into three types with two special states $g$, $b$ and the remaining $S - 2$ "bandit" states denoted by $s_i$, $i \in [S - 2]$. Each bandit state has an unknown best action $a_i^\star$ that provides the highest expected reward comparing to other actions.

The transition dynamics are defined as follows:

- for $h = 1, ..., 2H - 1$,

  - For bandit states $b_i$, there is probability $1 - \frac{1}{H}$ to transition back to itself ($b_i$) regardless of the action chosen. For the rest of $\frac{1}{H}$ probability, optimal action $a_i^\star$ have probability $\frac{1}{2} + \tau$ or $\frac{1}{2} - \tau$ transition to $g$ or $b$ respectively and all other actions $a$ will have equal probability $\frac{1}{2}$ for either $g$ or $b$, where $\tau$ is a parameter will be decided later. Or equivalently,

$$\mathbb{P}(\cdot|s_i, a_i^\star) = \begin{cases} 1 - \frac{1}{H} & \text{if } \cdot = s_i \\ (\frac{1}{2} + \tau) \cdot \frac{1}{H} & \text{if } \cdot = g \\ (\frac{1}{2} - \tau) \cdot \frac{1}{H} & \text{if } \cdot = b \end{cases} \qquad \mathbb{P}(\cdot|s_i, a) = \begin{cases} 1 - \frac{1}{H} & \text{if } \cdot = s_i \\ \frac{1}{2} \cdot \frac{1}{H} & \text{if } \cdot = g \\ \frac{1}{2} \cdot \frac{1}{H} & \text{if } \cdot = b \end{cases}$$

  - $g$ always transitions to $g$ and $b$ always transitions to $b$, *i.e.* for all $a \in \mathcal{A}$,

$$\mathbb{P}(g|g, a) = 1, \quad \mathbb{P}(b|b, a) = 1.$$

  We will determine parameter $\tau$ at the end of the proof.

- Reward assignment: the instantaneous reward is 1 if and only if state $s = g$ and the current time $t \in \{H, \dots, 2H - 1\}$. In all other cases, the reward is 0. *i.e.*,

$$\begin{cases} r(s_t, a) = 1 \ \ iff \ \ s_t = g \ \ and \ \ t \geq H, \\ r(s_t, a) = 0 \ \ o.w. \end{cases}$$

- The initial distribution is decided by:

$$\mathbb{P}(s_i) = \frac{1}{S}, \ \forall i \in [S - 2], \ \mathbb{P}(g) = \frac{1}{S}, \ \ \mathbb{P}(b) = \frac{1}{S} \tag{55}$$

By this construction the optimal policy must take $a_i^\star$ for each bandit state $s_i$ for at least the first half of the MDP (when $t \leq H$). In other words, this construction embeds $(S - 2)$ independent best arm identification problems that are identical to the stochastic multi-arm bandit problem in Lemma G.1 into the MDP for the following two reasons: **1.** the transition is stationary (the optimal arm $a_i^\star$ for state $s_i$ is identical across all time $t$) so instead of $H(S - 2)$ (for non-stationary case) MAB problems we only have $S - 2$ of them; **2.** all $S - 2$ problems are independent since each state $s_i$ can only transition to themselves or $g, b$.

Notice for any time $h$ with $h \leq H$, any bandit state $s_i$, the difference of the expected reward between optimal action $a_i^\star$ and other actions is:

$$\begin{aligned}
& (\frac{1}{2} + \tau) \cdot \frac{1}{H} \cdot \mathbb{E}[r_{(h+1):2H}|g] + (\frac{1}{2} - \tau) \cdot \frac{1}{H} \cdot \mathbb{E}[r_{(h+1):2H}|b] + (1 - \frac{1}{H}) \cdot \mathbb{E}[r_{(h+1):2H}|s_i] \\
& - \frac{1}{2H} \cdot \mathbb{E}[r_{(h+1):2H}|g] - \frac{1}{2H} \cdot \mathbb{E}[r_{(h+1):2H}|b] - (1 - \frac{1}{H}) \cdot \mathbb{E}[r_{(h+1):2H}|s_i] \\
=& (\frac{1}{2} + \tau) \cdot \frac{1}{H} \cdot \mathbb{E}[r_{(h+1):2H}|g] + (\frac{1}{2} - \tau) \cdot \frac{1}{H} \cdot \mathbb{E}[r_{(h+1):2H}|b] \\
& - \frac{1}{2H} \cdot \mathbb{E}[r_{(h+1):2H}|g] - \frac{1}{2H} \cdot \mathbb{E}[r_{(h+1):2H}|b] \\
=& (\frac{1}{2} + \tau) \frac{1}{H} \cdot H + (\frac{1}{2} - \tau) \frac{1}{H} \cdot 0 - \frac{1}{2H} \cdot H + \frac{1}{2H} \cdot 0 = \tau
\end{aligned} \tag{56}$$

so it seems by Lemma G.1 one suffices to use the least possible $\frac{A}{72(\tau)^2}$ samples to identify the best action $a_i^\star$. However, note observing $\sum_{t=1}^{2H} r_t = H$ is equivalent as observing $\sum_{t=1}^{H} r_t = 1$ (since $\sum_{t=1}^{H} r_t = 1$ is equivalent to $s_H = g$ and is equivalent to $\sum_{t=1}^{H} r_t = 1$). Therefore, for the bandit states in the first half the samples that provide information for identifying the best arm is up to time $H$. Or in other words, identify best arm in stationary transition setting can be decided in each single stage after $t \geq H$. As a result, the difference of the expected reward between optimal action $a_{h,i}^\star$ and other action for identifying the best arm should be corrected as:

$$(\frac{1}{2} + \tau) \cdot \frac{1}{H} \cdot \mathbb{E}[r_{(h+1):H}|g] + (\frac{1}{2} - \tau) \cdot \frac{1}{H} \cdot \mathbb{E}[r_{(h+1):H}|b] + (1 - \frac{1}{H}) \cdot \mathbb{E}[r_{(h+1):H}|s_i]$$

$$- \frac{1}{2H} \cdot \mathbb{E}[r_{(h+1):H}|g] - \frac{1}{2H} \cdot \mathbb{E}[r_{(h+1):H}|b] - (1 - \frac{1}{H}) \cdot \mathbb{E}[r_{(h+1):H}|s_i]$$

$$= (\frac{1}{2} + \tau)\frac{1}{H} \cdot 1 + (\frac{1}{2} - \tau)\frac{1}{H} \cdot 0 - \frac{1}{2H} \cdot 1 + \frac{1}{2H} \cdot 0 = \frac{\tau}{H}$$

or one can compute any bandit state in latter half ($h \geq H$):

$$(\frac{1}{2} + \tau) \cdot \frac{1}{H} \cdot \mathbb{E}[r_{h:h+1}|g] + (\frac{1}{2} - \tau) \cdot \frac{1}{H} \cdot \mathbb{E}[r_{h:h+1}|b] + (1 - \frac{1}{H}) \cdot \mathbb{E}[r_{h:h+1}|s_i]$$

$$- \frac{1}{2H} \cdot \mathbb{E}[r_{h:h+1}|g] - \frac{1}{2H} \cdot \mathbb{E}[r_{h:h+1}|g] - (1 - \frac{1}{H}) \cdot \mathbb{E}[r_{h:h+1}|s_i]$$

$$= (\frac{1}{2} + \tau)\frac{1}{H} \cdot 1 + (\frac{1}{2} - \tau)\frac{1}{H} \cdot 0 - \frac{1}{2H} \cdot 1 + \frac{1}{2H} \cdot 0 = \frac{\tau}{H},$$

which yields the same result. Now by Lemma G.1, unless $\frac{A}{72(\tau/H)^2}$ samples are collected from that bandit state, the learning algorithm fails to identify the optimal action $a_i^\star$ with probability at least $1/3$.

After running any algorithm, let $C$ be the set of bandit states for which the algorithm identifies the correct action. Let $D$ be the set of bandit states for which the algorithm collects fewer than $\frac{A}{72(\tau/H)^2}$ samples. Then by Lemma G.1 we have

$$\mathbb{E}[|C|] = \mathbb{E}\left[\sum_i \mathbb{1}[a_i = a_i^\star]\right] \leq (S - 2) - |D| + \mathbb{E}\left[\sum_{i \in D} \mathbb{1}[a_i = a_i^\star]\right]$$

$$\leq ((S - 2) - |D|) + \frac{2}{3}|D| = (S - 2) - \frac{1}{3}|D|.$$

If we have $n \leq \frac{(S-2)}{2} \times \frac{A}{72(\tau/H)^2}$, by pigeonhole principle the algorithm can collect $\frac{A}{72(\tau/H)^2}$ samples for at most half of the bandit problems, *i.e.* $|D| \geq (S - 2)/2$. Therefore we have

$$\mathbb{E}[|C|] \leq (S - 2) - \frac{1}{3}|D| \leq \frac{5}{6}(S - 2).$$

Then by Markov inequality

$$\mathbb{P}\left[|C| \geq \frac{11}{12}(S - 2)\right] \leq \frac{5/6}{11/12} = \frac{10}{11}$$

so the algorithm failed to identify the optimal action on 1/12 fraction of the bandit problems with probability at least $1/11$. Note for each failure in identification, the reward is differ by at least $\tau$ in terms of the value for $\hat{v}^\pi$ (see (56)), therefore under the event $\{|C'| \geq \frac{1}{12}(S - 2)\}$, the suboptimality of the policy produced by the algorithm is

$$\epsilon := v^\star - v^{\widehat{\pi}} = \mathbb{P}[\text{visit } C'] \times \tau + \mathbb{P}[\text{visit } C] \times 0 \geq \mathbb{P}[\bigcup_{i \in C'} \text{visit}(i)] \times \tau$$

$$= \sum_{i \in C'} \mathbb{P}[\text{visit}(i)] \times \tau = \sum_{i \in C'} \frac{1}{S}\tau = \frac{S - 2}{S}\tau := c_1\tau \qquad (57)$$

where the third equal sign uses all best arm identification problems are independent. Now we set $\tau = \min(\sqrt{1/8}, \epsilon/c_1)$ and under $n \leq cH^2SA/\epsilon^2$, we have

$$n \leq cH^2SA/\epsilon^2 \leq c'H^2SA/\tau^2 = c'72S \cdot \frac{A}{72(\tau/H)^2} := c''S \cdot \frac{A}{72(\tau/H)^2} \leq \frac{S - 2}{2} \cdot \frac{A}{72(\tau/H)^2},$$

the last inequality holds as long as $S \geq 2/(1 - 2c'')$. Therefore in this situation, with probability at least $1/11$, $v^\star - v^{\widehat{\pi}} \geq \epsilon$. Finally, we can use scaling to reduce the horizon from $2H$ to $H$. ∎

**Remark E.2.** *The suboptimality gap calculation* (57) *does not use the construction that each $s_i$ has $1 - \frac{1}{H}$ probability going back to itself so if we only need Theorem E.1 then one can assign all the probability to just $g$ or $b$, which reduces to the construction of Theorem 2 in Dann and Brunskill [2015]. However, our construction is essential for proving the following offline lower bound.*

### E.2 Information theoretical lower sample complexity bound over problems in $\mathcal{M}_{d_m}$ for identifying $\epsilon$-optimal policy.

For all $0 < d_m \leq \frac{1}{SA}$, let the class of problems be

$$\mathcal{M}_{d_m} := \left\{ (\mu, M) \mid \min_{t, s_t, a_t} d_t^\mu(s_t, a_t) \geq d_m \right\}.$$

**Theorem E.3** (Restate Theorem 4.2). *Under the condition of Theorem E.1. In addition assume $0 < d_m \leq \frac{1}{SA}$. There exists another universal constant $c$ such that when $n \leq cH^2/d_m\epsilon^2$, we always have*

$$\inf_{v^{\pi_{alg}}} \sup_{(\mu, M) \in \mathcal{M}_{d_m}} \mathbb{P}_{\mu, M} \left( v^* - v^{\pi_{alg}} \geq \epsilon \right) \geq p.$$

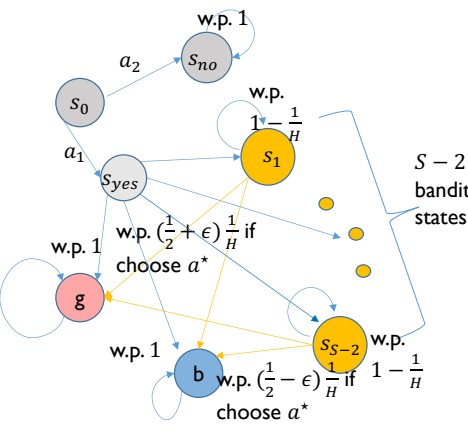

Figure 2: An illustration of transition diagram for Theorem E.3

*Proof.* The proof is mostly identical to Yin et al. [2021] except we concatenate all state together to ensure transition is stationary. The hard instances $(\mu, M)$ we used rely on Theorem E.1 as follow:

- for the MDP $M = (\mathcal{S} + 3, \mathcal{A}, r, P, d_1, 2H)$,

    - There are three extra states $s_0, s_{\text{yes}}, s_{\text{no}}$ in addition to Theorem E.1. Initial distribution $d_1$ will always enter state $s_0$, and there are two actions with action $a_1$ always transitions to $s_{\text{yes}}$ and action $a_2$ always transitions to $s_{\text{no}}$. The reward at the first time $r_1(s, a) = 0$ for any $s, a$.

    - For state $s_{\text{no}}$, it will always transition back to itself regardless of the action and receive reward 0, *i.e.*
    $$P_t(s_{\text{no}}|s_{\text{no}}, a) = 1, \ r_t(s_{\text{no}}, a) = 0, \ \forall t, \ \forall a.$$

    - For state $s_{\text{yes}}$, it will transition to the MDP construction in Theorem E.1 with horizon $2H$ and $s_{\text{yes}}$ always receives reward zero (see Figure 2).

    - For $t = 1$, choose $\mu(a_1|s_0) = \frac{1}{2}d_m SA$ and $\mu(a_2|s_0) = 1 - \frac{1}{2}d_m SA$. For all other states, choose $\mu$ to be uniform policy, *i.e.* $\mu(a_t|s_t) = 1/A$.

Based on this construction, the optimal policy has the form $\pi^\star = (a_1, \dots)$ and therefore the MDP branch that enters $s_{\text{no}}$ is uninformative. Hence, data collected by that part is uninformed about the optimal policy and there is only $\frac{1}{2}d_m SA$ proportion of data from $s_{\text{yes}}$ are useful. Moreover, by Theorem E.1 the rest of Markov chain succeeded from $s_{\text{yes}}$ requires $\Omega(H^2SA/\epsilon^2)$ episodes (regardless of the exploration strategy/logging policy), so the actual data complexity needed for the whole construction $(\mu, M)$ is $\frac{\Omega(H^2SA/\epsilon^2)}{d_m SA} = \Omega(H^2/d_m\epsilon^2)$.

It remains to check this construction $\mu$, $M$ stays within $\mathcal{M}_{d_m}$. The checking is mostly the same as Theorem G.2. in Yin et al. [2021] so we don't state here. We only highlight the checking for bandit state at different time steps. Indeed, for all $i \in [S-2]$,

$$d_{t+1}^\mu(s_i) \geq \mathbb{P}^\mu(\underbrace{s_i, s_i, \ldots, s_i}_{t \ times}, s_{\text{yes}}, s_0) = \left( \prod_{u=1}^t \mathbb{P}^\mu(s_i|s_i) \right) \mathbb{P}^\mu(s_i|s_{\text{yes}})\mathbb{P}^\mu(s_{\text{yes}}|s_0)$$

$$= (1 - \frac{1}{H})^t \left( \frac{1}{S} \right) \left( \frac{1}{2} d_m SA \right) \geq c d_m A,$$

now by $\mu$ is uniform we have $d_{t+1}^\mu(s_{t+1,i}, a) \geq \Omega(d_m A) \cdot \frac{1}{A} = \Omega(d_m)$ for all $a$. So the condition is satisfied in the stationary transition case. This concludes the proof. ∎

# F   More details for Discussion Section 5

## F.1   Proof of Lemma 5.1

*Proof.* Note data $\mathcal{D}$ comes from the logging policy $\mu$, therefore we can use extra $n(\geq 1/d_m \cdot \log(HSA/\delta))$ episodes to construct direct on-policy estimator as:

$$\widehat{d}_t^\mu = n_{s_t,a_t}/n.$$

Since $n_{s_t,a_t}$ is binomial, by the multiplicative Chernoff bound (Lemma G.2), we have

$$P\left[ n_{s_t,a_t} < \frac{1}{2} d_t^\mu(s_t, a_t) n \right] \leq e^{-\frac{d_t^\mu(s_t,a_t)\cdot n}{8}}, \qquad P\left[ n_{s_t,a_t} \geq \frac{3}{2} d_t^\mu(s_t, a_t) n \right] \leq e^{-\frac{d_t^\mu(s_t,a_t)\cdot n}{12}}.$$

this implies that for any $(s_t, a_t)$ such that $d_t^\mu(s_t, a_t) > 0$, when $n \geq 1/d_m \cdot \log(1/\delta) \geq 1/d_t^\mu(s_t, a_t) \cdot \log(1/\delta)$, we have with probability $1 - \delta$ that

$$\frac{1}{2} d_t^\mu(s_t, a_t) \leq \widehat{d}_t^\mu(s_t, a_t) \leq \frac{3}{2} d_t^\mu(s_t, a_t).$$

Applying a union bound, we have the above is true for all $(t, s_t, a_t)$ when $n \geq 1/d_m \cdot \log(HSA/\delta)$. Finally, take $\widehat{d}_m := \min_{(t,s_t,a_t):\widehat{d}_t^\mu(s_t,a_t)>0} \widehat{d}_t^\mu(s_t, a_t)$. On the above concentration event, we get

$$\frac{1}{2} d_m \leq \widehat{d}_m \leq \frac{3}{2} d_m,$$

by taking $\min$ on all sides. ∎

## F.2   On relationship between $1/d_m$ and $\beta_\mu, C$

In the function approximation regime, roughly speaking, the *concentration coefficient* assumption requires Munos [2003], Le et al. [2019], Chen and Jiang [2019], Xie and Jiang [2020b]

$$\beta_\mu = \sup_{\pi \in \mathcal{F}} \left\| \frac{d^\pi(s, a)}{d^\mu(s, a)} \right\|_\infty < \infty,$$

where $\mathcal{F}$ is the policy class induced by approximation functions. In the tabular case, since we want to maximize over all policies, $\mathcal{F} = \{all \ policies\}$, therefore above should be interpreted as:

$$\sup_{\pi \ \text{arbitrary}} \left\| \frac{d_t^\pi(s, a)}{d_t^\mu(s, a)} \right\|_\infty < \infty \Rightarrow ||d_t^\mu(s, a)||_\infty > 0,$$

since $\mathcal{F}$ is the largest possible class, if the transition kernel $P(s'|s, a)$ is able to reach some $s' \in \mathcal{S}$ given $s, a$, then that implies $d_t^\pi(s') > 0$. Next one can always pick $\pi_{t+1}(s') = a'$ such that $d_{t+1}^\pi(s', a') = d_t^\pi(s') > 0$, for all $a' \in \mathcal{A}$. This means $\mu$ has the chance to explore all states and actions whenever the transition $P$ can transition to all states (from some previous $s, a$).

On the other hand, our Assumption 2.1 only require $\mu$ to trace at least one optimal policy $\pi^\star$ and it is fine for $\mu$ to never visit certain state-action $s, a$ that is not related to $\mu$.

As a result, since $\beta_\mu$ or $C$ are explicitly incorporated, the upper bounds in Le et al. [2019], Chen and Jiang [2019], Xie and Jiang [2020b] may degenerate to $+\infty$ under our setting (Assumption 2.1), regardless of the dependence on horizon.

Nevertheless, we point out that function approximation+concentrability assumption is a powerful framework for handling realizability/agnostic case and related concepts (*e.g.* inherent Bellman error) and easier to scale the setting to general continuous case.

### F.3  Improved dependence on $(1-\gamma)^{-1}$ than prior work

The sample complexity bound $\widetilde{O}((1-\gamma)^{-3}/d_m\epsilon^2)$ in Theorem 4.3 can be compared with the line of recent works on offline RL with function approximation. For example, Le et al. [2019] consider doing batch learning based on fitted Q-iteration with constraints and in their Theorem 4.3 the sample complexity should be translated as $\tilde{O}((1-\gamma)^{-6}\beta_\mu/\epsilon^2)$, where $\beta_\mu$ is the "concentration factor" similar to $1/d_m$, but with stronger assumption that $\mu$ explores all $s, a$ that can be visited by the function approximation class. Chen and Jiang [2019], Xie and Jiang [2020b] also consider using FQI in different ways and prove $\epsilon V_{\max}$-optimal policy with sample complexity $\tilde{O}((1-\gamma)^{-4}C/\epsilon^2)$ and $\tilde{O}((1-\gamma)^{-2}C/\epsilon^2)$, where $C$ is again the "concentration-type coefficient". Their result should be translated as $\tilde{O}((1-\gamma)^{-6}C/\epsilon^2)$ and $\tilde{O}((1-\gamma)^{-4}C/\epsilon^2)$ for $\epsilon$-optimal policy.

### F.4  The doubling procedure overcomes the small defect in Sidford et al. [2018a]

Sidford et al. [2018a] first uses *variance reduction* technique to provide provable guarantee for identifying the $\epsilon$-optimal policy. However, their complexity may fail to be optimal under their initialization. In fact, in their Proof of Proposition 5.4.1. (page 23 of https://arxiv.org/pdf/1806.01492.pdf), they use the inequality

$$\left(\frac{(1-\gamma)^3 u^2}{C''(1-\gamma)^{8/3}}\right)^{3/4} = C''(1-\gamma)^{1/4}u^{3/2} \leq \frac{u}{16},$$

which is equivalent to $u \leq O(\sqrt{1/(1-\gamma)})$ (or $u \leq \sqrt{H}$) and based on their initialization $\boldsymbol{v}^{(0)} = \boldsymbol{0}$ they cannot guarantee $\|\boldsymbol{v}^{(0)}\| = \|\boldsymbol{v}^{(0)} - \boldsymbol{v}^\star\|_\infty := u \leq (1-\gamma)^{-1/2}$. Actually, $\|\boldsymbol{v}^{(0)} - \boldsymbol{v}^\star\|_\infty \leq (1-\gamma)^{-1/2}$ is never guaranteed even when choosing $\boldsymbol{v}^{(0)} = (1-\gamma)^{-1}$ (since the true $\boldsymbol{v}^\star$ could be small). Hence, to achieve the optimality Sidford et al. [2018a] essentially requires an additional stringent assumption $\|\boldsymbol{v}^{(0)} - \boldsymbol{v}^\star\|_\infty \leq (1-\gamma)^{-1/2}$, which is usually not enjoyed by practical applications since we could not observe $\boldsymbol{v}^\star$ beforehand. On the other hand, given that they consider only the sub-problem class where it always holds $\|\boldsymbol{v}^\star\| = \|0 - \boldsymbol{v}^\star\|_\infty \leq (1-\gamma)^{-1/2}$, then $(1-\gamma)^{-3}SA/\epsilon^2$ is no longer the minimax sample complexity due to the scaling (reward is scaled by $(1-\gamma)^{-1/2}$ and the sample complexity is $(1-\gamma)^{-5/2}SA/\epsilon^2$). Considering minimaxity are the worst case guarantees, this issue makes their algorithm has no improvement over the standard *simulation lemma*-based analysis or *Hoeffding-style* analysis ($\widetilde{O}((1-\gamma)^{-4}SA/\epsilon^2)$ see Agarwal et al. [2019]) in the worst case sense.

Our *Doubled Variance Reduction* addresses this issue completely so that minimaxity is preserved for the offline learning with arbitrary initialization.

## G  Technical lemmas

**Lemma G.1** (Best arm identification lower bound Krishnamurthy et al. [2016]). *For any $A \geq 2$ and $\tau \leq \sqrt{1/8}$ and any best arm identification algorithm that produces an estimate $\hat{a}$, there exists a multi-arm bandit problem for which the best arm $a^\star$ is $\tau$ better than all others, but $\mathbb{P}[\hat{a} \neq a^\star] \geq 1/3$ unless the number of samples $T$ is at least $\frac{A}{72\tau^2}$.*

**Lemma G.2** (Multiplicative Chernoff bound Chernoff et al. [1952]). *Let $X$ follows Binomial distribution, i.e. $X \sim Binom(n, p)$. For any $1 \geq \delta > 0$, we have that*

$$\mathbb{P}[Z < (1-\delta)pn] < e^{-\frac{\delta^2 pn}{2}}. \qquad and \qquad \mathbb{P}[Z \geq (1+\delta)pn] < e^{-\frac{\delta^2 pn}{3}}$$

**Lemma G.3** (Bernstein's Inequality). *Let $X_1, ..., X_n$ be independent random variables such that $\mathbb{E}[X_i] = 0$ and $|X_i| \leq C$. Let $\sigma^2 = \frac{1}{n}\sum_{i=1}^n \text{Var}[X_i]$, then we have*

$$\frac{1}{n}\sum_{i=1}^n X_i \leq \sqrt{\frac{2\sigma^2 \cdot \log(1/\delta)}{n}} + \frac{2C}{3n}\log(1/\delta)$$

*holds with probability $1 - \delta$.*

**Lemma G.4** (Freedman's inequality Tropp et al. [2011]). *Let $X$ be the martingale associated with a filter $\mathcal{F}$ (i.e. $X_i = \mathbb{E}[X|\mathcal{F}_i]$) satisfying $|X_i - X_{i-1}| \leq M$ for $i = 1, ..., n$. Denote $W := \sum_{i=1}^n \text{Var}(X_i|\mathcal{F}_{i-1}) \leq \sigma^2$ then we have*

$$\mathbb{P}(|X - \mathbb{E}[X]| \geq \epsilon) \leq 2e^{-\frac{\epsilon^2}{2(\sigma^2 + M\epsilon/3)}}.$$

*Or equivalently, with probability $1 - \delta$,*

$$|X - \mathbb{E}[X]| \leq \sqrt{8\sigma^2 \cdot \log(1/\delta)} + \frac{2M}{3} \cdot \log(1/\delta).$$

**Lemma G.5.** *Let $r_t^{(1)}, s_t^{(1)}, a_t^{(1)}$ denotes random variables. Then the following decomposition holds:*

$$\text{Var}_\pi\left[\sum_{t=h}^H r_t^{(1)} \bigg| s_h^{(1)} = s_h, a_h^{(1)} = a_h\right] = \sum_{t=h}^H \left(\mathbb{E}_\pi\left[\text{Var}\left[r_t^{(1)} + v_{t+1}^\pi(s_{t+1}^{(1)})\bigg|s_t^{(1)}, a_t^{(1)}\right]\bigg|s_h^{(1)} = s_h, a_h^{(1)} = a_h\right]\right.$$
$$\left. + \mathbb{E}_\pi\left[\text{Var}\left[\mathbb{E}[r_t^{(1)} + v_{t+1}^\pi(s_{t+1}^{(1)})|s_t^{(1)}, a_t^{(1)}]\bigg|s_t^{(1)}\right]\bigg|s_h^{(1)} = s_h, a_h^{(1)} = a_h\right]\right).$$

$$(58)$$

**Remark G.6.** *This is a conditional version of Lemma 3.4 in Yin and Wang [2020]. It can be proved using the identical trick as Lemma 3.4 in Yin and Wang [2020] except the law of total variance is replaced by the law of total conditional variance.*

## H  Summary of the fictitious estimators

To assist the readers following the design framework, we use a section to summarize the design of the fictitious estimators.

### H.1  finite horizon, non-stationary case

Now let us introduce our estimators $\mathbf{z}_t$ and $\mathbf{g}_t$ in the finite-horizon non-stationary case (the choices for the stationary case and the infinite-horizon case will be introduced later).

Given an offline dataset $\mathcal{D}$, we define LCB $z_t(s_t, a_t) = \tilde{z}_t(s_t, a_t) - e(s_t, a_t)$ where $\tilde{z}_t(s_t, a_t)$ is an unbiased estimator and $e(s_t, a_t) = O(\sqrt{\tilde{\sigma}_{V_{t+1}^{\text{in}}}(s_t, a_t)})$ is an "error bar" that depends on $\tilde{\sigma}_{V_{t+1}^{\text{in}}}(s_t, a_t)$ — an estimator of the variance of $\tilde{z}_t(s_t, a_t)$. $\tilde{z}_t(s_t, a_t)$ and $\tilde{\sigma}_{V_{t+1}^{\text{in}}}(s_t, a_t)$ are plug-in estimators at $(s_t, a_t)$ that use the available offline data $(r_t, s'_{t+1})$ to estimate the transition and rewards *only if* the number of visitations to $(s_t, a_t)$ (denoted by $n_{s_t, a_t}$) is greater than a statistical threshold. Let $m$ be the episode budget, we write:

$$\tilde{z}_t(s_t, a_t) = P_t^\top(\cdot|s_t, a_t)V_{t+1}^{\text{in}} \cdot \mathbf{1}(E_{m,t}^c) + \frac{1}{n_{s_t, a_t}}\sum_{i=1}^m V_{t+1}^{\text{in}}(s_{t+1}^{(i)}) \cdot \mathbf{1}_{[s_t^{(i)}, a_t^{(i)} = s_t, a_t]} \cdot \mathbf{1}(E_{m,t}),$$

$$\tilde{\sigma}_{V_{t+1}^{\text{in}}}(s_t, a_t) = \sigma_{V_{t+1}^{\text{in}}}(s_t, a_t)\mathbf{1}(E_{m,t}^c) + \left[\frac{1}{n_{s_t, a_t}}\sum_{i=1}^m [V_{t+1}^{\text{in}}(s_{t+1}^{(i)})]^2 \cdot \mathbf{1}_{[s_t^{(i)}, a_t^{(i)} = s_t, a_t]} - \tilde{z}_t^2(s_t, a_t)\right]\mathbf{1}(E_{m,t}),$$

$$e(s_t, a_t) = \sqrt{\frac{4\tilde{\sigma}_{V_{t+1}^{\text{in}}}\iota}{md_t^\mu(s_t, a_t)}} + 2\sqrt{6}V_{\max}\left(\frac{\iota}{md_t^\mu(s_t, a_t)}\right)^{3/4} + 16\frac{V_{\max}\iota}{md_t^\mu(s_t, a_t)}$$

$$(59)$$

where $E_{m,t} = \{n_{s_t,a_t} > \frac{1}{2}m \cdot d_t^\mu(s_t, a_t)\}$ and $n_{s_t,a_t}$ is the number of episodes visited $(s_t, a_t)$ at time $t$. We also note that we only aggregate the data at the same time step $t$, so that the observations are from different episodes and thus conditional independent given the current time step[8].

Similarly, our estimator $g_t(s_t, a_t) = \tilde{g}_t(s_t, a_t) - f(s_t, a_t)$ where $\tilde{g}_t(s_t, a_t)$ estimates $P_t^\top(\cdot|s_t, a_t)[V_{t+1} - V_{t+1}^{\text{in}}]$ using $l$ independent episodes (let $n'_{s_t,a_t}$ denote the visitation count from these $l$ episodes) and $f(s_t, a_t)$ is an error bar:

$$\tilde{g}_t(s_t, a_t) = P_t^\top(\cdot|s_t, a_t)[V_{t+1} - V_{t+1}^{\text{in}}] \cdot \mathbf{1}(E_{l,t}^c)$$
$$+ \frac{1}{n'_{s_t,a_t}} \sum_{j=1}^{l} [V_{t+1}(s_{t+1}'^{(j)}) - V_{t+1}^{\text{in}}(s_{t+1}'^{(j)})] \cdot \mathbf{1}_{[s_t'^{(j)}, a_t'^{(j)} = s_t, a_t]} \mathbf{1}(E_{l,t}) \tag{60}$$

Here, $E_{l,t} = \{n'_{s_t,a_t} > \frac{1}{2}l \cdot d_t^\mu(s_t, a_t)\}$ and $f(s_t, a_t, u) := 4u\sqrt{\iota/ld_t^\mu(s_t, a_t)}$. Notice that $f(s_t, a_t, u)$ depends on the additional input $u^{\text{in}}$ which measures the certified suboptimality of the input.

**Fictitious vs. actual estimators.** Careful readers must have noticed that that the above estimators $\mathbf{z}_t$ and $\mathbf{g}_t$ are *infeasible* to implement as they require the unobserved (population level-quantities) in some cases. We call them *fictitious* estimators as a result. Readers should rest assured since by the following proposition we can show their practical implementations (summarized in Figure 3.2) are *identical* to these *fictitious* estimators with high probability:

**Proposition H.1** (Summary of Section B.4). *Under the condition of Theorem 3.2, we have*

$$\mathbb{P}\left[\bigcup_{i \in [K_1], t \in [H]} \left(E_{l,t}^{(i)c} \cup E_{m,t}^{(i)c}\right) \bigcup_{j \in [K_2], t \in [H]} \left(E_{l,t}^{(j)c} \cup E_{m,t}^{(j)c}\right)\right] \leq \delta/2,$$

*this means with high probability $1 - \delta/2$, fictitious estimators $\tilde{z}_t, \tilde{g}_t, \tilde{\sigma}$ are all identical to their practical versions (summarized in Figure 3.2). Moreover, under the same high probability events, the empirical version of the "error bars" $e_t(s_t, a_t)$ and $f(s_t, a_t, u)$ are at most twice as large than their fictitious versions that depends on the unknown $d_t^\mu(s_t, a_t)$.*

These *fictitious* estimators, however, are easier to analyze and they are central to our extension of the Variance Reduction framework previously used in the generative model setting [Sidford et al., 2018a] to the offline setting. The idea is that it *replaces* the low-probability, but pathological cases due to random $n_{s_t,a_t}$ with ground truths. Another challenge of the offline setting is due to the dependence of data points within a single episode. Note that the estimators are only aggregating the data at the same time steps. Since the data pair at the same time must come from different episodes, then conditional independence (given data up to current time steps, the states transition to the next step are independent of each other) can be recovered by this design (59), (60).

## H.2  Finite horizion, Stationary case

Indeed, we modify the fictitious estimators (59) and (60) into the following:

$$\tilde{z}_t(s, a) = P^\top(\cdot|s, a)V_{t+1}^{\text{in}} \cdot \mathbf{1}(E_m^c) + \frac{1}{n_{s,a}}\sum_{i=1}^{m}\sum_{u=1}^{H}V_{t+1}^{\text{in}}(s_{u+1}^{(i)}) \cdot \mathbf{1}_{[s_u^{(i)}=s, a_u^{(i)}=a]}\mathbf{1}(E_m),$$

$$\tilde{\sigma}_{V_{t+1}^{\text{in}}}(s, a) = \sigma_{V_{t+1}^{\text{in}}}(s, a)\mathbf{1}(E_m^c) + [\frac{1}{n_{s,a}}\sum_{i=1}^{m}\sum_{u=1}^{H}[V_{t+1}^{\text{in}}(s_{u+1}^{(i)})]^2 \cdot \mathbf{1}_{[s_u^{(i)}=s, a_u^{(i)}=a]} - \tilde{z}_t^2(s, a)]\mathbf{1}(E_m),$$

$$e_t(s, a) = \sqrt{\frac{4\tilde{\sigma}_{V_{t+1}^{\text{in}}}\iota}{m\sum_{t=1}^{H}d_t^\mu(s, a)}} + 2\sqrt{6}V_{\max}\left(\frac{\iota}{m\sum_{t=1}^{H}d_t^\mu(s, a)}\right)^{3/4} + 16V_{\max}\frac{\iota}{m\sum_{t=1}^{H}d_t^\mu(s, a)}, \tag{61}$$

where $E_m = \{n_{s,a} > \frac{1}{2}m\cdot\sum_{t=1}^{H}d_t^\mu(s, a)\}$ and $n_{s,a} = \sum_{i=1}^{m}\sum_{t=1}^{H}\mathbf{1}[s_t^{(i)} = s, a_t^{(i)} = a]$ is the number of data pieces visited $(s, a)$ over ALL $m$ episodes. Moreover, $f_t(s, a, u) = 4u\sqrt{\frac{\iota}{l}\sum_{t=1}^{H}d_t^\mu(s, a)}$

---

[8]This is natural for the non-stationary transition setting; for the stationary transition setting we have an improved way for defining this estimators. See Section 4.

and

$$\tilde{g}_t(s,a) = P^\top(\cdot|s,a)[V_{t+1} - V_{t+1}^{\text{in}}]\mathbf{1}(E_l^c) + \frac{1}{n_{s,a}'} \sum_{j=1}^{l} \sum_{u=1}^{H} [V_{t+1}(s_{u+1}'^{(j)}) - V_{t+1}^{\text{in}}(s_{u+1}'^{(j)})] \cdot \mathbf{1}_{[s_u'^{(j)}, a_u'^{(j)} = s,a]} \mathbf{1}(E_l^c).$$

(62)

In this case, we collect all the data that enter the same state action $(s,a)$ to calculate $n_{s,a}$ across different times (compare $n_{s_t,a_t}$ in the non-stationary transition case). By this design we can further tighten the horizon dependence and improve the sample efficiency by preforming careful analysis. Note in this case the conditional independence property is wrecked.

### H.3  Infinite horizon, discounted case

An infinite-horizon discounted MDP is denoted by $(\mathcal{S}, \mathcal{A}, P, r, \gamma, d_0)$, where $\gamma$ is discount factor and $d_0$ is initial state distribution. Given a policy $\pi$, the induced trajectory $s_0, a_0, r_0, s_1, a_1, r_1, ...$ follows: $s_0 \sim d_0$, $a_t \sim \pi(\cdot|s_t)$, $r_t = r(s_t, a_t)$, $s_{t+1} \sim P(\cdot|s_t, a_t)$. The corresponding value function (or state-action value) function is defined as: $V^\pi(s) = \mathbb{E}_\pi[\sum_{t=0}^\infty \gamma^t r_t | s_0 = s]$, $Q^\pi(s) = \mathbb{E}_\pi[\sum_{t=0}^\infty \gamma^t r_t | s_0 = s, a_0 = a]$. Moreover, define the normalized marginal state distribution as $d^\pi(s) := (1 - \gamma) \sum_{t=0}^\infty \gamma^t \mathbb{P}[s_t = s | s_0 \sim d_0, \pi]$ and the state-action counterpart follows $d^\pi(s,a) := d^\pi(s)\pi(s|a)$. For the offline/batch learning problem, we adopt the same assumption of Chen and Jiang [2019], Xie and Jiang [2020b] that data $\mathcal{D} = \{s^{(i)}, a^{(i)}, r^{(i)}, s'^{(i)}\}_{i \in [n]}$ are i.i.d off-policy pieces with $(s,a) \sim d^\mu$ and $s' \sim P(\cdot|s,a)$. Under a fast-mixing assumption when rolling out with the logging policy $\mu$, such a data set that *approximately* satisfy this assumption can be constructed by taking one data point $(s,a,r,s')$ once in a while within a single trajectory.

Algorithm 1 and 2 are slighted modified to cater to the infinite horizon setting (detailed pseudo-code in Algorithm 3 and 4 in the appendix). Then the corresponding estimators are set to be:

The corresponding estimators in Algorithm 3 are defined as:

$$\tilde{z}(s,a) = \begin{cases} P^\top(\cdot|s,a)V^{\text{in}}, & if\ n_{s,a} \le \frac{1}{2}m \cdot d^\mu(s,a), \\ \frac{1}{n_{s,a}} \sum_{i=1}^{m} V^{\text{in}}(s'^{(i)}) \cdot \mathbf{1}_{[s^{(i)}=s,a^{(i)}=a]}, & if\ n_{s,a} > \frac{1}{2}m \cdot d^\mu(s,a). \end{cases}$$

$$\tilde{\sigma}_{V^{\text{in}}}(s,a) = \begin{cases} \sigma_{V^{\text{in}}}(s,a), & if\ n_{s,a} \le \frac{1}{2}m \cdot d^\mu(s,a), \\ \frac{1}{n_{s,a}} \sum_{i=1}^{m} [V^{\text{in}}(s'^{(i)})]^2 \cdot \mathbf{1}_{[s^{(i)}=s,a^{(i)}=a]} - \tilde{z}^2(s,a), & otherwise. \end{cases}$$

(63)

where $n_{s,a} := \sum_{i=1}^{n} \mathbf{1}[s^{(i)} = s, a^{(i)} = a]$ is the number of samples start at $(s,a)$. Similarly, $P^\top(\cdot|s,a)[V - V^{\text{in}}]$ is later updated using different $l$ episodes ($n_{s,a}'$ is the number count from $l$ episodes):

$$g^{(i)}(s,a) = \begin{cases} P^\top(\cdot|s,a)[V^{(i)} - V^{\text{in}}] - f(s,a), & if\ \ n_{s,a}' \le \frac{1}{2}l \cdot d^\mu(s,a), \\ \frac{1}{n_{s,a}'} \sum_{j=1}^{l} [V^{(i)}(s'^{(j)}) - V^{\text{in}}(s'^{(j)})] \cdot \mathbf{1}_{[s'^{(j)},a'^{(j)}=s,a]} - f(s,a), o.w. \end{cases}$$

(64)

where $f = 4u^{\text{in}}\sqrt{\log(2RSA/\delta)/ld^\mu}$ and $R = \ln(4/u^{\text{in}}(1 - \gamma))$.

From technical perspective, this is the easiest setting to analyze due to the i.i.d. assumption on data $\mathcal{D}$.