# OpenReview forum: "Near-Optimal Offline Reinforcement Learning via Double Variance Reduction"
_NeurIPS.cc/2021/Conference — NeurIPS 2021 Poster_

### Official Review · Reviewer_yHp3 · 2021-07-08

**Rating:** 7
**Confidence:** 2

**Summary:**

The paper focuses on the theoretical understanding of offline reinforcement learning. Assuming weak coverage (Assumption 2.1), which appears to be necessary, the authors propose a double variance deduction learning algorithm. The algorithm has two key components: variance reduction (Eq. (2)) that decompose the key estimation of P_t(\cdot|s,a)V_{t+1} into two components that are estimated separately at different stages; and doubling where parameters are instantiated twice. The authors show strong theoretical results of the proposed algorithm: (new) order-optimal sample complexity for the stationary case, order optimal results for the finite-horizon non-stationary setting and improved results for the infinite horizon discounted setting.

**Limitations And Societal Impact:**

I would like to suggest some discussions on the practical implications of the proposed algorithm.

**Main Review:**

The paper is well written. Although highly theoretical, the paper presents clear structure and intuition to facilitate its understanding.

The work is a bit outside my area of expertise. While I understood the ideas in the paper, I was not able to fully follow its analysis. The results look good to me with sufficient novelty and technical depth. However, I will yield to reviewers with better expertise in this area.

I understand that the paper is theoretical in nature and such papers often do not have numerical evaluations. Still, I wonder about the empirical performance and implications of the proposed algorithm. After all, one purpose of theoretical work is to inspire practitioners to develop useful algorithms in practice. Therefore, it would be desirable to see some numerical evaluations of the proposed algorithm, as well as the comparisons with existing algorithms.




**Time Spent Reviewing:**

3

---

> ### Author Response · Authors · 2021-08-10
> **Response for the reviewer yHp3**
>
> We appreciate the reviewer for the encouraging comments and the positive feedback! The followings are our detailed responses.
>
> ----- "theoretical work is to inspire practitioners ..." reviewer 9aNL:"why someone should care about improving sample complexity by a factor of $H$" -----
>
> Thanks! The theoretical study of sample complexity guides the empirical work in the following way: for obtaining the $\epsilon$-optimal policy, our result tells the complexity needed is of order $\approx H^2/\epsilon^2$ in the worst case scenario (since minimaxity is the worst case bound), and from the practical view, this means even for the hardest problem, the sample that needed to be collected (in principle) is of order $H^2$. Hence, this message provides one heuristic way to measure the empirical work: the empirical methods that are designed for the purpose of sample efficiency may be considered as "state-of-the-art" if it can exhibit good performances for hard practical problems with the total sample used of order $H^2$ (e.g. for a long horizon problem with $H=1000$, using $H^2=10^6$ offline episodes to solve the task is much better than using $H^3=10^9$ in terms of both computation and storage).
>
>
> ----- "numerical simulation ..." -----
>
> Thanks. We do agree a simulation study is helpful for the illustration purpose of our theoretical results and we are happy to add one as follows. To verify our OPDVR-stationary improves over the existing works in terms of scaling in $H$, we use a new stationary environment by modifying the MDP example used in [Yin et al. 2021] Section 6 (more concretely, in their Appendix I we can remove one version of the MDP to make it stationary but keep the reward assignments changing at each time step). We will compare the performance of OPDVR-stationary to OPEMA (from Yin et al.) and OPDVR-nonstationary (nonstationary algorithm can run in the stationary environment). We will follow [Yin et al. 2021] to use a fixed offline data set and test the methods on the varying $H$, but in our setting. In particular, there are universal constants in OPDVR that need to be identified (we regard them as the ``hyperparameters'' in our algorithm) and we will use cross validation (by data splitting). We will add this case study in our revision and we are happy to take any question regrading this in the later discussions.

---

> > ### Comment · Reviewer_yHp3 · 2021-09-11
> > **Re: response**
> >
> > Thanks for the clarifications.

---

### Official Review · Reviewer_YWeJ · 2021-07-16

**Rating:** 7
**Confidence:** 4

**Summary:**

This paper considers a variance-reduced Q learning algorithm for offline policy optimization, and shows that the proposed algorithm is minimax-optimal up to logarithmic factors.

**Limitations And Societal Impact:**

The authors have properly discussed them.

**Main Review:**

The authors extend the variance reduction technique from the generative model to offline scenarios, and achieve sharp error bound on finite horizon homogeneous, finite horizon inhomogeneous and infinite horizon setting. The results are technically sound. I have some minor concerns on the significance of the paper. I want to remark that, such doubling procedure have already been introduced in [1], see Section 3.4 of the corresponding paper. Meanwhile, as the authors point out, in the offline scenarios, the data are highly dependent which leads to additional technical difficulties. However, if I understand correctly, what the author do is only sampling sufficient episodes to guarantee each state-action pair has sufficient number of data with high probability, then we can follow the analysis of the generative model setting. I think the authors should make this difference between generative model setting more clear, if there are some other technical efforts to deal with this issue. Furthermore, I have a question regarding the range of $\epsilon$, as in [1], the results only hold when the number of data exceeds certain threshold, which means the result can only hold for $\epsilon \in (0, 1]$. Does the same issue come here? I think the authors better make some discussions on that.

[1] Martin J Wainwright. Variance-reduced q-learning is minimax optimal. arXiv preprint arXiv:1906.04697, 2019.

**Time Spent Reviewing:**

1

---

> ### Author Response · Authors · 2021-08-10
> **Response for the reviewer YWeJ**
>
> We appreciate the reviewer for the feedbacks and the expert understanding of the technical details. The followings are our detailed responses.
>
> ----- Q1: "want to remark that ... introduced in [1] ...", Q2:"Furthermore, I have a question regarding the range of $\epsilon$, as in [1], the results only hold when the number of data exceeds certain threshold, which means the result can only hold for $\epsilon\in(0,1]$" -------
>
> Thank you for the in-depth questions!
>
> Q1: We agree [1] also used a form of double variance reduction. However, [1] only tackles the problem of q-learning, where it learns an $\epsilon$-optimal Q function. In contrast, we can guarantee the output of an $\epsilon$-optimal policy. It is known that $\epsilon$-optimal Q functions can only yield a suboptimal policy in horizon $H$ (or effective horizon $(1-\gamma)^{-1}$) by the argmax reduction for the learned Q functions [MA, RM, HK, 2013]. Moreover, they consider the generative model setting, whose analysis is more straightforward as it assumes i.i.d for simplicity and does not deal with the sequential dependence of the data trajectories (see our second response for details).
>
> Q2: Yes and this is stated implicitly in the Proposition 3.1 (Line199), such a requirement is due to the condition $\epsilon'=\sqrt{H}\epsilon\leq \sqrt{H}$ (line 205). This is the sample barrier consideration raised for generative model setting [Gen Li et al. 2020] and we will include clear discussion of this aspect in our revision.
>
>
>
> [2] [MA, RM, HK]  Minimax PAC bounds on the sample complexity of reinforcement learning with a generative model, Machine learning, 2013
>
> [3] [Gen Li et al.] Breaking the Sample Size Barrier in Model-Based Reinforcement Learning with a Generative Model, NeurIPS, 2020.
>
> Those are excellent catches by the reviewer and we will add careful discussions in our revision.
>
>
> ----- "difference between generative model setting more clear, if there are some other technical efforts to deal with this issue" -----
>
> Technically speaking, the challenge in analyzing finite horizon (stationary) offline setting is that we need to deal with data-dependence (the data are coming in the sequential fashion and the data at different time steps are aggregated together which causes statistical dependence) and the generative model does not have this technical issue since each data sample is independent from each other (e.g. [Sidford et al.]). To be more specific, we need to leverage the stationarity of the transition kernel in a careful manner for this variance reduction framework to achieve the tight result. In particular, the key quantity is the sum of conditional variance (Line 271) as it equals to $n_{s,a}\cdot \sigma_{V_t}(s,a)$. This is important since when $P_t$'s are different (non-stationarity), the sum of conditional variance can only be bouned by $n_{s,a}\cdot \max_{V_t}\sigma_{V_t}(s,a)$ (line 265-278). In contrast, the analysis for the generative model setting can directly apply Bernstein inequality to obtain the tight result directly without using such techniques (this is the same as our Theorem 4.3 since we adopt the i.i.d. protocol of [Chen and Jiang 2019]).

---

### Official Review · Reviewer_9aNL · 2021-07-17

**Rating:** 6
**Confidence:** 3

**Summary:**

The papers presents an algorithms called OPDVR (Off-Policy Double Variance Reduction) based on variance reduction, which is seemingly designed with offline RL in mind, even though the main attribute of offline RL (data constraint and lack of support) is fully missed. The paper contains several theoretical results; their core contribution (among others) is for the stationary case, where transition probabilities are not function of time. They basically present a sample complexity that grows with $H^2$, an improvement over previous results by a factor of $H$. The authors provide similar result for the infinite-horizon (discounted) case as well, and expand their findings for the non-stationary case too.

Main weakness: While this paper is a nice theoretical contribution, discussions on the "practical" impact of the basic assumptions, and how they can be relaxed to become useful in real-world, are missing, [see the comments below].

**Limitations And Societal Impact:**

A theoretical work; no direct societal impact.

**Main Review:**

-- The data coverage assumption, even though it is less strong than others in the literature, is still non-realistic. It must be remarked that in practice (where offline RL matters and indeed aims for), data is given; hence, any assumption about data is meaningless. As a matter of course, in almost all the problems that I know as someone who has been involved in real-world offline RL, only a tiny fraction of the state-action joint space is covered by the data. While I appreciate an interesting theoretical work like the results of this paper, the authors must note that the first step to make something useful in the offline RL literature is to set aside coverage assumptions. Note also that when dealing with situations under data constraints (which should be the definition of offline RL), optimality is no longer a concern, since we know it in advance that it is not feasible; yet in many domains of interest (like healthcare) even small data still contains valuable information and must be used. My strong recommendation always is to focus on how to use limited information in such data, if you hope that your algorithm goes beyond interesting mathematical proofs.

-- Line 157-160: A policy that satisfies monotonicity condition -> this needs explanation and example. In particular, if you are given an offline fixed data, how likely is this assumption to hold? Without such intuitions the results can easily become misleading.

-- No experimental result is given. Most importantly, you need to justify why someone should care about improving sample complexity by a factor of $H$. As a suggestion, you may start with some fixed dataset and shrink its size in a few stages by randomly removing trajectories from the dataset. And, show the final/best performance of your algorithm against others. Also note that in fixed data scenarios, convergence rate is not a concern for offline RL, almost always.

Minor: Line 87: in the definition of MDP, T is used for transition probability, but replaced by P right after.


**Time Spent Reviewing:**

5

---

> ### Author Response · Authors · 2021-08-10
> **Response for the reviewer 9aNL**
>
> We appreciate the reviewer for the feedbacks. We really enjoyed reading your comments and your practical perspectives are valuable for us. The followings are our detailed responses.
>
>
> ----- "The data coverage assumption ... any assumption about data is meaningless" -----
>
> The data coverage assumption for offline RL essentially mirrors the learnability notion of the supervised learning, i.e. the data are sampled from certain distribution that can cover the problem of interest / can generalize. There are (at least) two threads of theoretical work on offline learning: 1. try to learn valuable information from any given data (where the data could have limited sample size or insufficient coverage); 2. try to learn the optimal policy.
>
> There are a lot of works who follow the second thread  (e.g. [Xie. et al. 2020a,b, Liu et al. 2020b.] and this paper is an addition to them. For this thread of learning the optimal strategy, the data coverage assumption (of certain form) is needed for the purpose of statistical / mathematical derivation. Such coverage assumption also conveys practical messages on how good the dataset needs to be for it to be possible to learn the optimal policy in real-world scenarios. For example, for learning a good treatment policy in medical practices, the data needs to contain sufficient variations of different treatments in order for an offline algorithm to be able to learn.
>
>
>
>
>
> In addition, we fully agree with the reviewer that a precise theoretical understanding of the offline RL should ultimately achieve the "Assumption-Free RL", where the statistical result should tell what can we achieve when no assumption is made and should be able to learn the optimal policy when reduced to the case where data have the coverage property. As mentioned by the reviewer, ``make something useful in the offline RL literature is to set aside coverage assumptions" is our goal. As a step towards this goal, we weaken the previous assumption (e.g. uniform concentrability) and we appreciate it that the reviewer noticed this. Understanding the ultimate Assumption-free RL in the general sense remains an open question and is our future direction. For the first thread, please see the response below.
>
>
> ----- "yet in many domains of interest (like healthcare) even small data still contains valuable information and must be used. My strong recommendation ... use limited information in such data" ----
>
> First of all, we agree that learning from limited data is an area of great interest and there are numerous real-world problems where only limited historical data is available (e.g. treat ICU patients) and can relate to frameworks like contextual bandits or causal inferences. One other hand, there are also problems where the data are not limited and we wish to learn the optimal policy. Examples include deep learning based methods like imagine recognition or NLP tasks. Offline reinforcement learning is usually considered as the second case since one needs to plan over $H$ steps and, for $H$ large, accurately learning over the $H$ horizon (or $(1-\gamma)^{-1}$ effective horizon) is challenging due to the curse of horizon [Jiang and Agarwal, 2018] [Qiang Liu et al. 2018] and that is why our sharpening the horizon dependence is important for saving sample complexity.
>
> [1] [Jiang, Agarwal] Open Problem: The Dependence of Sample Complexity Lower Bounds on Planning Horizon, COLT, 2018
>
> [2] [Qiang Liu, Lihong Li, Ziyang Tang, Dengyong Zhou] Breaking the curse of horizon: Infinite-horizon off-policy estimation, NeurIPS, 2018
>
>
>
> ----- "A policy that satisfies monotonicity condition -> this needs explanation and example. In particular, if you are given an offline fixed data, how likely is this assumption to hold?" -----
>
>
> The monotonicity condition $V_t \leq \mathcal{T}{\pi_t} V_{t+1}$ is a joint condition on a (learned) policy $\pi$ as well as an (estimated) value function $V$, not a condition on the dataset. Our algorithm construction guarantees that this kind of monotonicity is true throughout the algorithm execution, so that guarantees on the estimated value function can be translated to guarantees on the learned policy. We will provide more explanations and examples in our revision.
>
> ----- "No experimental result is given." ------
>
> Thanks. We do agree a simulation study is helpful for the illustration purpose of our theoretical results and we are happy to add one as follows. To verify our OPDVR-stationary improves over the existing works in terms of scaling in $H$, we use a new stationary environment by modifying the MDP example used in [Yin et al. 2021] Section 6 (more concretely, in their Appendix I we can remove one version of the MDP to make it stationary but keep the reward assignments changing at each time step). We will compare the performance of OPDVR-stationary to OPEMA (from Yin et al.) and OPDVR-nonstationary (nonstationary algorithm can run in the stationary environment). We will follow [Yin et al. 2021] to use a fixed offline data set and test the methods on the varying $H$, but in our setting. In particular, there are universal constants in OPDVR that need to be identified (we regard them as the ``hyperparameters'' in our algorithm) and we will use cross validation (by data splitting). We will add this case study in our revision and we are happy to take any question regrading this in the later discussions.

---

### Decision · Program_Chairs · 2021-09-27

**Decision:**

Accept (Poster)

**Comment:**

The submission proposes OPDVR for offline policy optimization, and sample complexities for finite-horizon case and discounted infinite-horizon case are presented. Though some concerns on the pratical side are proposed, all reviewers agree that the paper has a great theoretical contribution. Thus I recommend accept, and also encourage the authors to have some pratical results in the camera ready version.